# Self-Supervised Learning with Data Augmentations Provably Isolates Content from Style

**Julius von Kügelgen**[*1,2]  **Yash Sharma**[*3,4]  **Luigi Gresele**[*1]

**Wieland Brendel**[3]  **Bernhard Schölkopf**[†1]  **Michel Besserve**[†1]  **Francesco Locatello**[†5]

[1] Max Planck Institute for Intelligent Systems Tübingen  [2] University of Cambridge
[3] Tübingen AI Center, University of Tübingen  [4] IMPRS for Intelligent Systems  [5] Amazon

## Abstract

Self-supervised representation learning has shown remarkable success in a number of domains. A common practice is to perform data augmentation via hand-crafted transformations intended to leave the semantics of the data invariant. We seek to understand the empirical success of this approach from a theoretical perspective. We formulate the augmentation process as a latent variable model by postulating a partition of the latent representation into a *content* component, which is assumed invariant to augmentation, and a *style* component, which is allowed to change. Unlike prior work on disentanglement and independent component analysis, we allow for both nontrivial statistical and causal dependencies in the latent space. We study the identifiability of the latent representation based on pairs of views of the observations and prove sufficient conditions that allow us to identify the invariant content partition up to an invertible mapping in both generative and discriminative settings. We find numerical simulations with dependent latent variables are consistent with our theory. Lastly, we introduce *Causal3DIdent*, a dataset of high-dimensional, visually complex images with rich causal dependencies, which we use to study the effect of data augmentations performed in practice.

## 1 Introduction

Learning good representations of high-dimensional observations from large amounts of unlabelled data is widely recognised as an important step for more capable and data-efficient learning systems [10, 72]. Over the last decade, *self-supervised learning* (SSL) has emerged as the dominant paradigm for such unsupervised representation learning [1, 20, 21, 34, 41, 47, 48, 90, 91, 115, 122, 125, 126]. The main idea behind SSL is to extract a supervisory signal from unlabelled observations by leveraging known structure of the data, which allows for the application of supervised learning techniques. A common approach is to directly predict some part of the observation from another part (e.g., future from past, or original from corruption), thus forcing the model to learn a meaningful representation in the process. While this technique has shown remarkable success in natural language processing [13, 23, 30, 81, 84, 86, 95, 99] and speech recognition [5, 6, 100, 104], where a finite dictionary allows one to output a distribution over the missing part, such *predictive* SSL methods are not easily applied to continuous or high-dimensional domains such as vision. Here, a common approach is to learn a *joint embedding* of similar observations or *views* such that their representation is close [7, 12, 22, 44]. Different views can come, for example, from different modalities (text & speech; video & audio) or time points. As still images lack such multi-modality or temporal structure, recent advances in representation learning have relied on generating similar views by means of *data augmentation*.

---

[*]Joint first author. [†]Joint senior author. Correspondence to: jvk@tue.mpg.de

Code available at: https://www.github.com/ysharma1126/ssl_identifiability

35th Conference on Neural Information Processing Systems (NeurIPS 2021).

In order to be useful, data augmentation is thought to require the transformations applied to generate additional views to be generally chosen to *preserve the semantic characteristics* of an observation, while changing other "nuisance" aspects. While this intuitively makes sense and has shown remarkable empirical results, the success of data augmentation techniques in practice is still not very well understood from a theoretical perspective—despite some efforts [17, 19, 28]. In the present work, we seek to better understand the empirical success of SSL with data augmentation by formulating the generative process as a latent variable model (LVM) and studying *identifiability* of the representation, i.e., under which conditions the ground truth latent factors can provably be inferred from the data [77].

**Related work and its relation to the current.** Prior work on unsupervised representation learning from an LVM perspective often postulates *mutually independent latent factors*: this independence assumption is, for example, at the heart of independent component analysis (ICA) [24, 56] and disentanglement [10, 14, 18, 49, 65, 71]. Since it is impossible to identify the true latent factors without any supervisory signal in the general nonlinear case [57, 82], recent work has turned to weakly- or self-supervised approaches which leverage additional information in the form of multiple views [39, 83, 108, 129], auxiliary variables [58, 63], or temporal structure [45, 54, 55, 69]. To identify or disentangle the individual independent latent factors, it is typically assumed that there is a chance that *each factor changes* across views, environments, or time points.

Our work—being directly motivated by common practices in SSL with data augmentation—differs from these works in the following two key aspects (see Fig. 1 for an overview). First, we do not assume independence and instead *allow for both nontrivial statistical and causal relations between latent variables*. This is in line with a recently proposed [105] shift towards causal representation learning [40, 76, 85, 87, 106, 107, 112, 123, 127], motivated by the fact that many underlying variables of interest may not be independent but causally related to each other.[1] Second, instead of a scenario wherein all latent factors may change as a result of augmentation, we assume a *partition of the latent space* into two blocks: a *content* block which is shared or *invariant* across different augmented views, and a *style* block that *may change*. This is aligned with the notion that augmentations leave certain semantic aspects (i.e., content) intact and only affect style, and is thus a more appropriate assumption for studying SSL. In line with earlier work [39, 54, 57, 58, 63, 69, 82, 83, 129], we focus on the setting of continuous ground-truth latents, though we believe our results to hold more broadly.

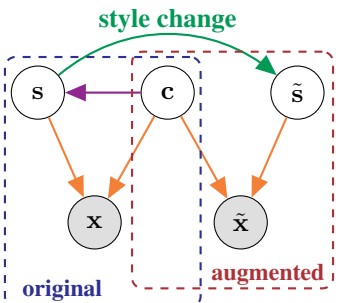

Figure 1: **Overview of our problem formulation.** We partition the latent variable $\mathbf{z}$ into content $\mathbf{c}$ and style $\mathbf{s}$, and allow for statistical and causal dependence of style on content. We assume that only style changes between the original view $\mathbf{x}$ and the augmented view $\tilde{\mathbf{x}}$, i.e., they are obtained by applying the same deterministic function $\mathbf{f}$ to $\mathbf{z} = (\mathbf{c}, \mathbf{s})$ and $\tilde{\mathbf{z}} = (\mathbf{c}, \tilde{\mathbf{s}})$.

**Structure and contributions.** Following a review of SSL with data augmentation and identifiability theory (§ 2), we formalise the process of data generation and augmentation as an LVM with content and style variables (§ 3). We then establish identifiability results of the invariant content partition (§ 4), validate our theoretical insights experimentally (§ 5), and discuss our findings and their limitations in the broader context of SSL with data augmentation (§ 6). We highlight the following contributions:

- we prove that SSL with data augmentations identifies the invariant content partition of the representation in generative (Thm. 4.2) and discriminative learning with invertible (Thm. 4.3) and non-invertible encoders with entropy regularisation (Thm. 4.4); in particular, Thm. 4.4 provides a theoretical justification for the empirically observed effectiveness of contrastive SSL methods that use data augmentation and InfoNCE [91] as an objective, such as SimCLR [20];

- we show that our theory is consistent with results in simulating statistical dependencies within blocks of content and style variables, as well as with style causally dependent on content (§ 5.1);

- we introduce *Causal3DIdent*, a dataset of 3D objects which allows for the study of identifiability in a causal representation learning setting, and use it to perform a systematic study of data augmentations used in practice, yielding novel insights on what particular data augmentations are truly isolating as invariant content and discarding as varying style when applied (§ 5.2).

---

[1]E.g., [69], Fig. 11 where dependence between latents was demonstrated for multiple natural video data sets.

## 2   Preliminaries and background

**Self-supervised representation learning with data augmentation.**   Given an unlabelled dataset of observations (e.g., images) $\mathbf{x}$, data augmentation techniques proceed as follows. First, a set of observation-level transformations $\mathbf{t} \in \mathcal{T}$ are specified together with a distribution $p_\mathbf{t}$ over $\mathcal{T}$. Both $\mathcal{T}$ and $p_\mathbf{t}$ are typically designed using human intelligence and domain knowledge with the intention of *not changing the semantic characteristics* of the data (which arguably constitutes a form of weak supervision).[2] For images, for example, a common choice for $\mathcal{T}$ are combinations of random crops [113], horizontal or vertical flips, blurring, colour distortion [52, 113], or cutouts [31]; and $p_\mathbf{t}$ is a distribution over the parameterisation of these transformations, e.g., the centre and size of a crop [20, 31]. For each observation $\mathbf{x}$, a pair of transformations $\mathbf{t}, \mathbf{t}' \sim p_\mathbf{t}$ is sampled and applied separately to $\mathbf{x}$ to generate a pair of augmented views $(\tilde{\mathbf{x}}, \tilde{\mathbf{x}}') = (\mathbf{t}(\mathbf{x}), \mathbf{t}'(\mathbf{x}))$.

The joint-embedding approach to SSL then uses a pair of encoder functions $(\mathbf{g}, \mathbf{g}')$, i.e. deep nets, to map the pair $(\tilde{\mathbf{x}}, \tilde{\mathbf{x}}')$ to a typically lower-dimensional representation $(\tilde{\mathbf{z}}, \tilde{\mathbf{z}}') = (\mathbf{g}(\tilde{\mathbf{x}}), \mathbf{g}'(\tilde{\mathbf{x}}'))$. Often, the two encoders are either identical, $\mathbf{g} = \mathbf{g}'$, or directly related (e.g., via shared parameters or asynchronous updates). Then, the encoder(s) $(\mathbf{g}, \mathbf{g}')$ are trained such that the representations $(\tilde{\mathbf{z}}, \tilde{\mathbf{z}}')$ are "close", i.e., such that $\mathrm{sim}(\tilde{\mathbf{z}}, \tilde{\mathbf{z}}')$ is large for some similarity metric $\mathrm{sim}(\cdot)$, e.g., the cosine similarity [20, 129], or negative L2 norm [129]. The advantage of directly optimising for similarity in representation space over generative alternatives is that reconstruction can be very challenging for high-dimensional data. The disadvantage is the problem of *collapsed representations*.[3] To avoid collapsed representations and force the encoder(s) to learn a meaningful representation, two main families of approaches have been used: (i) *contrastive learning* (CL) [20, 47, 48, 91, 115, 126]; and (ii) *regularisation-based* SSL [21, 41, 128].

The idea behind CL is to not only learn similar representations for augmented views $(\tilde{\mathbf{x}}_i, \tilde{\mathbf{x}}'_i)$ of the same $\mathbf{x}_i$, or *positive pairs*, but to also use other observations $\mathbf{x}_j$ $(j \neq i)$ to contrast with, i.e., to enforce a dissimilar representation across *negative pairs* $(\tilde{\mathbf{x}}_i, \tilde{\mathbf{x}}'_j)$. In other words, CL pulls representations of positive pairs together, and pushes those of negative pairs apart. Since both aims cannot be achieved simultaneously with a constant representation, collapse is avoided. A popular CL objective function (used, e.g., in SimCLR [20]) is InfoNCE [91] (based on noise-contrastive estimation [42, 43]):

$$\mathcal{L}_{\text{InfoNCE}}(\mathbf{g}; \tau, K) = \mathbb{E}_{\{\mathbf{x}_i\}_{i=1}^K \sim p_\mathbf{x}} \left[ -\sum_{i=1}^K \log \frac{\exp\{\mathrm{sim}(\tilde{\mathbf{z}}_i, \tilde{\mathbf{z}}'_i)/\tau\}}{\sum_{j=1}^K \exp\{\mathrm{sim}(\tilde{\mathbf{z}}_i, \tilde{\mathbf{z}}'_j)/\tau\}} \right] \tag{1}$$

where $\tilde{\mathbf{z}} = \mathbb{E}_{\mathbf{t} \sim p_\mathbf{t}}[\mathbf{g}(\mathbf{t}(\mathbf{x}))]$, $\tau$ is a temperature, and $K-1$ is the number of negative pairs. InfoNCE (1) has an interpretation as multi-class logistic regression, and lower bounds the mutual information across similar views $(\tilde{\mathbf{z}}, \tilde{\mathbf{z}}')$—a common representation learning objective [4, 9, 15, 50, 75, 79, 80, 97, 120]. Moreover, (1) can be interpreted as *alignment* (numerator) and *uniformity* (denominator) terms, the latter constituting a nonparametric entropy estimator of the representation as $K \to \infty$ [124]. CL with InfoNCE can thus be seen as alignment of positive pairs with (approximate) entropy regularisation.

Instead of using negative pairs, as in CL, a set of recent SSL methods only optimise for alignment and avoid collapsed representations through different forms of regularisation. For example, BYOL [41] and SimSiam [21] rely on "architectural regularisation" in the form of moving-average updates for a separate "target" net $\mathbf{g}'$ (BYOL only) or a stop-gradient operation (both). BarlowTwins [128], on the other hand, optimises the cross correlation between $(\tilde{\mathbf{z}}, \tilde{\mathbf{z}}')$ to be close to the identity matrix, thus enforcing redundancy reduction (zero off-diagonals) in addition to alignment (ones on the diagonal).

**Identifiability of learned representations.**   In this work, we address the question of whether SSL with data augmentation can reveal or uncover properties of the underlying data generating process. Whether a representation learned from observations can be expected to match the true underlying latent factors—up to acceptable ambiguities and subject to suitable assumptions on the generative process and inference model—is captured by the notion of identifiability [77].

Within representation learning, identifiability has mainly been studied in the framework of (nonlinear) ICA which assumes a model of the form $\mathbf{x} = \mathbf{f}(\mathbf{z})$ and aims to recover the independent latents, or *sources*, $\mathbf{z}$, typically up to permutation or element-wise transformation. A crucial negative result states that, with i.i.d. data and without further assumptions, this is fundamentally impossible [57]. However, recent breakthroughs have shown that identifiability can be achieved if an auxiliary variable (e.g.,

---

[2]Note that recent work has investigated automatically discovering good augmentations [26, 27].

[3]If the only goal is to make representations of augmented views similar, a degenerate solution which simply maps any observation to the origin trivially achieves this goal.

a time stamp or environment index) renders the sources *conditionally* independent [45, 54, 55, 58]. These methods rely on constructing positive and negative pairs using the auxiliary variable and learning a representation with CL. This development has sparked a renewed interest in identifiability in the context of deep representation learning [63, 64, 69, 83, 102, 108, 109, 129].

Most closely related to SSL with data augmentation are works which study identifiability when given a second view $\tilde{\mathbf{x}}$ of an observation $\mathbf{x}$, resulting from a modified version $\tilde{\mathbf{z}}$ of the underlying latents or sources $\mathbf{z}$ [39, 69, 83, 101, 108, 129]. Here, $\tilde{\mathbf{z}}$ is either an element-wise corruption of $\mathbf{z}$ [39, 69, 101, 129] or may share a random subset of its components [83, 108]. Crucially, all previously mentioned works assume that *any* of the independent latents (are allowed to) change, and aim to identify the individual factors. However, in the context of SSL with data augmentation, where the semantic (content) part of the representation is intended to be shared between views, this assumption does not hold.

## 3 Problem formulation

We specify our problem setting by formalising the processes of data generation and augmentation. We take a latent-variable model perspective and assume that observations $\mathbf{x}$ (e.g., images) are generated by a *mixing* function $\mathbf{f}$ which takes a latent code $\mathbf{z}$ as input. Importantly, we describe the augmentation process through changes in this latent space as captured by a conditional distribution $p_{\tilde{\mathbf{z}}|\mathbf{z}}$, as opposed to traditionally describing the transformations $\mathbf{t}$ as acting directly at the observation level.

Formally, let $\mathbf{z}$ be a continuous r.v. taking values in an open, simply-connected $n$-dim. *representation space* $\mathcal{Z} \subseteq \mathbb{R}^n$ with associated probability density $p_{\mathbf{z}}$. Moreover, let $\mathbf{f} : \mathcal{Z} \to \mathcal{X}$ be a *smooth and invertible* mapping to an *observation space* $\mathcal{X} \subseteq \mathbb{R}^d$ and let $\mathbf{x}$ be the continuous r.v. defined as $\mathbf{x} = \mathbf{f}(\mathbf{z})$.[4] The generative process for the dataset of original observations of $\mathbf{x}$ is thus given by:

$$\mathbf{z} \sim p_{\mathbf{z}}, \qquad \mathbf{x} = \mathbf{f}(\mathbf{z}). \tag{2}$$

Next, we formalise the data augmentation process. As stated above, we take a representation-centric view, i.e., we assume that an augmentation $\tilde{\mathbf{x}}$ of the original $\mathbf{x}$ is obtained by applying the same mixing or rendering function $\mathbf{f}$ to a modified representation $\tilde{\mathbf{z}}$ which is (stochastically) related to the original representation $\mathbf{z}$ of $\mathbf{x}$. Specifying the effect of data augmentation thus corresponds to specifying a conditional distribution $p_{\tilde{\mathbf{z}}|\mathbf{z}}$ which captures the relation between $\mathbf{z}$ and $\tilde{\mathbf{z}}$.

In terms of the transformation-centric view presented in § 2, we can view the modified representation $\tilde{\mathbf{z}} \in \mathcal{Z}$ as obtained by applying $\mathbf{f}^{-1}$ to a transformed observation $\tilde{\mathbf{x}} = \mathbf{t}(\mathbf{x}) \in \mathcal{X}$ where $\mathbf{t} \sim p_{\mathbf{t}}$, i.e., $\tilde{\mathbf{z}} = \mathbf{f}^{-1}(\tilde{\mathbf{x}})$. The conditional distribution $p_{\tilde{\mathbf{z}}|\mathbf{z}}$ in the representation space can thus be viewed as being induced by the distribution $p_{\mathbf{t}}$ over transformations applied at the observation level.[5]

We now encode the notion that the set of transformations $\mathcal{T}$ used for augmentation is typically chosen such that any transformation $\mathbf{t} \in \mathcal{T}$ leaves certain aspects of the data invariant. To this end, we assume that *the representation $\mathbf{z}$ can be uniquely partitioned into two disjoint parts*:

   (i) an *invariant* part $\mathbf{c}$ which will *always be shared* across $(\mathbf{z}, \tilde{\mathbf{z}})$, and which we refer to as *content*;
   (ii) a *varying* part $\mathbf{s}$ which *may change* across $(\mathbf{z}, \tilde{\mathbf{z}})$, and which we refer to as *style*.

We assume that $\mathbf{c}$ and $\mathbf{s}$ take values in content and style subspaces $\mathcal{C} \subseteq \mathbb{R}^{n_c}$ and $\mathcal{S} \subseteq \mathbb{R}^{n_s}$, respectively, i.e., $n = n_c + n_s$ and $\mathcal{Z} = \mathcal{C} \times \mathcal{S}$. W.l.o.g., we let $\mathbf{c}$ corresponds to the first $n_c$ dimensions of $\mathbf{z}$:

$$\mathbf{z} = (\mathbf{c}, \mathbf{s}), \qquad \mathbf{c} := \mathbf{z}_{1:n_c}, \qquad \mathbf{s} := \mathbf{z}_{(n_c+1):n},$$

We formalise the process of data augmentation with content-preserving transformations by defining the conditional $p_{\tilde{\mathbf{z}}|\mathbf{z}}$ such that only a (random) subset of the style variables change at a time.

**Assumption 3.1** (Content-invariance). The conditional density $p_{\tilde{\mathbf{z}}|\mathbf{z}}$ over $\mathcal{Z} \times \mathcal{Z}$ takes the form

$$p_{\tilde{\mathbf{z}}|\mathbf{z}}(\tilde{\mathbf{z}}|\mathbf{z}) = \delta(\tilde{\mathbf{c}} - \mathbf{c})p_{\tilde{\mathbf{s}}|\mathbf{s}}(\tilde{\mathbf{s}}|\mathbf{s})$$

for some continuous density $p_{\tilde{\mathbf{s}}|\mathbf{s}}$ on $\mathcal{S} \times \mathcal{S}$, where $\delta(\cdot)$ is the Dirac delta function, i.e., $\tilde{\mathbf{c}} = \mathbf{c}$ a.e.

**Assumption 3.2** (Style changes). Let $\mathcal{A}$ be the set of subsets of style variables $A \subseteq \{1, ..., n_s\}$ and let $p_A$ be a distribution on $\mathcal{A}$. Then, the style conditional $p_{\tilde{\mathbf{s}}|\mathbf{s}}$ is obtained via

$$A \sim p_A, \qquad p_{\tilde{\mathbf{s}}|\mathbf{s},A}(\tilde{\mathbf{s}}|\mathbf{s}, A) = \delta(\tilde{\mathbf{s}}_{A^c} - \mathbf{s}_{A^c})p_{\tilde{\mathbf{s}}_A|\mathbf{s}_A}(\tilde{\mathbf{s}}_A|\mathbf{s}_A),$$

where $p_{\tilde{\mathbf{s}}_A|\mathbf{s}_A}$ is a continuous density on $\mathcal{S}_A \times \mathcal{S}_A$, $\mathcal{S}_A \subseteq \mathcal{S}$ denotes the subspace of changing style variables specified by $A$, and $A^c = \{1, ..., n_s\} \setminus A$ denotes the complement of $A$.

---

[4]While $\mathbf{x}$ may be high-dimensional $n \ll d$, invertibility of $\mathbf{f}$ implies that $\mathcal{X}$ is an $n$-dim. sub-manifold of $\mathbb{R}^d$.

[5]We investigate this correspondence between changes in observation and latent space empirically in § 5.

Note that Assumption 3.2 is less restrictive than assuming that all style variables need to change, since it also allows for only a (possibly different) subset of style variables to change for any given observation. This is in line with the intuition that not all transformations affect all changeable (i.e., style) properties of the data: e.g., a colour distortion should not affect positional information, and, in the same vein, a (horizontal or vertical) flip should not affect the colour spectrum.

The generative process of an augmentation or transformed observation $\tilde{\mathbf{x}}$ is thus given by

$$A \sim p_A, \qquad \tilde{\mathbf{z}}|\mathbf{z}, A \sim p_{\tilde{\mathbf{z}}|\mathbf{z},A}, \qquad \tilde{\mathbf{x}} = \mathbf{f}(\tilde{\mathbf{z}}). \qquad (3)$$

Our setting for modelling data augmentation differs from that commonly assumed in (multi-view) disentanglement and ICA in that *we do not assume that the latent factors* $\mathbf{z} = (\mathbf{c}, \mathbf{s})$ *are mutually (or conditionally) independent*, i.e., we allow for *arbitrary* (non-factorised) marginals $p_{\mathbf{z}}$ in (2).[6]

**Causal interpretation: data augmentation as counterfactuals under soft style intervention.** We now provide a causal account of the above data generating process by describing the (allowed) causal dependencies among latent variables using a structural causal model (SCM) [94]. As we will see, this leads to an interpretation of data augmentations as counterfactuals in the underlying latent SCM. The assumption that $\mathbf{c}$ stays invariant as $\mathbf{s}$ changes is consistent with the view that content may causally influence style, $\mathbf{c} \to \mathbf{s}$, but not vice versa, see Fig. 1. We therefore formalise their relation as:

$$\mathbf{c} := \mathbf{f_c}(\mathbf{u_c}), \qquad \mathbf{s} := \mathbf{f_s}(\mathbf{c}, \mathbf{u_s}), \qquad (\mathbf{u_c}, \mathbf{u_s}) \sim p_{\mathbf{u_c}} \times p_{\mathbf{u_s}}$$

where $\mathbf{u_c}, \mathbf{u_s}$ are independent exogenous variables, and $\mathbf{f_c}, \mathbf{f_s}$ are deterministic functions. The latent causal variables $(\mathbf{c}, \mathbf{s})$ are subsequently decoded into observations $\mathbf{x} = \mathbf{f}(\mathbf{c}, \mathbf{s})$. Given a factual observation $\mathbf{x}^{\mathrm{F}} = \mathbf{f}(\mathbf{c}^{\mathrm{F}}, \mathbf{s}^{\mathrm{F}})$ which resulted from $(\mathbf{u_c^F}, \mathbf{u_s^F})$, we may ask the counterfactual question: "*what would have happened if the style variables had been (randomly) perturbed, all else being equal?*". Consider, e.g., a *soft intervention* [35] on $\mathbf{s}$, i.e., an intervention that changes the mechanism $\mathbf{f_s}$ to

$$do(\mathbf{s} := \tilde{\mathbf{f}}_\mathbf{s}(\mathbf{c}, \mathbf{u_s}, \mathbf{u}_A)),$$

where $\mathbf{u}_A$ is an additional source of stochasticity accounting for the randomness of the augmentation process ($p_A \times p_{\tilde{\mathbf{s}}|\mathbf{s},A}$). The resulting distribution over counterfactual observations $\mathbf{x}^{\mathrm{CF}} = \mathbf{f}(\mathbf{c}^{\mathrm{F}}, \mathbf{s}^{\mathrm{CF}})$ can be computed from the modified SCM by fixing the exogenous variables to their factual values and performing the soft intervention:

$$\mathbf{c}^{\mathrm{CF}} := \mathbf{c}^{\mathrm{F}}, \qquad \mathbf{s}^{\mathrm{CF}} := \tilde{\mathbf{f}}_\mathbf{s}(\mathbf{c}^{\mathrm{F}}, \mathbf{u_s^F}, \mathbf{u}_A), \qquad \mathbf{u}_A \sim p_{\mathbf{u}_A}.$$

This aligns with our intuition and assumed problem setting of data augmentations as style corruptions. We note that the notion of augmentation as (hard) style interventions is also at the heart of ReLIC [87], a recently proposed, causally-inspired SSL regularisation term for instance-discrimination [44, 126]. However, ReLIC assumes independence between content and style and does not address identifiability. For another causal perspective on data augmentation in the context of domain generalisation, c.f. [59].

## 4 Theory: block-identifiability of the invariant content partition

Our goal is to prove that we can identify the invariant content partition $\mathbf{c}$ under a distinct, weaker set of assumptions, compared to existing results in disentanglement and nonlinear ICA [39, 69, 83, 108, 129]. We stress again that our primary interest is not to identify or disentangle individual (and independent) latent factors $z_j$, but instead to separate content from style, such that the content variables can be subsequently used for downstream tasks. We first define this distinct notion of *block-identifiability*.

**Definition 4.1** (Block-identifiability). We say that the true content partition $\mathbf{c} = \mathbf{f}^{-1}(\mathbf{x})_{1:n_c}$ is *block-identified* by a function $\mathbf{g} : \mathcal{X} \to \mathcal{Z}$ if the inferred content partition $\hat{\mathbf{c}} = \mathbf{g}(\mathbf{x})_{1:n_c}$ contains *all* and *only* information about $\mathbf{c}$, i.e., if there exists an *invertible* function $\mathbf{h} : \mathbb{R}^{n_c} \to \mathbb{R}^{n_c}$ s.t. $\hat{\mathbf{c}} = \mathbf{h}(\mathbf{c})$.

Defn. 4.1 is related to independent subspace analysis [16, 53, 73, 114], which also aims to identify blocks of random variables as opposed to individual factors, though under an *independence assumption across blocks*, and typically not within a multi-view setting as studied in the present work.

### 4.1 Generative self-supervised representation learning

First, we consider *generative* SSL, i.e., fitting a generative model to pairs $(\mathbf{x}, \tilde{\mathbf{x}})$ of original and augmented views.[7] We show that under our specified data generation and augmentation process (§ 3),

---

[6]The recently proposed Independently *Modulated* Component Analysis (IMCA) [64] extension of ICA is a notable exception, but only allows for trivial dependencies across $\mathbf{z}$ in the form of a shared base measure.

[7]For notational simplicity, we present our theory for pairs $(\mathbf{x}, \tilde{\mathbf{x}})$ rather than for two augmented views $(\tilde{\mathbf{x}}, \tilde{\mathbf{x}}')$, as typically used in practice but it also holds for the latter, see § 6 for further discussion.

as well as suitable additional assumptions (stated and discussed in more detail below), it is possible to isolate (i.e., block-identify) the invariant content partition. Full proofs are included in Appendix A.

**Theorem 4.2** (Identifying content with a generative model). *Consider the data generating process described in § 3, i.e., the pairs $(\mathbf{x}, \tilde{\mathbf{x}})$ of original and augmented views are generated according to (2) and (3) with $p_{\tilde{\mathbf{z}}|\mathbf{z}}$ as defined in Assumptions 3.1 and 3.2. Assume further that*

*(i) $\mathbf{f} : \mathcal{Z} \to \mathcal{X}$ is smooth and invertible with smooth inverse (i.e., a diffeomorphism);*

*(ii) $p_{\mathbf{z}}$ is a smooth, continuous density on $\mathcal{Z}$ with $p_{\mathbf{z}}(\mathbf{z}) > 0$ almost everywhere;*

*(iii) for any $l \in \{1, ..., n_s\}$, $\exists A \subseteq \{1, ..., n_s\}$ s.t. $l \in A$; $p_A(A) > 0$; $p_{\tilde{\mathbf{s}}_A|\mathbf{s}_A}$ is smooth w.r.t. both $\mathbf{s}_A$ and $\tilde{\mathbf{s}}_A$; and for any $\mathbf{s}_A$, $p_{\tilde{\mathbf{s}}_A|\mathbf{s}_A}(\cdot|\mathbf{s}_A) > 0$ in some open, non-empty subset containing $\mathbf{s}_A$.*

*If, for a given $n_s$ ($1 \le n_s < n$), a generative model $(\hat{p}_{\mathbf{z}}, \hat{p}_A, \hat{p}_{\tilde{\mathbf{s}}|\mathbf{s}, A}, \hat{\mathbf{f}})$ assumes the same generative process (§ 3), satisfies the above assumptions (i)-(iii), and matches the data likelihood,*

$$p_{\mathbf{x}, \tilde{\mathbf{x}}}(\mathbf{x}, \tilde{\mathbf{x}}) = \hat{p}_{\mathbf{x}, \tilde{\mathbf{x}}}(\mathbf{x}, \tilde{\mathbf{x}}) \qquad \forall (\mathbf{x}, \tilde{\mathbf{x}}) \in \mathcal{X} \times \mathcal{X},$$

*then it block-identifies the true content variables via $\mathbf{g} = \hat{\mathbf{f}}^{-1}$ in the sense of Defn. 4.1.*

**Proof sketch.** First, show (using *(i)* and the matching likelihoods) that the representation $\hat{\mathbf{z}} = \mathbf{g}(\mathbf{x})$ extracted by $\mathbf{g}$ is related to the true $\mathbf{z}$ by a smooth invertible mapping $\mathbf{h} = \mathbf{g} \circ \mathbf{f}$ such that $\hat{\mathbf{c}} = \mathbf{h}(\mathbf{z})_{1:n_c}$ is invariant across $(\mathbf{z}, \tilde{\mathbf{z}})$ almost surely w.r.t. $p_{\mathbf{z}, \tilde{\mathbf{z}}}$.[8] Second, show by contradiction (using *(ii)*, *(iii)*) that $\mathbf{h}(\cdot)_{1:n_c}$ can, in fact, only depend on the true content $\mathbf{c}$ and not on style $\mathbf{s}$, for otherwise the invariance from step 1 would be violated in a region of the style (sub)space of measure greater than zero.

**Intuition.** Thm. 4.2 assumes that the number of content ($n_c$) and style ($n_s$) variables is known, and that there is a positive probability that each *style* variable may change, though not necessarily on its own, according to *(iii)*. In this case, training a generative model of the form specified in § 3 (i.e., with an invariant content partition and subsets of changing style variables) by maximum likelihood on pairs $(\mathbf{x}, \tilde{\mathbf{x}})$ will asymptotically (in the limit of infinite data) recover the true invariant content partition up to an invertible function, i.e., it isolates, or unmixes, content from style.

**Discussion.** The identifiability result of Thm. 4.2 for generative SSL is of potential relevance for existing variational autoencoder (VAE) [68] variants such as the `GroupVAE` [51],[9] or its adaptive version `AdaGVAE` [83]. Since, contrary to existing results, Thm. 4.2 does not assume independent latents, it may also provide a principled basis for generative causal representation learning algorithms [76, 107, 127]. However, an important limitation to its practical applicability is that generative modelling does not tend to scale very well to complex high-dimensional observations, such as images.

## 4.2 Discriminative self-supervised representation learning

We therefore next turn to a discriminative approach, i.e., directly learning an encoder function $\mathbf{g}$ which leads to a similar embedding across $(\mathbf{x}, \tilde{\mathbf{x}})$. As discussed in § 2, this is much more common for SSL with data augmentations. First, we show that if an invertible encoder $\mathbf{g}$ is used, then learning a representation which is aligned in the first $n_c$ dimensions is sufficient to block-identify content.

**Theorem 4.3** (Identifying content with an invertible encoder). *Assume the same data generating process (§ 3) and conditions (i)-(iv) as in Thm. 4.2. Let $\mathbf{g} : \mathcal{X} \to \mathcal{Z}$ be any smooth and* invertible *function which minimises the following functional:*

$$\mathcal{L}_{\text{Align}}(\mathbf{g}) := \mathbb{E}_{(\mathbf{x}, \tilde{\mathbf{x}}) \sim p_{\mathbf{x}, \tilde{\mathbf{x}}}} \left[ \left\| \mathbf{g}(\mathbf{x})_{1:n_c} - \mathbf{g}(\tilde{\mathbf{x}})_{1:n_c} \right\|_2^2 \right] \tag{4}$$

*Then $\mathbf{g}$ block-identifies the true content variables in the sense of Definition 4.1.*

**Proof sketch.** First, we show that the global minimum of (4) is reached by the smooth invertible function $\mathbf{f}^{-1}$. Thus, any other minimiser $\mathbf{g}$ must satisfy the same invariance across $(\mathbf{x}, \tilde{\mathbf{x}})$ used in step 1 of the proof of Thm. 4.2. The second step uses the same argument by contradiction as in Thm. 4.2.

**Intuition.** Thm. 4.3 states that if—under the same assumptions on the generative process as in Thm. 4.2—we directly learn a representation with an *invertible* encoder, then enforcing alignment between the first $n_c$ latents is sufficient to isolate the invariant content partition. Intuitively, invertibility guarantees that all information is preserved, thus avoiding a collapsed representation.

---

[8]This step is partially inspired by [83]; the technique used to prove the second *main* step is entirely novel.

[9]which also uses a fixed content-style partition for multi-view data, but assumes that all latent factors are mutually independent, and that all style variables change between views, independent of the original style;

**Discussion.** According to Thm. 4.3, content can be isolated if, e.g., a flow-based architecture [32, 33, 67, 92, 93] is used, or invertibility is enforced otherwise during training [8, 60]. However, the applicability of this approach is limited as it *places strong constraints on the encoder architecture which makes it hard to scale these methods up to high-dimensional settings.* As discussed in § 2, state-of-the-art SSL methods such as SimCLR [20], BYOL [41], SimSiam [21], or BarlowTwins [128] do not use invertible encoders, but instead avoid collapsed representations—which would result from naively optimising (4) for arbitrary, non-invertible $\mathbf{g}$—using different forms of regularisation.

To close this gap between theory and practice, finally, we investigate how to block-identify content without assuming an invertible encoder. We show that, if we add a regularisation term to (4) that encourages maximum entropy of the learnt representation, the invertibility assumption can be dropped.

**Theorem 4.4** (Identifying content with discriminative learning and a non-invertible encoder). *Assume the same data generating process (§ 3) and conditions (i)-(iv) as in Thm. 4.2. Let $\mathbf{g} : \mathcal{X} \to (0,1)^{n_c}$ be any smooth function which minimises the following functional:*

$$\mathcal{L}_{\text{AlignMaxEnt}}(\mathbf{g}) := \mathbb{E}_{(\mathbf{x},\tilde{\mathbf{x}})\sim p_{\mathbf{x},\tilde{\mathbf{x}}}} \left[ \left|\left| \mathbf{g}(\mathbf{x}) - \mathbf{g}(\tilde{\mathbf{x}}) \right|\right|_2^2 \right] - H\left(\mathbf{g}(\mathbf{x})\right) \tag{5}$$

*where $H(\cdot)$ denotes the differential entropy of the random variable $\mathbf{g}(\mathbf{x})$ taking values in $(0,1)^{n_c}$. Then $\mathbf{g}$ block-identifies the true content variables in the sense of Defn. 4.1.*

**Proof sketch.** First, use the Darmois construction [29, 57] to build a function $\mathbf{d} : \mathcal{C} \to (0,1)^{n_c}$ mapping $\mathbf{c} = \mathbf{f}^{-1}(\mathbf{x})_{1:n_c}$ to a uniform random variable. Then $\mathbf{g}^\star = \mathbf{d} \circ \mathbf{f}_{1:n_c}^{-1}$ attains the global minimum of (5) because $\mathbf{c}$ is invariant across $(\mathbf{x},\tilde{\mathbf{x}})$ and the uniform distribution is the maximum entropy distribution on $(0,1)^{n_c}$. Thus, any other minimiser $\mathbf{g}$ of (5) must satisfy invariance across $(\mathbf{x},\tilde{\mathbf{x}})$ and map to a uniform r.v. Then, use the same step 2 as in Thms. 4.2 and 4.3 to show that $\mathbf{h} = \mathbf{g} \circ \mathbf{f} : \mathcal{Z} \to (0,1)^{n_c}$ cannot depend on style, i.e., it is a function from $\mathcal{C}$ to $(0,1)^{n_c}$. Finally, we show that $\mathbf{h}$ must be invertible since it maps $p_{\mathbf{c}}$ to a uniform distribution, using a result from [129]. $\blacksquare$

**Intuition.** Thm. 4.4 states that if we do not explicitly enforce invertibility of $\mathbf{g}$ as in Thm. 4.3, additionally maximising the entropy of the learnt representation (i.e., optimising alignment *and* uniformity [124]) avoids a collapsed representation and recovers the invariant content block. Intuitively, this is because any function that only depends on $\mathbf{c}$ will be invariant across $(\mathbf{x},\tilde{\mathbf{x}})$, so it is beneficial to preserve all content information to maximise entropy.

**Discussion.** Of our theoretical results, Thm. 4.4 requires the weakest set of assumptions, and is most closely aligned with common SSL practice. As discussed in § 2, contrastive SSL with negative samples using InfoNCE (1) as an objective can asymptotically be understood as alignment with entropy regularisation [124], i.e., objective (5). *Thm. 4.4 thus provides a theoretical justification for the empirically observed effectiveness of CL with InfoNCE*: subject to our assumptions, CL with InfoNCE asymptotically isolates content, i.e., the part of the representation that is always left invariant by augmentation. For example, the strong image classification performance based on representations learned by SimCLR [20], which uses color distortion and random crops as augmentations, can be explained in that object class is a content variable in this case. We extensively evaluate the effect of various augmentation techniques on different ground-truth latent factors in our experiments in § 5. There is also an interesting connection between Thm. 4.4 and BarlowTwins [128], which only uses positive pairs and combines alignment with a redundancy reduction regulariser that enforces decorrelation between the inferred latents. Intuitively, redundancy reduction is related to increased entropy: $\mathbf{g}^\star$ constructed in the proof of Thm. 4.4—and thus also any other minimiser of (5)—attains the global optimum of the BarlowTwins objective, though the reverse implication may not hold.

## 5 Experiments

We perform two main experiments. First, we numerically test our main result, Thm. 4.4, in a *fully-controlled*, finite sample setting (§ 5.1), using CL to estimate the entropy term in (5). Second, we seek to better understand the effect of data augmentations used *in practice* (§ 5.2). To this end, we introduce a new dataset of 3D objects with dependencies between a number of known ground-truth factors, and use it to evaluate the effect of different augmentation techniques on what is identified as content. Additional experiments are summarised in § 5.3 and described in more detail in Appendix C.

### 5.1 Numerical data

**Experimental setup.** We generate synthetic data as described in § 3. We consider $n_c = n_s = 5$, with content and style latents distributed as $\mathbf{c} \sim \mathcal{N}(0, \Sigma_c)$ and $\mathbf{s}|\mathbf{c} \sim \mathcal{N}(\mathbf{a} + B\mathbf{c}, \Sigma_s)$, thus allowing

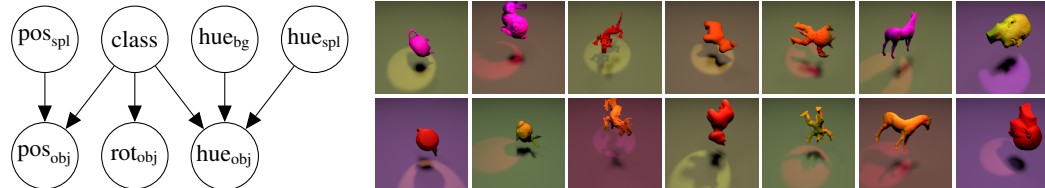

Figure 2: *(Left)* Causal graph for the *Causal3DIdent* dataset. *(Right)* Two samples from each object class.

for *statistical dependence* within the two blocks (via $\Sigma_c$ and $\Sigma_s$) and *causal dependence* between content and style (via $B$). For $\mathbf{f}$, we use a 3-layer MLP with LeakyReLU activation functions.[10] The distribution $p_A$ over subsets of changing style variables is obtained by independently flipping the same biased coin for each $s_i$. The conditional style distribution is taken as $p_{\tilde{\mathbf{s}}_A|\mathbf{s}_A} = \mathcal{N}(\mathbf{s}_A, \Sigma_A)$. We train an encoder $\mathbf{g}$ on pairs $(\mathbf{x}, \tilde{\mathbf{x}})$ with InfoNCE using the negative L2 loss as the similarity measure, i.e., we approximate (5) using empirical averages and negative samples. For evaluation, we use kernel ridge regression [88] to predict the ground truth $\mathbf{c}$ and $\mathbf{s}$ from the learnt representation $\hat{\mathbf{c}} = \mathbf{g}(\mathbf{x})$ and report the $R^2$ coefficient of determination. For a more detailed account, we refer to Appendix D.

**Results.** In the inset table, we report mean ± std. dev. over 3 random seeds across four generative processes of increasing complexity covered by Thm. 4.4: "p(chg.)", "Stat.", and "Cau." denote respectively the change probability for each $s_i$, statistical dependence within blocks ($\Sigma_c \neq I \neq \Sigma_s$), and causal dependence of style on content ($B \neq 0$). An $R^2$ close to

| Generative process | | | $R^2$ (nonlinear) | |
|---|---|---|---|---|
| p(chg.) | Stat. | Cau. | Content c | Style s |
| 1.0 | ✗ | ✗ | **1.00** ± 0.00 | 0.07 ± 0.00 |
| 0.75 | ✗ | ✗ | **1.00** ± 0.00 | 0.06 ± 0.05 |
| 0.75 | ✓ | ✗ | **0.98** ± 0.03 | 0.37 ± 0.05 |
| 0.75 | ✓ | ✓ | **0.99** ± 0.01 | **0.80** ± 0.08 |

one indicates that almost all variation is explained by $\hat{\mathbf{c}}$, i.e., that there is a 1-1 mapping, as required by Defn. 4.1. As can be seen, *across all settings, content is block-identified.* Regarding style, we observe an increased score with the introduction of dependencies, which we explain in an extended discussion in Appendix C.1. Finally, we show in Appendix C.1 that a high $R^2$ score can be obtained even if we use linear regression to predict $\mathbf{c}$ from $\hat{\mathbf{c}}$ ($R^2 = 0.98 \pm 0.01$, for the last row).

### 5.2 High-dimensional images: *Causal3DIdent*

***Causal3DIdent* dataset.** *3DIdent* [129] is a benchmark for evaluating identifiability with rendered $224 \times 224$ images which contains hallmarks of natural environments (e.g. shadows, different lighting conditions, a 3D object). For influence of the latent factors on the renderings, see Fig. 2 of [129]. In *3DIdent*, there is a single object class (Teapot [89]), and all 10 latents are sampled independently. For *Causal3DIdent*, we introduce **six** additional classes: Hare [121], Dragon [110], Cow [62], Armadillo [70], Horse [98], and Head [111]; and impose a causal graph over the latent variables, see Fig. 2. While object class and all environment variables (spotlight position & hue, background hue) are sampled independently, all object latents are dependent,[11] see Appendix B for details.[12]

**Experimental setup.** For $\mathbf{g}$, we train a convolutional encoder composed of a ResNet18 [46] and an additional fully-connected layer, with LeakyReLU activation. As in SimCLR [20], we use InfoNCE with cosine similarity, and train on pairs of augmented examples $(\tilde{\mathbf{x}}, \tilde{\mathbf{x}}')$. As $n_c$ is unknown and variable depending on the augmentation, we fix $\dim(\hat{\mathbf{c}}) = 8$ throughout. Note that we find the results to be, for the most part, robust to the choice of $\dim(\hat{\mathbf{c}})$, see inset figure. We consider the following data augmentations (DA): crop, resize & flip; colour distortion (jitter & drop);

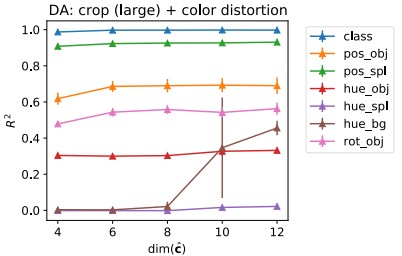

and rotation $\in \{90°, 180°, 270°\}$. For comparison, we also consider directly imposing a content-style

---

[10]chosen to lead to invertibility almost surely by following the settings used by previous work [54, 55]

[11]e.g., our causal graph entails hares blend into the environment (object hue centered about background & spotlight hue), a form of active camouflage observed in Alaskan [78], Arctic [2], & Snowshoe hares.

[12]We made the Causal3DIdent dataset publicly available at this URL.

Table 1: *Causal3DIdent* results: $R^2$ mean $\pm$ std. dev. over 3 random seeds. DA: data augmentation, LT: latent transformation, bold: $R^2 \geq 0.5$, red: $R^2 < 0.25$. Results for individual axes of object position & rotation are aggregated, see Appendix C for the full table.

| Views generated by | Class | Positions | | Hues | | | Rotations |
|---|---|---|---|---|---|---|---|
| | | object | spotlight | object | spotlight | background | |
| DA: colour distortion | $0.42 \pm 0.01$ | $\mathbf{0.61} \pm 0.10$ | $0.17 \pm 0.00$ | $0.10 \pm 0.01$ | $0.01 \pm 0.00$ | $0.01 \pm 0.00$ | $0.33 \pm 0.02$ |
| LT: change hues | $\mathbf{1.00} \pm 0.00$ | $\mathbf{0.59} \pm 0.33$ | $\mathbf{0.91} \pm 0.00$ | $0.30 \pm 0.00$ | $0.00 \pm 0.00$ | $0.00 \pm 0.00$ | $0.30 \pm 0.01$ |
| DA: crop (large) | $0.28 \pm 0.04$ | $0.09 \pm 0.08$ | $0.21 \pm 0.13$ | $\mathbf{0.87} \pm 0.00$ | $0.09 \pm 0.02$ | $\mathbf{1.00} \pm 0.00$ | $0.02 \pm 0.02$ |
| DA: crop (small) | $0.14 \pm 0.00$ | $0.00 \pm 0.01$ | $0.00 \pm 0.01$ | $0.00 \pm 0.00$ | $0.00 \pm 0.00$ | $\mathbf{1.00} \pm 0.00$ | $0.00 \pm 0.00$ |
| LT: change positions | $\mathbf{1.00} \pm 0.00$ | $0.16 \pm 0.23$ | $0.00 \pm 0.01$ | $0.46 \pm 0.02$ | $0.00 \pm 0.00$ | $\mathbf{0.97} \pm 0.00$ | $0.29 \pm 0.01$ |
| DA: crop (large) + colour distortion | $\mathbf{0.97} \pm 0.00$ | $\mathbf{0.59} \pm 0.07$ | $\mathbf{0.59} \pm 0.05$ | $0.28 \pm 0.00$ | $0.01 \pm 0.01$ | $0.01 \pm 0.00$ | $\mathbf{0.74} \pm 0.03$ |
| DA: crop (small) + colour distortion | $\mathbf{1.00} \pm 0.00$ | $\mathbf{0.69} \pm 0.04$ | $\mathbf{0.93} \pm 0.00$ | $0.30 \pm 0.01$ | $0.00 \pm 0.00$ | $0.02 \pm 0.03$ | $\mathbf{0.56} \pm 0.03$ |
| LT: change positions + hues | $\mathbf{1.00} \pm 0.00$ | $0.22 \pm 0.22$ | $0.07 \pm 0.08$ | $0.32 \pm 0.02$ | $0.00 \pm 0.01$ | $0.02 \pm 0.03$ | $0.34 \pm 0.06$ |
| DA: rotation | $0.33 \pm 0.06$ | $0.17 \pm 0.09$ | $0.23 \pm 0.12$ | $\mathbf{0.83} \pm 0.01$ | $0.30 \pm 0.12$ | $\mathbf{0.99} \pm 0.00$ | $0.05 \pm 0.03$ |
| LT: change rotations | $\mathbf{1.00} \pm 0.00$ | $\mathbf{0.53} \pm 0.33$ | $\mathbf{0.90} \pm 0.00$ | $0.41 \pm 0.00$ | $0.00 \pm 0.00$ | $\mathbf{0.97} \pm 0.00$ | $0.28 \pm 0.00$ |
| DA: rotation + colour distortion | $\mathbf{0.59} \pm 0.01$ | $\mathbf{0.58} \pm 0.06$ | $0.21 \pm 0.01$ | $0.12 \pm 0.02$ | $0.01 \pm 0.00$ | $0.01 \pm 0.00$ | $0.33 \pm 0.04$ |
| LT: change rotations + hues | $\mathbf{1.00} \pm 0.00$ | $\mathbf{0.57} \pm 0.34$ | $\mathbf{0.91} \pm 0.00$ | $0.30 \pm 0.00$ | $0.00 \pm 0.00$ | $0.00 \pm 0.00$ | $0.28 \pm 0.00$ |

partition by performing a latent transformation (LT) to generate views. For evaluation, we use linear logistic regression to predict object class, and kernel ridge to predict the other latents from $\hat{\mathbf{c}}$.[13]

**Results.** The results are presented in Tab. 1. Overall, our main findings can be summarised as:

(i) it can be difficult to design image-level augmentations that leave *specific* latent factors invariant;

(ii) augmentations & latent transformations generally have a similar effect on groups of latents;

(iii) augmentations that yield good classification performance induce variation in all other latents.

We observe that, similar to directly varying the hue latents, colour distortion leads to a discarding of hue information as style, and a preservation of (object) position as content. Crops, similar to varying the position latents, lead to a discarding of position as style, and a preservation of background and object hue as content, the latter assuming crops are sufficiently large. In contrast, image-level rotation affects both the object rotation and position, and thus deviates from only varying the rotation latents.

Whereas class is always preserved as content when generating views with latent transformations, when using data augmentations, we can only reliably decode class when crops & colour distortion are used in conjunction—a result which mirrors evaluation on ImageNet [20]. As can be seen by our evaluation of crops & colour distortion in isolation, while colour distortion leads to a discarding of hues as style, crops lead to a discarding of position & rotation as style. Thus, when used in conjunction, class is isolated as the sole content variable. See Appendix C.2 for additional analysis.

### 5.3 Additional experiments and ablations

We also perform an ablation on $\dim(\hat{\mathbf{c}})$ for the synthetic setting from § 5.1, see Appendix C.1 for details. Generally, we find that if $\dim(\hat{\mathbf{c}}) < n_c$, there is insufficient capacity to encode all content, so a lower-dimensional mixture of content is learnt. Conversely, if $\dim(\hat{\mathbf{c}}) > n_c$, the excess capacity is used to encode some style information (as that increases entropy). Further, we repeat our analysis from § 5.2 using `BarlowTwins` [128] (instead of `SimCLR`) which, as discussed at the end of § 4.2, is also loosely related to Thm. 4.4. The results mostly mirror those obtained for `SimCLR` and presented in Tab. 1, see Appendix C.2 for details. Finally, we ran the same experimental setup as in § 5.2 also on the *MPI3D-real* dataset [38] containing $> 1$ million *real* images with ground-truth annotations of 3D objects being moved by a robotic arm. Subject to some caveats, the results show a similar trend as those on *Causal3DIdent*, see Appendix C.3 for details.

## 6 Discussion

**Theory vs practice.** We have made an effort to tailor our problem formulation (§ 3) to the setting of data augmentation with content-preserving transformations. However, some of our more technical assumptions, which are necessary to prove block-identifiability of the invariant content partition, may not hold exactly in practice. This is apparent, e.g., from our second experiment (§ 5.2), where we observe that—while class should, in principle, always be invariant across views (i.e., content)—when

---

[13]See Appendix C.2 for results with linear regression, as well as evaluation using a higher-dimensional intermediate layer by considering a projection head [20].

using *only* crops, colour distortion, or rotation, $\mathbf{g}$ appears to encode *shortcuts* [37, 96].[14] Data augmentation, unlike latent transformations, generates views $\tilde{\mathbf{x}}$ which are not restricted to the 11-dim. image manifold $\mathcal{X}$ corresponding to the generative process of *Causal3DIdent*, but may introduce additional variation: e.g., colour distortion leads to a rich combination of colours, typically a 3-dim. feature, whereas *Causal3DIdent* only contains one degree of freedom (hue). With additional factors, any introduced invariances may be encoded as content in place of class. Image-level augmentations also tend to change multiple latent factors in a correlated way, which may violate assumption *(iii)* of our theorems, i.e., that $p_{\tilde{\mathbf{s}}_A|\mathbf{s}_A}$ is fully-supported locally. We also assume that $\mathbf{z}$ is continuous, even though *Causal3DIdent* and most disentanglement datasets also contain discrete latents. This is a very common assumption in the related literature [39, 54, 57, 58, 63, 69, 82, 83, 129] that may be relaxed in future work. Moreover, our theory holds asymptotically and at the global optimum, whereas in practice we solve a non-convex optimisation problem with a finite sample and need to approximate the entropy term in (5), e.g., using a finite number of negative pairs. The resulting challenges for optimisation may be further accentuated by the higher dimensionality of $\mathcal{X}$ induced by image-level augmentations. Finally, we remark that while, for simplicity, we have presented our theory for pairs $(\mathbf{x}, \tilde{\mathbf{x}})$ of original and augmented examples, in practice, using pairs $(\tilde{\mathbf{x}}, \tilde{\mathbf{x}}')$ of two augmented views typically yields better performance. All of our assumptions (content invariance, changing style, etc) and theoretical results still apply to the latter case. We believe that using two augmented views helps because it leads to *increased variability* across the pair: for if $\tilde{\mathbf{x}}$ and $\tilde{\mathbf{x}}'$ differ from $\mathbf{x}$ in style subsets $A$ and $A'$, respectively, then $(\tilde{\mathbf{x}}, \tilde{\mathbf{x}}')$ differ from each other (a.s.) in the union $A \cup A'$.

**Beyond entropy regularisation.** We have shown a clear link between an identifiable maximum entropy approach to SSL (Thm. 4.4) and SimCLR [20] based on the analysis of [124], and have discussed an intuitive connection to the notion of redundancy reduction used in BarlowTwins [128]. Whether other types of regularisation such as the architectural approach pursued in BYOL [41] and SimSiam [21] can also be linked to entropy maximisation, remains an open question. Deriving similar results to Thm. 4.4 with other regularisers is a promising direction for future research, c.f. [116].

**The choice of augmentation technique implicitly defines content and style.** As we have defined content as the part of the representation which is always left invariant across views, the choice of augmentation implicitly determines the content-style partition. This is particularly important to keep in mind when applying SSL with data augmentation to safety-critical domains, such as medical imaging. We also advise caution when using data augmentation to identify specific latent properties, since, as observed in § 5.2, image-level transformations may affect the underlying ground-truth factors in unanticipated ways. Also note that, *for a given downstream task*, we may not want to discard all style information since style variables may still be correlated with the task of interest and may thus help improve predictive performance. *For arbitrary downstream tasks*, however, where style may change in an adversarial way, it can be shown that only using content is optimal [103].

**What vs how information is encoded.** We focus on *what* information is learnt by SSL with data augmentations by specifying a generative process and studying identifiability of the latent representation. Orthogonal to this, a different line of work instead studies *how* information is encoded by analysing the sample complexity needed to solve a *given downstream task* using a *linear* predictor [3, 74, 116–119]. Provided that downstream tasks only involve content, we can draw some comparisons. Whereas our results recover content only up to arbitrary invertible nonlinear functions (see Defn. 4.1), our problem setting is more general: [3, 74] assume (approximate) independence of views $(\mathbf{x}, \tilde{\mathbf{x}})$ given the task (content), while [118, 119] assume (approximate) independence between one view and the task (content) given the other view, neither of which hold in our setting.

**Conclusion.** Existing representation learning approaches typically assume mutually independent latents, though dependencies clearly exist in nature [106]. We demonstrate that in a *non-i.i.d.* scenario, e.g., by constructing multiple views of the same example with data augmentation, we can learn useful representations in the presence of this neglected phenomenon. More specifically, the present work contributes, to the best of our knowledge, the first: (i) identifiability result under *arbitrary dependence* between latents; and (ii) empirical study that evaluates the effect of data augmentations not only on classification, but also on other *continuous* ground-truth latents. Unlike existing identifiability results which rely on *change* as a learning signal, our approach aims to identify what is always shared across views, i.e., also using *invariance* as a learning signal. We hope that this change in perspective will be helpful for applications such as optimal style transfer or disentangling shape from pose in vision, and inspire other types of *counterfactual training* to recover a more fine-grained causal representation.

---

[14]class is distinguished by shape, a feature commonly unused in downstream tasks on natural images [36]

## Acknowledgements

We thank: the anonymous reviewers for several helpful suggestions that triggered improvements in theory and additional experiments; Cian Eastwood, Ilyes Khemakem, Michael Lohaus, Osama Makansi, Ricardo Pio Monti, Roland Zimmermann, Weiyang Liu, and the MPI Tübingen causality group for helpful discussions and comments; Hugo Yèche for pointing out a mistake in §2 of an earlier version of the manuscript; github user `TangTangFei` for catching a bug in the implementation of the experiments from § 5.1 (that has been corrected in this version); and the International Max Planck Research School for Intelligent Systems (IMPRS-IS) for supporting YS.

## Funding Transparency Statement

WB acknowledges support via his Emmy Noether Research Group funded by the German Science Foundation (DFG) under grant no. BR 6382/1-1 as well as support by Open Philantropy and the Good Ventures Foundation. This work was supported by the German Federal Ministry of Education and Research (BMBF): Tübingen AI Center, FKZ: 01IS18039A, 01IS18039B; and by the Machine Learning Cluster of Excellence, EXC number 2064/1 – Project number 390727645.

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
