# APPENDIX

## Overview:

- Appendix A contains the full proofs for all theoretical results from the main paper.
- Appendix B contains additional details and plots for the *Causal3DIdent* dataset.
- Appendix C contains additional experimental results and analysis.
- Appendix D contains additional implementation details for our experiments.

## A  Proofs

We now present the full detailed proofs of our three theorems which were briefly sketched in the main paper. We remark that these proofs build on each other, in the sense that the (main) step 2 of the proof of Thm. 4.2 is also used in the proofs of Thms. 4.3 and 4.4.

### A.1  Proof of Thm. 4.2

**Theorem 4.2** (Identifying content with a generative model). *Consider the data generating process described in § 3, i.e., the pairs $(\mathbf{x}, \tilde{\mathbf{x}})$ of original and augmented views are generated according to (2) and (3) with $p_{\tilde{\mathbf{z}}|\mathbf{z}}$ as defined in Assumptions 3.1 and 3.2. Assume further that*

*(i)* $\mathbf{f} : \mathcal{Z} \rightarrow \mathcal{X}$ *is smooth and invertible with smooth inverse (i.e., a diffeomorphism);*

*(ii)* $p_{\mathbf{z}}$ *is a smooth, continuous density on $\mathcal{Z}$ with $p_{\mathbf{z}}(\mathbf{z}) > 0$ almost everywhere;*

*(iii) for any $l \in \{1, ..., n_s\}$, $\exists A \subseteq \{1, ..., n_s\}$ s.t. $l \in A$; $p_A(A) > 0$; $p_{\tilde{\mathbf{s}}_A|\mathbf{s}_A}$ is smooth w.r.t. both $\mathbf{s}_A$ and $\tilde{\mathbf{s}}_A$; and for any $\mathbf{s}_A$, $p_{\tilde{\mathbf{s}}_A|\mathbf{s}_A}(\cdot|\mathbf{s}_A) > 0$ in some open, non-empty subset containing $\mathbf{s}_A$.*

*If, for a given $n_s$ ($1 \leq n_s < n$), a generative model $(\hat{p}_{\mathbf{z}}, \hat{p}_A, \hat{p}_{\tilde{\mathbf{s}}|\mathbf{s},A}, \hat{\mathbf{f}})$ assumes the same generative process (§ 3), satisfies the above assumptions (i)-(iii), and matches the data likelihood,*

$$p_{\mathbf{x},\tilde{\mathbf{x}}}(\mathbf{x}, \tilde{\mathbf{x}}) = \hat{p}_{\mathbf{x},\tilde{\mathbf{x}}}(\mathbf{x}, \tilde{\mathbf{x}}) \qquad \forall (\mathbf{x}, \tilde{\mathbf{x}}) \in \mathcal{X} \times \mathcal{X},$$

*then it block-identifies the true content variables via $\mathbf{g} = \hat{\mathbf{f}}^{-1}$ in the sense of Defn. 4.1.*

*Proof.* The proof consists of two main steps.

In the first step, we use assumption *(i)* and the matching likelihoods to show that the representation $\hat{\mathbf{z}} = \mathbf{g}(\mathbf{x})$ extracted by $\mathbf{g} = \hat{\mathbf{f}}^{-1}$ is related to the true latent $\mathbf{z}$ by a smooth invertible mapping $\mathbf{h}$, and that $\hat{\mathbf{z}}$ must satisfy invariance across $(\mathbf{x}, \tilde{\mathbf{x}})$ in the first $n_c$ (content) components almost surely (a.s.) with respect to (w.r.t.) the true generative process.

In the second step, we then use assumptions *(ii)* and *(iii)* to prove (by contradiction) that $\hat{\mathbf{c}} := \hat{\mathbf{z}}_{1:n_c} = \mathbf{h}(\mathbf{z})_{1:n_c}$ can, in fact, only depend on the true content $\mathbf{c}$ and not on the true style $\mathbf{s}$, for otherwise the invariance established in the first step would have be violated with probability greater than zero.

To provide some further intuition for the second step, the assumed generative process implies that $(\mathbf{c}, \mathbf{s}, \tilde{\mathbf{s}})|A$ is constrained to take values (a.s.) in a subspace $\mathcal{R}$ of $\mathcal{C} \times \mathcal{S} \times \mathcal{S}$ of dimension $n_c + n_s + |A|$ (as opposed to dimension $n_c + 2n_s$ for $\mathcal{C} \times \mathcal{S} \times \mathcal{S}$). In this context, assumption *(iii)* implies that $(\mathbf{c}, \mathbf{s}, \tilde{\mathbf{s}})|A$ has a density with respect to a measure on this subspace equivalent to the Lebesgue measure on $\mathbb{R}^{n_c + n_s + |A|}$. This equivalence implies, in particular, that this "subspace measure" is strictly positive: it takes strictly positive values on open sets of $\mathcal{R}$ seen as a topological subspace of $\mathcal{C} \times \mathcal{S} \times \mathcal{S}$. These open sets are defined by the induced topology: they are the intersection of the open sets of $\mathcal{C} \times \mathcal{S} \times \mathcal{S}$ with $\mathcal{R}$. An open set $B$ of $V$ on which $p(\mathbf{c}, \mathbf{s}, \tilde{\mathbf{s}}|A) > 0$ then satisfies $P(B|A) > 0$. We look for such an open set to prove our result.

**Step 1.**   From the assumed data generating process described in § 3—in particular, from the form of the model conditional $\hat{p}_{\tilde{\mathbf{z}}|\mathbf{z}}$ described in Assumptions 3.1 and 3.2—it follows that

$$\mathbf{g}(\mathbf{x})_{1:n_c} = \mathbf{g}(\tilde{\mathbf{x}})_{1:n_c} \tag{6}$$

a.s., i.e., with probability one, w.r.t. the model distribution $\hat{p}_{\mathbf{x},\tilde{\mathbf{x}}}$.

Due to the assumption of matching likelihoods, the invariance in (6) must also hold (a.s.) w.r.t. the true data distribution $p_{\mathbf{x}, \tilde{\mathbf{x}}}$.

Next, since $\mathbf{f}, \hat{\mathbf{f}} : \mathcal{Z} \to \mathcal{X}$ are smooth and invertible functions by assumption *(i)*, there exists a smooth and invertible function $\mathbf{h} = \mathbf{g} \circ \mathbf{f} : \mathcal{Z} \to \mathcal{Z}$ such that

$$\mathbf{g} = \mathbf{h} \circ \mathbf{f}^{-1}. \tag{7}$$

Substituting (7) into (6), we obtain (a.s. w.r.t. $p$):

$$\hat{\mathbf{c}} := \hat{\mathbf{z}}_{1:n_c} = \mathbf{g}(\mathbf{x})_{1:n_c} = \mathbf{h}(\mathbf{f}^{-1}(\mathbf{x}))_{1:n_c} = \mathbf{h}(\mathbf{f}^{-1}(\tilde{\mathbf{x}}))_{1:n_c} \tag{8}$$

Substituting $\mathbf{z} = \mathbf{f}^{-1}(\mathbf{x})$ and $\tilde{\mathbf{z}} = \mathbf{f}^{-1}(\tilde{\mathbf{x}})$ into (8), we obtain (a.s. w.r.t. $p$)

$$\hat{\mathbf{c}} = \mathbf{h}(\mathbf{z})_{1:n_c} = \mathbf{h}(\tilde{\mathbf{z}})_{1:n_c}. \tag{9}$$

It remains to show that $\mathbf{h}(\cdot)_{1:n_c}$ can only be a function of $\mathbf{c}$, i.e., does not depend on any other (style) dimension of $\mathbf{z} = (\mathbf{c}, \mathbf{s})$.

**Step 2.** Suppose *for a contradiction* that $\mathbf{h}_c(\mathbf{c}, \mathbf{s}) := \mathbf{h}(\mathbf{c}, \mathbf{s})_{1:n_c} = \mathbf{h}(\mathbf{z})_{1:n_c}$ depends on some component of the style variable $\mathbf{s}$:

$$\exists l \in \{1, ..., n_s\}, (\mathbf{c}^*, \mathbf{s}^*) \in \mathcal{C} \times \mathcal{S}, \qquad \text{s.t.} \qquad \frac{\partial \mathbf{h}_c}{\partial s_l}(\mathbf{c}^*, \mathbf{s}^*) \neq 0, \tag{10}$$

that is, we assume that the partial derivative of $\mathbf{h}_c$ w.r.t. some style variable $s_l$ is non-zero at some point $\mathbf{z}^* = (\mathbf{c}^*, \mathbf{s}^*) \in \mathcal{Z} = \mathcal{C} \times \mathcal{S}$.

Since $\mathbf{h}$ is smooth, so is $\mathbf{h}_c$. Therefore, $\mathbf{h}_c$ has continuous (first) partial derivatives.

By continuity of the partial derivative, $\frac{\partial \mathbf{h}_c}{\partial s_l}$ must be non-zero in a neighbourhood of $(\mathbf{c}^*, \mathbf{s}^*)$, i.e.,

$$\exists \eta > 0 \qquad \text{s.t.} \qquad s_l \mapsto \mathbf{h}_c\big(\mathbf{c}^*, (\mathbf{s}_{-l}^*, s_l)\big) \quad \text{is strictly monotonic on} \quad (s_l^* - \eta, s_l^* + \eta), \tag{11}$$

where $\mathbf{s}_{-l} \in \mathcal{S}_{-l}$ denotes the vector of remaining style variables except $s_l$.

Next, define the auxiliary function $\psi : \mathcal{C} \times \mathcal{S} \times \mathcal{S} \to \mathbb{R}_{\geq 0}$ as follows:

$$\psi(\mathbf{c}, \mathbf{s}, \tilde{\mathbf{s}}) := |\mathbf{h}_c(\mathbf{c}, \mathbf{s}) - \mathbf{h}_c(\mathbf{c}, \tilde{\mathbf{s}})| \geq 0. \tag{12}$$

To obtain a contradiction to the invariance condition (9) from Step 1 under assumption (10), it remains to show that $\psi$ from (12) is *strictly positive* with probability greater than zero (w.r.t. $p$).

First, the strict monotonicity from (11) implies that

$$\psi\big(\mathbf{c}^*, (\mathbf{s}_{-l}^*, s_l), (\mathbf{s}_{-l}^*, \tilde{s}_l)\big) > 0, \quad \forall (s_l, \tilde{s}_l) \in (s_l^*, s_l^* + \eta) \times (s_l^* - \eta, s_l^*). \tag{13}$$

Note that in order to obtain the strict inequality in (13), it is important that $s_l$ and $\tilde{s}_l$ take values in *disjoint* open subsets of the interval $(s_l^* - \eta, s_l^* + \eta)$ from (11).

Since $\psi$ is a composition of continuous functions (absolute value of the difference of two continuous functions), $\psi$ is continuous.

Consider the open set $\mathbb{R}_{>0}$, and recall that, under a continuous function, pre-images (or inverse images) of open sets are always *open*.

Applied to the continuous function $\psi$, this pre-image corresponds to an *open* set

$$\mathcal{U} \subseteq \mathcal{C} \times \mathcal{S} \times \mathcal{S} \tag{14}$$

in the domain of $\psi$ on which $\psi$ is strictly positive.

Moreover, due to (13):

$$\{\mathbf{c}^*\} \times \big(\{\mathbf{s}_{-l}^*\} \times (s_l^*, s_l^* + \eta)\big) \times \big(\{\mathbf{s}_{-l}^*\} \times (s_l^* - \eta, s_l^*)\big) \subset \mathcal{U}, \tag{15}$$

so $\mathcal{U}$ is *non-empty*.

Next, by assumption *(iii)*, there exists at least one subset $A \subseteq \{1, ..., n_s\}$ of changing style variables such that $l \in A$ and $p_A(A) > 0$; pick one such subset and call it $A$.

Then, also by assumption *(iii)*, for any $\mathbf{s}_A \in \mathcal{S}_A$, there is an open subset $\mathcal{O}(\mathbf{s}_A) \subseteq \mathcal{S}_A$ containing $\mathbf{s}_A$, such that $p_{\tilde{\mathbf{s}}_A|\mathbf{s}_A}(\cdot|\mathbf{s}_A) > 0$ within $\mathcal{O}(\mathbf{s}_A)$.

Define the following space

$$\mathcal{R}_A := \{(\mathbf{s}_A, \tilde{\mathbf{s}}_A) : \mathbf{s}_A \in \mathcal{S}_A, \tilde{\mathbf{s}}_A \in \mathcal{O}(\mathbf{s}_A)\} \tag{16}$$

and, recalling that $A^{\mathrm{c}} = \{1, ..., n_s\} \setminus A$ denotes the complement of $A$, define

$$\mathcal{R} := \mathcal{C} \times \mathcal{S}_{A^{\mathrm{c}}} \times \mathcal{R}_A \tag{17}$$

which is a topological subspace of $\mathcal{C} \times \mathcal{S} \times \mathcal{S}$.

By assumptions *(ii)* and *(iii)*, $p_{\mathbf{z}}$ is smooth and fully supported, and $p_{\tilde{\mathbf{s}}_A|\mathbf{s}_A}(\cdot|\mathbf{s}_A)$ is smooth and fully supported on $\mathcal{O}(\mathbf{s}_A)$ for any $\mathbf{s}_A \in \mathcal{S}_A$. Therefore, the measure $\mu_{(\mathbf{c}, \mathbf{s}_{A^{\mathrm{c}}}, \mathbf{s}_A, \tilde{\mathbf{s}}_A)|A}$ has fully supported, strictly-positive density on $\mathcal{R}$ w.r.t. a strictly positive measure on $\mathcal{R}$. In other words, $p_{\mathbf{z}} \times p_{\tilde{\mathbf{s}}_A|\mathbf{s}_A}$ is fully supported (i.e., strictly positive) on $\mathcal{R}$.

Now consider the intersection $\mathcal{U} \cap \mathcal{R}$ of the open set $\mathcal{U}$ with the topological subspace $\mathcal{R}$.

Since $\mathcal{U}$ is open, by the definition of topological subspaces, the intersection $\mathcal{U} \cap \mathcal{R} \subseteq \mathcal{R}$ is *open* in $\mathcal{R}$, (and thus has the same dimension as $\mathcal{R}$ if non-empty).

Moreover, since $\mathcal{O}(\mathbf{s}_A^*)$ is open containing $\mathbf{s}_A^*$, there exists $\eta' > 0$ such that $\{\mathbf{s}_{-l}^*\} \times (s_l^* - \eta', s_l^*) \subset \mathcal{O}(\mathbf{s}_A^*)$. Thus, for $\eta'' = \min(\eta, \eta') > 0$,

$$\{\mathbf{c}^*\} \times \{\mathbf{s}_{A^{\mathrm{c}}}^*\} \times \left(\{\mathbf{s}_{A\setminus\{l\}}^*\} \times (s_l^*, s_l^* + \eta)\right) \times \left(\{\mathbf{s}_{A\setminus\{l\}}^*\} \times (s_l^* - \eta'', s_l^*)\right) \subset \mathcal{R}. \tag{18}$$

In particular, this implies that

$$\{\mathbf{c}^*\} \times \left(\{\mathbf{s}_{-l}^*\} \times (s_l^*, s_l^* + \eta)\right) \times \left(\{\mathbf{s}_{-l}^*\} \times (s_l^* - \eta'', s_l^*)\right) \subset \mathcal{R}, \tag{19}$$

Now, since $\eta'' \leq \eta$, the LHS of (19) is also in $\mathcal{U}$ according to (15), so the intersection $\mathcal{U} \cap \mathcal{R}$ is *non-empty*.

In summary, the intersection $\mathcal{U} \cap \mathcal{R} \subseteq \mathcal{R}$:

- is non-empty (since both $\mathcal{U}$ and $\mathcal{R}$ contain the LHS of (15));
- is an open subset of the topological subspace $\mathcal{R}$ of $\mathcal{C} \times \mathcal{S} \times \mathcal{S}$ (since it is the intersection of an open set, $\mathcal{U}$, with $\mathcal{R}$);
- satisfies $\psi > 0$ (since this holds for all of $\mathcal{U}$);
- is fully supported w.r.t. the generative process (since this holds for all of $\mathcal{R}$).

As a consequence,

$$\mathbb{P}\left(\psi(\mathbf{c}, \mathbf{s}, \tilde{\mathbf{s}}) > 0|A\right) \geq \mathbb{P}(\mathcal{U} \cap \mathcal{R}) > 0, \tag{20}$$

where $\mathbb{P}$ denotes probability w.r.t. the true generative process $p$.

Since $p_A(A) > 0$, this is a **contradiction** to the invariance (9) from Step 1.

Hence, assumption (10) cannot hold, i.e., $\mathbf{h}_c(\mathbf{c}, \mathbf{s})$ does not depend on any style variable $s_l$. It is thus only a function of $\mathbf{c}$, i.e., $\hat{\mathbf{c}} = \mathbf{h}_c(\mathbf{c})$.

Finally, smoothness and invertibility of $\mathbf{h}_c : \mathcal{C} \to \mathcal{C}$ follow from smoothness and invertibility of $\mathbf{h}$, as established in Step 1.

This concludes the proof that $\hat{\mathbf{c}}$ is related to the true content $\mathbf{c}$ via a smooth invertible mapping. $\qquad\square$

## A.2  Proof of Thm. 4.3

**Theorem 4.3** (Identifying content with an invertible encoder). *Assume the same data generating process (§ 3) and conditions (i)-(iv) as in Thm. 4.2. Let $\mathbf{g} : \mathcal{X} \to \mathcal{Z}$ be any smooth and invertible function which minimises the following functional:*

$$\mathcal{L}_{\mathrm{Align}}(\mathbf{g}) := \mathbb{E}_{(\mathbf{x}, \tilde{\mathbf{x}}) \sim p_{\mathbf{x}, \tilde{\mathbf{x}}}} \left[\left\|\left\|\mathbf{g}(\mathbf{x})_{1:n_c} - \mathbf{g}(\tilde{\mathbf{x}})_{1:n_c}\right\|\right\|_2^2\right] \tag{4}$$

*Then $\mathbf{g}$ block-identifies the true content variables in the sense of Definition 4.1.*

*Proof.* As in the proof of Thm. 4.2, the proof again consists of two main steps.

In the first step, we show that the representation $\hat{\mathbf{z}} = \mathbf{g}(\mathbf{x})$ extracted by any $\mathbf{g}$ that minimises $\mathcal{L}_{\mathrm{Align}}$ is related to the true latent $\mathbf{z}$ through a smooth invertible mapping $\mathbf{h}$, and that $\hat{\mathbf{z}}$ must satisfy invariance across $(\mathbf{x}, \tilde{\mathbf{x}})$ in the first $n_c$ (content) components almost surely (a.s.) with respect to (w.r.t.) the true generative process.

In the second step, we use the same argument by contradiction as in Step 2 of the proof of Thm. 4.2, to show that $\hat{\mathbf{c}} = \mathbf{h}(\mathbf{z})_{1:n_c}$ can only depend on the true content $\mathbf{c}$ and not on style $\mathbf{s}$.

**Step 1.** From the form of the objective (4), it is clear that $\mathcal{L}_{\mathrm{Align}} \geq 0$ with equality if and only if $\mathbf{g}(\tilde{\mathbf{x}})_{1:n_c} = \mathbf{g}(\mathbf{x})_{1:n_c}$ for all $(\mathbf{x}, \tilde{\mathbf{x}})$ s.t. $p_{\mathbf{x}, \tilde{\mathbf{x}}}(\mathbf{x}, \tilde{\mathbf{x}}) > 0$.

Moreover, it follows from the assumed generative process that the global minimum of zero is attained by the true unmixing $\mathbf{f}^{-1}$ since

$$\mathbf{f}^{-1}(\mathbf{x})_{1:n_c} = \mathbf{c} = \tilde{\mathbf{c}} = \mathbf{f}^{-1}(\tilde{\mathbf{x}})_{1:n_c} \tag{21}$$

holds a.s. (i.e., with probability one) w.r.t. the true generative process $p$.

Hence, there exists at least one smooth invertible function $(\mathbf{f}^{-1})$ which attains the global minimum.

Let $\mathbf{g}$ be *any* function attaining the global minimum of $\mathcal{L}_{\mathrm{Align}}$ of zero.

As argued above, this implies that (a.s. w.r.t. $p$):

$$\mathbf{g}(\tilde{\mathbf{x}})_{1:n_c} = \mathbf{g}(\mathbf{x})_{1:n_c}. \tag{22}$$

Writing $\mathbf{g} = \mathbf{h} \circ \mathbf{f}^{-1}$, where $\mathbf{h}$ is the smooth, invertible function $\mathbf{h} = \mathbf{g} \circ \mathbf{f}$ we obtain (a.s. w.r.t. $p$):

$$\hat{\mathbf{c}} = \mathbf{h}(\tilde{\mathbf{z}})_{1:n_c} = \mathbf{h}(\mathbf{z})_{1:n_c}. \tag{23}$$

Note that this is the same invariance condition as (9) derived in Step 1 of the proof of Thm. 4.2.

**Step 2.** It remains to show that $\mathbf{h}(\mathbf{z})_{1:n_c}$ can only depend on the true content $\mathbf{c}$ and not on any of the style variables $\mathbf{s}$. To show this, we use the same Step 2 as in the proof of Thm. 4.2. $\square$

### A.3 Proof of Thm. 4.4

**Theorem 4.4** (Identifying content with discriminative learning and a non-invertible encoder). *Assume the same data generating process (§ 3) and conditions (i)-(iv) as in Thm. 4.2. Let $\mathbf{g} : \mathcal{X} \to (0, 1)^{n_c}$ be any smooth function which minimises the following functional:*

$$\mathcal{L}_{\mathrm{AlignMaxEnt}}(\mathbf{g}) := \mathbb{E}_{(\mathbf{x}, \tilde{\mathbf{x}}) \sim p_{\mathbf{x}, \tilde{\mathbf{x}}}} \left[ \left\| \mathbf{g}(\mathbf{x}) - \mathbf{g}(\tilde{\mathbf{x}}) \right\|_2^2 \right] - H\left( \mathbf{g}(\mathbf{x}) \right) \tag{5}$$

*where $H(\cdot)$ denotes the differential entropy of the random variable $\mathbf{g}(\mathbf{x})$ taking values in $(0, 1)^{n_c}$. Then $\mathbf{g}$ block-identifies the true content variables in the sense of Defn. 4.1.*

*Proof.* The proof consists of three main steps.

In the first step, we show that the representation $\hat{\mathbf{c}} = \mathbf{g}(\mathbf{x})$ extracted by any smooth function $\mathbf{g}$ that minimises (5) is related to the true latent $\mathbf{z}$ through a smooth mapping $\mathbf{h}$; that $\hat{\mathbf{c}}$ must satisfy invariance across $(\mathbf{x}, \tilde{\mathbf{x}})$ almost surely (a.s.) with respect to (w.r.t.) the true generative process $p$; and that $\hat{\mathbf{c}}$ must follow a uniform distribution on $(0, 1)^{n_c}$.

In the second step, we use the same argument by contradiction as in Step 2 of the proof of Thm. 4.2, to show that $\hat{\mathbf{c}} = \mathbf{h}(\mathbf{z})$ can only depend on the true content $\mathbf{c}$ and not on style $\mathbf{s}$.

Finally, in the third step, we show that $\mathbf{h}$ must be a bijection, i.e., invertible, using a result from [129].

**Step 1.** The global minimum of $\mathcal{L}_{\mathrm{AlignMaxEnt}}$ is reached when the first term (alignment) is minimised (i.e., equal to zero) and the second term (entropy) is maximised.

Without additional moment constraints, the *unique* maximum entropy distribution on $(0, 1)^{n_c}$ is the uniform distribution [25, 61].

First, we show that there exists a smooth function $\mathbf{g}^* : \mathcal{X} \to (0,1)^{n_c}$ which attains the global minimum of $\mathcal{L}_{\text{AlignMaxEnt}}$.

To see this, consider the function $\mathbf{f}^{-1}_{1:n_c} : \mathcal{X} \to \mathcal{C}$, i.e., the inverse of the true mixing $\mathbf{f}$, restricted to its first $n_c$ dimensions. This exists and is smooth since $\mathbf{f}$ is smooth and invertible by assumption *(i)*. Further, we have $\mathbf{f}^{-1}(\mathbf{x})_{1:n_c} = \mathbf{c}$ by definition.

We now build a function $\mathbf{d} : \mathcal{C} \to (0,1)^{n_c}$ which maps $\mathbf{c}$ to a uniform random variable on $(0,1)^{n_c}$ using a recursive construction known as the *Darmois construction* [29, 57].

Specifically, we define

$$d_i(\mathbf{c}) := F_i(c_i|\mathbf{c}_{1:i-1}) = \mathbb{P}(C_i \le c_i|\mathbf{c}_{1:i-1}), \qquad i = 1, ..., n_c, \tag{24}$$

where $F_i$ denotes the conditional cumulative distribution function (CDF) of $c_i$ given $\mathbf{c}_{1:i-1}$.

By construction, $\mathbf{d}(\mathbf{c})$ is uniformly distributed on $(0,1)^{n_c}$ [29, 57].

Further, $\mathbf{d}$ is smooth by the assumption that $p_{\mathbf{z}}$ (and thus $p_{\mathbf{c}}$) is a smooth density.

Finally, we define

$$\mathbf{g}^* := \mathbf{d} \circ \mathbf{f}^{-1}_{1:n_c} : \mathcal{X} \to (0,1)^{n_c}, \tag{25}$$

which is a smooth function since it is a composition of two smooth functions.

**Claim A.1.** $\mathbf{g}^*$ as defined in (25) attains the global minimum of $\mathcal{L}_{\text{AlignMaxEnt}}$.

**Proof of Claim A.1.** Using $\mathbf{f}^{-1}(\mathbf{x})_{1:n_c} = \mathbf{c}$ and $\mathbf{f}^{-1}(\tilde{\mathbf{x}})_{1:n_c} = \tilde{\mathbf{c}}$, we have

$$\mathcal{L}_{\text{AlignMaxEnt}}(\mathbf{g}^*) = \mathbb{E}_{(\mathbf{x},\tilde{\mathbf{x}}) \sim p_{(\mathbf{x},\tilde{\mathbf{x}})}} \left[ \left\| \mathbf{g}^*(\mathbf{x}) - \mathbf{g}^*(\tilde{\mathbf{x}}) \right\|_2^2 \right] - H\left(\mathbf{g}^*(\mathbf{x})\right) \tag{26}$$

$$= \mathbb{E}_{(\mathbf{x},\tilde{\mathbf{x}}) \sim p_{(\mathbf{x},\tilde{\mathbf{x}})}} \left[ \left\| \mathbf{d}(\mathbf{c}) - \mathbf{d}(\tilde{\mathbf{c}}) \right\|_2^2 \right] - H\left(\mathbf{d}(\mathbf{c})\right) \tag{27}$$

$$= 0 \tag{28}$$

where in the last step we have used the fact that $\mathbf{c} = \tilde{\mathbf{c}}$ almost surely w.r.t. to the ground truth generative process $p$ described in § 3, so the first term is zero; and the fact that $\mathbf{d}(\mathbf{c})$ is uniformly distributed on $(0,1)^{n_c}$ and the uniform distribution on the unit hypercube has zero entropy, so the second term is also zero.

Next, let $\mathbf{g} : \mathcal{X} \to (0,1)^{n_c}$ be *any* smooth function which attains the global minimum of (5), i.e.,

$$\mathcal{L}_{\text{AlignMaxEnt}}(\mathbf{g}) = \mathbb{E}_{(\mathbf{x},\tilde{\mathbf{x}}) \sim p_{(\mathbf{x},\tilde{\mathbf{x}})}} \left[ \left\| \mathbf{g}(\mathbf{x}) - \mathbf{g}(\tilde{\mathbf{x}}) \right\|_2^2 \right] - H\left(\mathbf{g}(\mathbf{x})\right) = 0. \tag{29}$$

Define $\mathbf{h} := \mathbf{g} \circ \mathbf{f} : \mathcal{Z} \to (0,1)^{n_c}$ which is smooth because both $\mathbf{g}$ and $\mathbf{f}$ are smooth.

Writing $\mathbf{x} = \mathbf{f}(\mathbf{z})$, (29) then implies in terms of $\mathbf{h}$:

$$\mathbb{E}_{(\mathbf{x},\tilde{\mathbf{x}}) \sim p_{(\mathbf{x},\tilde{\mathbf{x}})}} \left[ \left\| \mathbf{h}(\mathbf{z}) - \mathbf{h}(\tilde{\mathbf{z}}) \right\|_2^2 \right] = 0 \,, \tag{30}$$

$$H\left(\mathbf{h}(\mathbf{z})\right) = 0 \,. \tag{31}$$

Equation (30) implies that the same invariance condition (9) used in the proofs of Thms. 4.2 and 4.3 must hold (a.s. w.r.t. $p$), and (31) implies that $\hat{\mathbf{c}} = \mathbf{h}(\mathbf{z})$ must be uniformly distributed on $(0,1)^{n_c}$.

**Step 2.** Next, we show that $\mathbf{h}(\mathbf{z}) = \mathbf{h}(\mathbf{c}, \mathbf{s})$ can only depend on the true content $\mathbf{c}$ and not on any of the style variables $\mathbf{s}$. For this we use the same Step 2 as in the proofs of Thms. 4.2 and 4.3.

**Step 3.** Finally, we show that the mapping $\hat{\mathbf{c}} = \mathbf{h}(\mathbf{c})$ is invertible.

To this end, we make use of the following result from [129].

**Proposition A.2** (Proposition 5 of [129]). *Let $\mathcal{M}, \mathcal{N}$ be simply connected and oriented $\mathcal{C}^1$ manifolds without boundaries and $h : \mathcal{M} \to \mathcal{N}$ be a differentiable map. Further, let the random variable $\mathbf{z} \in \mathcal{M}$ be distributed according to $\mathbf{z} \sim p(\mathbf{z})$ for a regular density function $p$, i.e., $0 < p < \infty$. If the pushforward $p_{\#h}(\mathbf{z})$ of $p$ through $h$ is also a regular density, i.e., $0 < p_{\#h} < \infty$, then $h$ is a bijection.*

We apply this result to the simply connected and oriented $\mathcal{C}^1$ manifolds without boundaries $\mathcal{M} = \mathcal{C}$ and $\mathcal{N} = (0,1)^{n_c}$, and the smooth (hence, differentiable) map $\mathbf{h} : \mathcal{C} \to (0,1)^{n_c}$ which maps the random variable $\mathbf{c}$ to a uniform random variable $\hat{\mathbf{c}}$ (as established in Step 1).

Since both $p_{\mathbf{c}}$ (by assumption) and the uniform distribution (the pushforward of $p_{\mathbf{c}}$ through $\mathbf{h}$) are regular densities in the sense of Prop. A.2, we conclude that $\mathbf{h}$ is a bijection, i.e., invertible.

We have shown that for any smooth $\mathbf{g} : \mathcal{X} \to (0,1)^{n_c}$ which minimises $\mathcal{L}_{\text{AlignMaxEnt}}$, we have that $\hat{\mathbf{c}} = \mathbf{g}(\mathbf{x}) = \mathbf{h}(\mathbf{c})$ for a smooth and invertible $\mathbf{h} : \mathcal{C} \to (0,1)^{n_c}$, i.e., $\mathbf{c}$ is block-identified by $\mathbf{g}$. $\square$

# B    Additional details on the Causal3DIdent data set

Using the Blender rendering engine [11], 3DIdent [129] is a recently proposed benchmark which contains hallmarks of natural environments (e.g. shadows, different lighting conditions, a 3D object), but allows for identifiability evaluation by exposing the underlying generative factors.

Each $224 \times 224 \times 3$ image in the dataset shows a coloured 3D object which is located and rotated above a coloured ground in a 3D space. Furthermore, each scene contains a coloured spotlight which is focused on the object and located on a half-circle around the scene. The images are rendered based on a 10-dimensional latent, where: (i) three dimensions describe the XYZ position of the object, (ii) three dimensions describe the rotation of the object in Euler angles, (iii) two dimensions describe the colour (hue) of the object and the ground of the scene, respectively, and (iv) two dimensions describe the position and colour (hue) of the spotlight. For influence of the latent factors on the renderings, see Fig. 2 of [129].

## B.1    Details on introduced object classes

3DIdent contained a single object class, Teapot [89]. We add **six** additional object classes: Hare [121], Dragon [110], Cow [62], Armadillo [70], Horse [98], Head [111].

## B.2    Details on latent causal graph

In 3DIdent, the latents are uniformly sampled independently. We instead impose a causal graph over the variables (see Fig. 2). While object class and all environment variables (spotlight position, spotlight hue, background hue) are sampled independently, all object variables are dependent. Specifically, for spotlight position, spotlight hue, and background hue, we sample from $U(-1, 1)$. We impose the dependence by varying the mean ($\mu$) of a truncated normal distribution with standard deviation $\sigma = 0.5$, truncated to the range $[-1, 1]$.

Object rotation is dependent solely on object class, see Tab. 2 for details. Object position is dependent on both object class & spotlight position, see Tab. 3. Object hue is dependent on object class, background hue, & object hue, see Tab. 4. Hares blending into their environment as a form of active camouflage has been observed in Alaskan [78], Arctic [2], & Snowshoe hares.

## B.3    Dataset Visuals

We show 40 random samples from the marginal of each object class in Causal3DIdent in Figs. 3 to 9.

Table 2: Given a certain object class, the center of the truncated normal distribution from which we sample *rotation* latents varies.

| object class | $\mu(\phi)$ | $\mu(\theta)$ | $\mu(\psi)$ |
|---|---|---|---|
| Teapot | -0.35 | 0.35 | 0.35 |
| Hare | 0.35 | -0.35 | 0.35 |
| Dragon | 0.35 | 0.35 | -0.35 |
| Cow | 0.35 | -0.35 | -0.35 |
| Armadillo | -0.35 | 0.35 | -0.35 |
| Horse | -0.35 | -0.35 | 0.35 |
| Head | -0.35 | -0.35 | -0.35 |

Table 3: Given a certain object class & spotlight position, the center of the truncated normal distribution from which we sample $xy$-*position* latents varies. Note the spotlight position $\text{pos}_\text{spl}$ is rescaled from $[-1, 1]$ to $[-\pi/2, \pi/2]$.

| object class | $\mu(x)$ | $\mu(y)$ | $\mu(z)$ |
|---|---|---|---|
| Teapot | $0$ | $0$ | $0$ |
| Hare | $-\sin(\text{pos}_\text{spl})$ | $-\cos(\text{pos}_\text{spl})$ | $0$ |
| Dragon | $-\sin(\text{pos}_\text{spl})$ | $-\cos(\text{pos}_\text{spl})$ | $0$ |
| Cow | $\sin(\text{pos}_\text{spl})$ | $\cos(\text{pos}_\text{spl})$ | $0$ |
| Armadillo | $\sin(\text{pos}_\text{spl})$ | $\cos(\text{pos}_\text{spl})$ | $0$ |
| Horse | $-\sin(\text{pos}_\text{spl})$ | $-\cos(\text{pos}_\text{spl})$ | $0$ |
| Head | $\sin(\text{pos}_\text{spl})$ | $\cos(\text{pos}_\text{spl})$ | $0$ |

Table 4: Given a certain object class, background hue, and spotlight hue, the center of the truncated normal distribution from which we sample the *object hue* latent varies. Note that for the Hare and Dragon classes, in particular, the object either blends in or stands out from the environment.

| object class | $\mu(\text{hue})$ |
|---|---|
| Teapot | $0$ |
| Hare | $\frac{\text{hue}_\text{bg}+\text{hue}_\text{spl}}{2}$ |
| Dragon | $-\frac{\text{hue}_\text{bg}+\text{hue}_\text{spl}}{2}$ |
| Cow | $-0.35$ |
| Armadillo | $0.7$ |
| Horse | $-0.7$ |
| Head | $0.35$ |

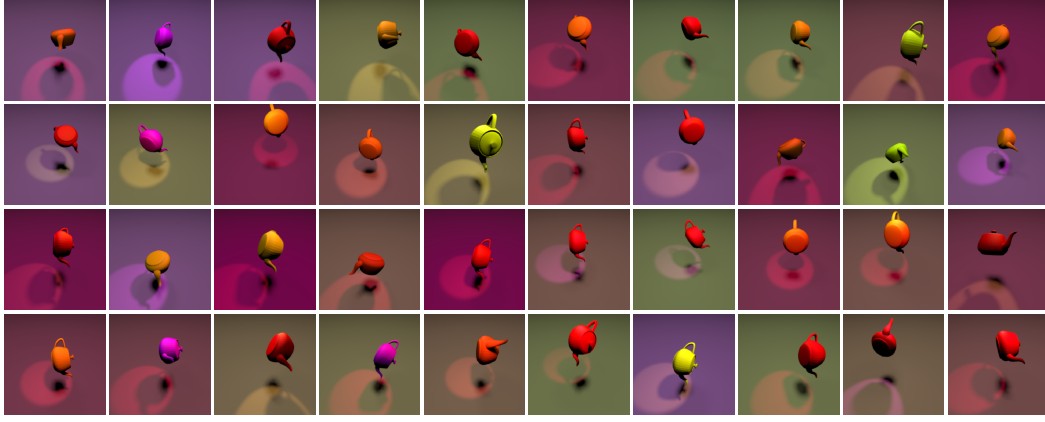

Figure 3: 40 random samples from the marginal distribution of the *Teapot* object class.

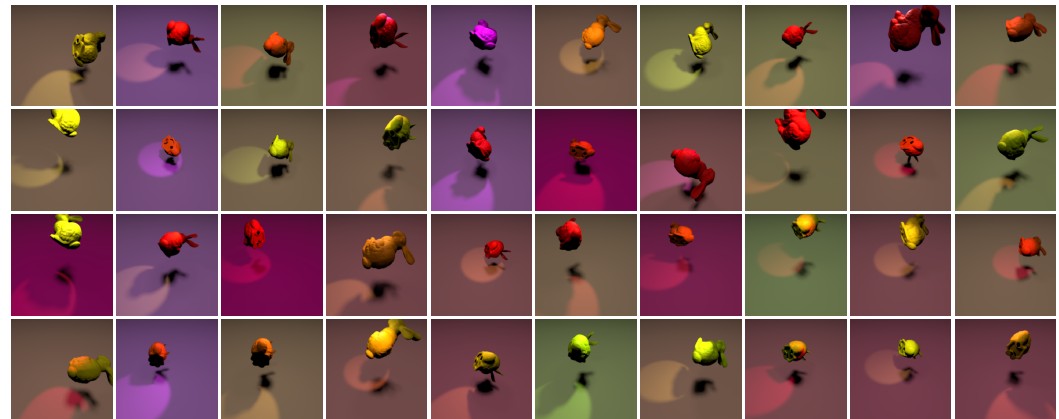

Figure 4: 40 random samples from the marginal distribution of the *Hare* object class.

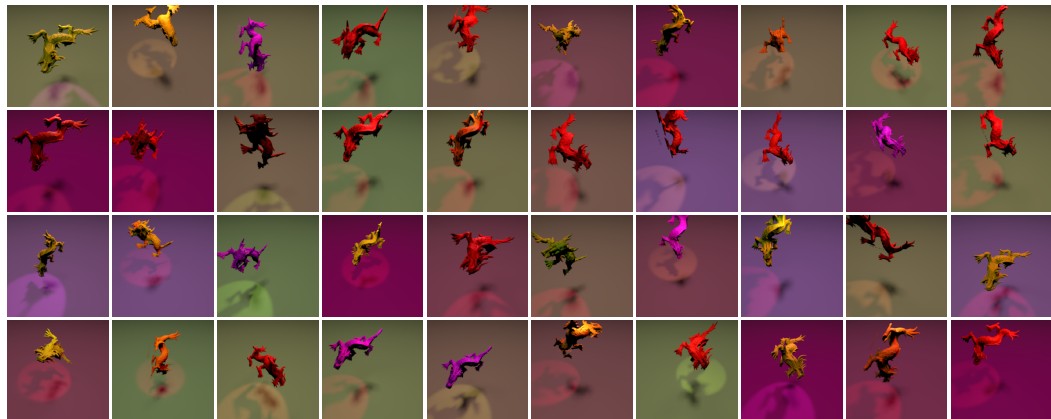

Figure 5: 40 random samples from the marginal distribution of the *Dragon* object class.

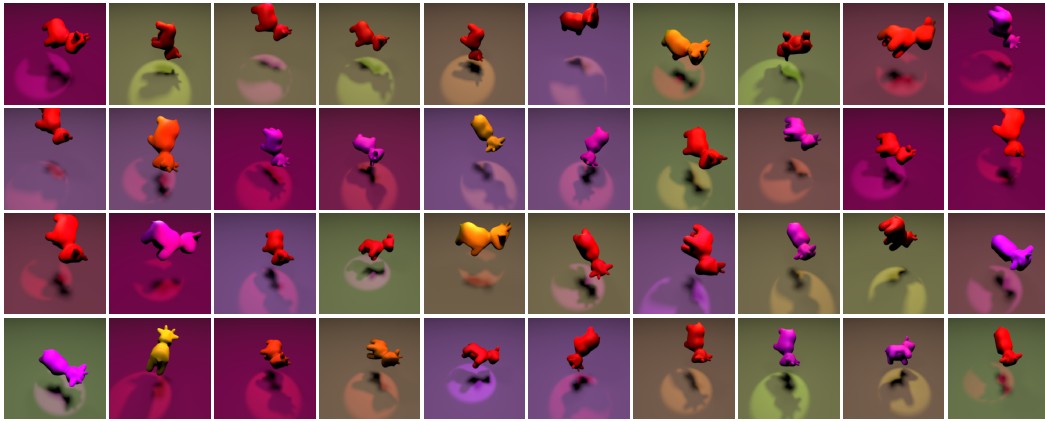

Figure 6: 40 random samples from the marginal distribution of the *Cow* object class.

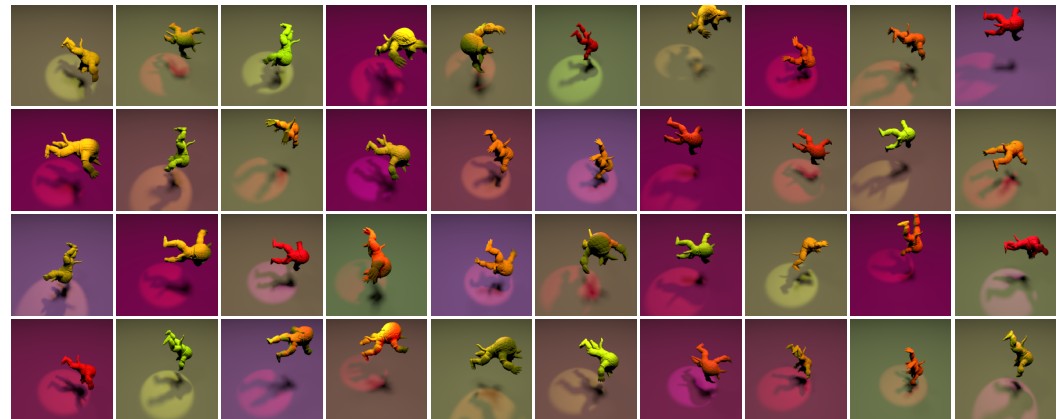

Figure 7: 40 random samples from the marginal distribution of the *Armadillo* object class.

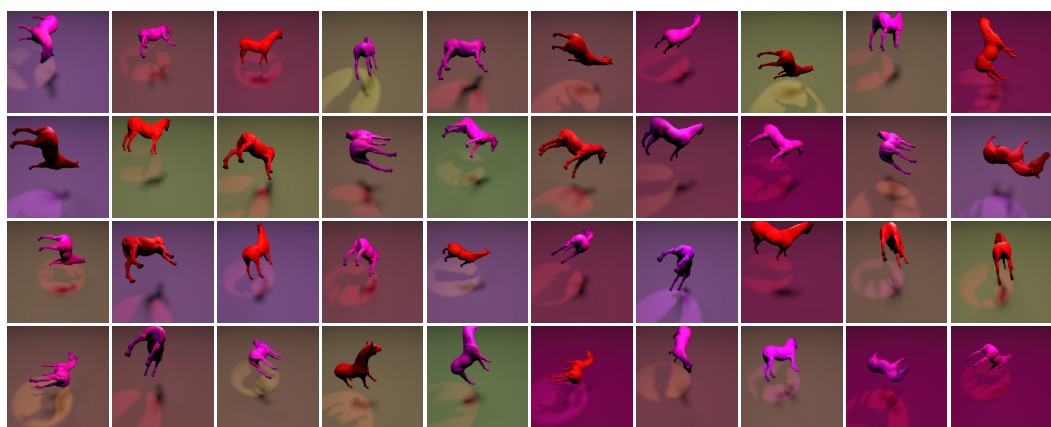

Figure 8: 40 random samples from the marginal distribution of the *Horse* object class.

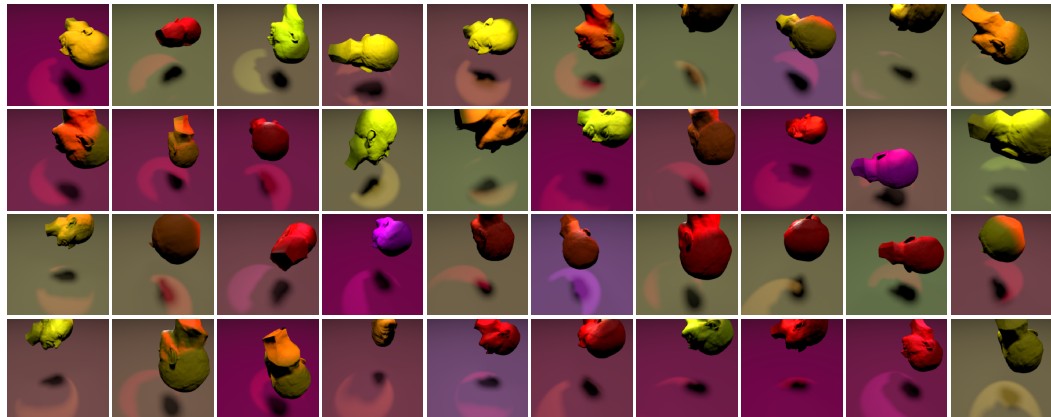

Figure 9: 40 random samples from the marginal distribution of the *Head* object class.

# C Additional results

- Appendix C.1 contains numerical experiments, namely linear evaluation & an ablation on $\dim(\hat{\mathbf{c}})$.
- Appendix C.2 contains experiments on *Causal3DIdent*, namely (i) nonlinear & linear evaluation results of the output & intermediate feature representation of `SimCLR` with results for the individual axes of object position & rotation, and (ii) evaluation of `BarlowTwins`.
- Appendix C.3 contains experiments on the *MPI3D-real* dataset [38], namely SimCLR & a supervised sanity check.

## C.1 Numerical Data

In Tab. 5, we report mean $\pm$ std. dev. $R^2$ over 3 random seeds across four generative processes of increasing complexity using *linear* (instead of nonlinear) regression to predict $\mathbf{c}$ from $\hat{\mathbf{c}}$. The block-identification of content can clearly still be seen even if we consider a linear fit.

In Fig. 10, we perform an ablation on $\dim(\hat{\mathbf{c}})$, visualising how varying the dimensionality of the learnt representation affects identifiability of the ground-truth content & style partition. Generally, if $\dim(\hat{\mathbf{c}}) < n_c$, there is insufficient capacity to encode all content, so a lower-dimensional mixture of content is learnt. Conversely, if $\dim(\hat{\mathbf{c}}) > n_c$, the excess capacity is used to encode some style information, as that increases entropy.

Table 5: Results using linear regression for the experiment on numerical data presented in § 5.1

| Generative process | | | $R^2$ (linear) | |
|---|---|---|---|---|
| p(chg.) | Stat. | Cau. | Content c | Style s |
| 1.0 | ✗ | ✗ | **1.00** $\pm$ 0.00 | 0.00 $\pm$ 0.00 |
| 0.75 | ✗ | ✗ | **0.99** $\pm$ 0.00 | 0.00 $\pm$ 0.00 |
| 0.75 | ✓ | ✗ | **0.97** $\pm$ 0.03 | 0.37 $\pm$ 0.05 |
| 0.75 | ✓ | ✓ | **0.98** $\pm$ 0.01 | **0.78** $\pm$ 0.07 |

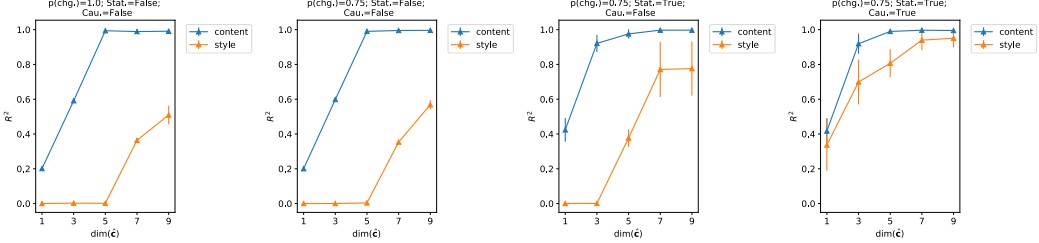

Figure 10: Identifiability of the content & style partition in the numerical experiment as a function of the model latent dimensionality

**On Dependence.** As can be seen from Tab. 5, the corresponding inset table in § 5.1, and Fig. 10, scores for identifying style increase substantially when statistical dependence within blocks and causal dependence between blocks are included. This finding can be explained as follows.

If we compare the performance for small latent dimensionalities ($\dim(\hat{\mathbf{c}}) < n_c$) between the first two (without) and the third plot (with statistical dependence) of Fig. 10, we observe a significantly higher score in identifying content for the latter (e.g., $R^2$ of ca. 0.4 vs 0.2 at $\dim(\hat{\mathbf{c}}) = 1$). This suggest that the introduction of statistical dependence between content variables (as well as between style variables, and in how style variables change) in the third plot/row, reduces the effective dimensionality of the ground-truth latents and thus leads to higher content identifiability for the same $\dim(\hat{\mathbf{c}}) < n_c$. Since the $R^2$ for content is already close to 1 for $\dim(\hat{\mathbf{c}}) = 3$ in the third plot of Fig. 10 (due to the smaller effective dimensionality induced by statistical dependence between $\mathbf{c}$), when $\dim(\hat{\mathbf{c}}) = n_c = 5$ is used (as reported in Tab. 5), excess capacity is used to encode style, leading to a positive $R^2$.

Regarding causal dependence (i.e., the fourth plot in Fig. 10 and fourth row in Tab. 5), we note that the ground truth dependence between $\mathbf{c}$ and $\mathbf{s}$ is linear, i.e., $p(\mathbf{s}|\mathbf{c})$ is centred at a linear transformation $\mathbf{a} + B\mathbf{c}$ of $\mathbf{c}$, see the data generating process in Appendix D for details. Given that our evaluation

consists of predicting the ground truth $\mathbf{c}$ and $\mathbf{s}$ from the learnt representation $\hat{\mathbf{c}} = \mathbf{g}(\mathbf{x})$, if we were to block-identify $\mathbf{c}$ according to Defn. 4.1, we should be able to also predict some aspects of $\mathbf{s}$ from $\hat{\mathbf{c}}$, due to the linear dependence between $\mathbf{c}$ and $\mathbf{s}$. This manifests in a relatively large $R^2$ for $\mathbf{s}$ in the last row of Tab. 5 and the corresponding table in § 5.1.

To summarise, we highlight two main takeaways: (i) when latent dependence is present, this may reduce the effective dimensionality, so that some style is encoded in addition to content unless a smaller representation size is chosen; (ii) even though the learnt representation isolates content in the sense of Defn. 4.1, it may still be predictive of style when content and style are (causally) dependent.

## C.2  *Causal3DIdent*

**Full version of Tab. 1:**  In Tab. 6, we a) provide the results for the individual axes of object position & rotation and b) present additional rows omitted from Tab. 1 for space considerations.

Interestingly, we find that the variance across the individual axes is significantly higher for object position than object rotation. If we compare the causal dependence imposed for object position (see Tab. 3) to the causal dependence imposed for object rotation (see Tab. 2), we can observe that the dependence imposed over individual axes is also significantly more variable for position than rotation, i.e., for $x$ the sine nonlinearity is used, for $y$ the cosine nonlinearity is used, while for $z$, no dependence is imposed.

Regarding the additional rows, we can observe that the composition of image-level rotation & crops yields results quite similar to solely using crops, a relationship which mirrors how transforming the rotation & position latents yields results quite similar to solely transforming the position latents. This suggests that the rotation variables are difficult to disentangle from the position variables in Causal3DIdent, regardless of whether data augmentation or latent transforms are used.

Finally, we can observe that applying image-level rotation in conjunction with small crops & colour distortion does lead to a difference in the encoding, background hue is preserved, while the scores for object position & rotation appear to slightly decrease. When using three augmentations as opposed to two, the effects of the individual augmentations are lessened. While colour distortion discourages the encoding of background hue, both small crops & image-level rotation encourages it, and thus it is preserved when all three augmentations are used. While colour distortion encourages the encoding of object position & rotation, both small crops & image-level rotation discourage it, but as a causal relationship exists between the class variable and said latents, the scores merely decrease, the latents are still for the most part preserved. In reality, where complex interactions between latent variables abound, the effect of data augmentations may be uninterpretable, however with Causal3DIdent, we are able to interpret their effects in the presence of rich visual complexity and causal dependencies, even when applying three distinct augmentations in tandem.

Table 6: Full version of Tab. 1.

| Views generated by | Class | Positions | | | | Hues | | | Rotations | | |
|---|---|---|---|---|---|---|---|---|---|---|---|
| | | object($x$) | object($y$) | object($z$) | spotlight | object | spotlight | background | object($\phi$) | object($\theta$) | object($\psi$) |
| DA: colour distortion | 0.42 ± 0.01 | **0.58** ± 0.01 | **0.75** ± 0.00 | **0.52** ± 0.01 | 0.17 ± 0.00 | 0.10 ± 0.01 | 0.01 ± 0.00 | 0.01 ± 0.00 | 0.36 ± 0.01 | 0.33 ± 0.01 | 0.32 ± 0.00 |
| LT: change hues | **1.00** ± 0.00 | **0.81** ± 0.02 | **0.81** ± 0.02 | 0.15 ± 0.02 | **0.91** ± 0.00 | 0.30 ± 0.00 | 0.00 ± 0.00 | 0.00 ± 0.00 | 0.30 ± 0.02 | 0.30 ± 0.01 | 0.30 ± 0.01 |
| DA: crop (large) | 0.28 ± 0.04 | 0.04 ± 0.02 | 0.03 ± 0.01 | 0.19 ± 0.02 | 0.21 ± 0.13 | **0.87** ± 0.00 | 0.09 ± 0.02 | **1.00** ± 0.00 | 0.00 ± 0.00 | 0.05 ± 0.00 | 0.02 ± 0.00 |
| DA: crop (small) | 0.14 ± 0.00 | 0.00 ± 0.00 | 0.01 ± 0.02 | 0.00 ± 0.00 | 0.00 ± 0.01 | 0.00 ± 0.00 | 0.00 ± 0.00 | **1.00** ± 0.00 | 0.00 ± 0.00 | 0.00 ± 0.00 | 0.00 ± 0.00 |
| LT: change positions | **1.00** ± 0.00 | 0.01 ± 0.00 | 0.47 ± 0.01 | 0.01 ± 0.00 | 0.00 ± 0.01 | 0.46 ± 0.02 | 0.00 ± 0.00 | **0.97** ± 0.00 | 0.30 ± 0.00 | 0.29 ± 0.00 | 0.28 ± 0.00 |
| DA: crop (large) + colour distortion | **0.97** ± 0.00 | **0.59** ± 0.03 | **0.52** ± 0.01 | **0.68** ± 0.01 | **0.59** ± 0.05 | 0.28 ± 0.00 | 0.01 ± 0.01 | 0.01 ± 0.00 | **0.74** ± 0.01 | **0.78** ± 0.00 | **0.72** ± 0.00 |
| DA: crop (small) + colour distortion | **1.00** ± 0.00 | **0.72** ± 0.02 | **0.65** ± 0.02 | **0.70** ± 0.00 | **0.93** ± 0.00 | 0.30 ± 0.01 | 0.00 ± 0.00 | 0.02 ± 0.03 | **0.53** ± 0.00 | **0.57** ± 0.01 | **0.58** ± 0.01 |
| LT: change positions + hues | **1.00** ± 0.00 | 0.10 ± 0.10 | 0.49 ± 0.02 | 0.06 ± 0.05 | 0.07 ± 0.08 | 0.32 ± 0.02 | 0.00 ± 0.01 | 0.02 ± 0.03 | 0.34 ± 0.09 | 0.34 ± 0.04 | 0.34 ± 0.08 |
| DA: rotation | 0.33 ± 0.06 | 0.29 ± 0.03 | 0.11 ± 0.01 | 0.12 ± 0.04 | 0.23 ± 0.12 | **0.83** ± 0.01 | 0.30 ± 0.12 | **0.99** ± 0.00 | 0.02 ± 0.01 | 0.06 ± 0.03 | 0.07 ± 0.01 |
| LT: change rotations | **1.00** ± 0.00 | **0.78** ± 0.01 | **0.72** ± 0.03 | 0.09 ± 0.03 | **0.90** ± 0.00 | 0.41 ± 0.00 | 0.00 ± 0.00 | **0.97** ± 0.00 | 0.28 ± 0.00 | 0.28 ± 0.00 | 0.28 ± 0.00 |
| DA: rotation + colour distortion | **0.59** ± 0.01 | **0.63** ± 0.01 | **0.57** ± 0.08 | **0.54** ± 0.02 | 0.21 ± 0.01 | 0.12 ± 0.02 | 0.01 ± 0.00 | 0.01 ± 0.00 | 0.36 ± 0.03 | 0.34 ± 0.04 | 0.30 ± 0.03 |
| LT: change rotations + hues | **1.00** ± 0.00 | **0.80** ± 0.02 | **0.77** ± 0.01 | 0.13 ± 0.02 | **0.91** ± 0.00 | 0.30 ± 0.00 | 0.00 ± 0.00 | 0.00 ± 0.00 | 0.28 ± 0.00 | 0.28 ± 0.01 | 0.28 ± 0.00 |
| DA: rot. + crop (lg) | 0.26 ± 0.01 | 0.03 ± 0.02 | 0.03 ± 0.01 | 0.15 ± 0.04 | 0.04 ± 0.03 | **0.84** ± 0.06 | 0.10 ± 0.01 | **1.00** ± 0.00 | 0.00 ± 0.00 | 0.04 ± 0.02 | 0.02 ± 0.00 |
| DA: rot. + crop (sm) | 0.15 ± 0.00 | 0.00 ± 0.00 | 0.00 ± 0.00 | 0.00 ± 0.00 | 0.00 ± 0.00 | 0.00 ± 0.00 | 0.00 ± 0.00 | **1.00** ± 0.00 | 0.00 ± 0.00 | 0.00 ± 0.00 | 0.00 ± 0.00 |
| LT: change rot. + pos. | **1.00** ± 0.00 | 0.02 ± 0.03 | 0.48 ± 0.02 | 0.01 ± 0.01 | 0.02 ± 0.03 | 0.49 ± 0.03 | 0.03 ± 0.02 | **0.98** ± 0.00 | 0.29 ± 0.01 | 0.28 ± 0.01 | 0.28 ± 0.01 |
| DA: rot. + crop (lg) + col. dist. | 0.99 ± 0.00 | **0.69** ± 0.03 | **0.60** ± 0.01 | **0.70** ± 0.02 | **0.86** ± 0.03 | 0.28 ± 0.00 | 0.01 ± 0.00 | 0.01 ± 0.00 | **0.60** ± 0.01 | **0.64** ± 0.02 | **0.61** ± 0.01 |
| DA: rot. + crop (sm) + col. dist. | **1.00** ± 0.00 | **0.61** ± 0.02 | **0.59** ± 0.01 | **0.64** ± 0.01 | **0.82** ± 0.01 | 0.38 ± 0.00 | 0.01 ± 0.01 | **0.78** ± 0.03 | 0.44 ± 0.00 | 0.48 ± 0.02 | 0.45 ± 0.01 |
| LT: change rot. + pos. + hues | **1.00** ± 0.00 | 0.20 ± 0.12 | **0.50** ± 0.04 | 0.14 ± 0.11 | 0.15 ± 0.12 | 0.32 ± 0.01 | 0.00 ± 0.00 | 0.02 ± 0.01 | 0.33 ± 0.04 | 0.33 ± 0.02 | 0.32 ± 0.03 |

**Linear identifiability:**  In Tab. 7, we present results evaluating all continuous variables with linear regression. While, as expected, $R^2$ scores are reduced across the board, we can observe that even with a linear fit, the patterns observed in Tab. 6 persist.

**Intermediate feature evaluation:**  In Tab. 8 and Tab. 9, we present evaluation based on the representation from an intermediate layer (i.e., prior to applying a projection layer [20]) with nonlinear and linear regression for the continuous variables, respectively. Note the intermediate layer has an

Table 7: Evaluation results using a linear fit for not only class, but all continuous variables.

| Views generated by | Class | Positions | | | | Hues | | | Rotations | | |
|---|---|---|---|---|---|---|---|---|---|---|---|
| | | object($x$) | object($y$) | object($z$) | spotlight | object | spotlight | background | object($\phi$) | object($\theta$) | object($\psi$) |
| DA: colour distortion | 0.42±0.01 | 0.37±0.03 | 0.20±0.16 | 0.23±0.02 | 0.01±0.01 | 0.03±0.01 | −0.00±0.00 | −0.00±0.00 | 0.13±0.01 | 0.04±0.01 | 0.09±0.02 |
| LT: change hues | 1.00±0.00 | 0.72±0.07 | 0.56±0.04 | −0.00±0.00 | 0.65±0.07 | 0.29±0.01 | −0.00±0.00 | −0.00±0.00 | 0.27±0.01 | 0.26±0.03 | 0.26±0.01 |
| DA: crop (large) | 0.28±0.04 | 0.00±0.00 | 0.02±0.00 | 0.04±0.07 | 0.08±0.13 | 0.51±0.05 | 0.03±0.02 | 0.20±0.04 | 0.00±0.00 | 0.02±0.00 | 0.01±0.00 |
| DA: crop (small) | 0.14±0.00 | −0.00±0.00 | −0.00±0.00 | −0.00±0.00 | −0.00±0.00 | −0.00±0.00 | −0.00±0.00 | 0.17±0.05 | −0.00±0.00 | −0.00±0.00 | −0.00±0.00 |
| LT: change positions | 1.00±0.00 | −0.00±0.00 | 0.44±0.02 | −0.00±0.00 | −0.00±0.00 | 0.29±0.04 | 0.00±0.00 | 0.73±0.16 | 0.26±0.01 | 0.25±0.03 | 0.25±0.04 |
| DA: crop (large) + colour distortion | 0.97±0.00 | 0.12±0.02 | 0.24±0.03 | 0.21±0.00 | 0.08±0.03 | 0.13±0.01 | −0.00±0.00 | −0.00±0.00 | 0.14±0.04 | 0.18±0.05 | 0.22±0.02 |
| DA: crop (small) + colour distortion | 1.00±0.00 | 0.35±0.02 | 0.50±0.01 | 0.19±0.03 | 0.80±0.01 | 0.28±0.00 | −0.00±0.00 | −0.00±0.00 | 0.29±0.00 | 0.30±0.00 | 0.29±0.01 |
| LT: change positions + hues | 1.00±0.00 | 0.00±0.00 | 0.42±0.06 | 0.00±0.00 | 0.00±0.00 | 0.27±0.02 | −0.00±0.00 | −0.00±0.00 | 0.23±0.07 | 0.26±0.03 | 0.25±0.04 |
| DA: rotation | 0.33±0.06 | 0.04±0.04 | 0.04±0.00 | 0.02±0.03 | 0.12±0.08 | 0.46±0.06 | 0.06±0.04 | 0.30±0.13 | 0.00±0.00 | 0.04±0.02 | 0.02±0.00 |
| LT: change rotations | 1.00±0.00 | 0.34±0.21 | 0.48±0.03 | −0.00±0.00 | 0.60±0.15 | 0.28±0.00 | 0.00±0.00 | 0.59±0.26 | 0.27±0.01 | 0.27±0.00 | 0.27±0.01 |
| DA: rotation + colour distortion | 0.59±0.01 | 0.31±0.02 | 0.26±0.06 | 0.25±0.07 | 0.02±0.00 | 0.03±0.02 | −0.00±0.00 | −0.00±0.00 | 0.07±0.01 | 0.06±0.01 | 0.10±0.01 |
| LT: change rotations + hues | 1.00±0.00 | 0.68±0.02 | 0.57±0.01 | −0.00±0.00 | 0.72±0.10 | 0.29±0.00 | −0.00±0.00 | −0.00±0.00 | 0.28±0.00 | 0.28±0.00 | 0.28±0.00 |
| DA: rot. + crop (lg) | 0.26±0.01 | −0.00±0.00 | 0.02±0.00 | 0.00±0.00 | 0.00±0.00 | 0.59±0.05 | 0.02±0.01 | 0.20±0.04 | 0.00±0.00 | 0.01±0.00 | 0.01±0.00 |
| DA: rot. + crop (sm) | 0.15±0.00 | −0.00±0.00 | −0.00±0.00 | −0.00±0.00 | −0.00±0.00 | −0.00±0.00 | 0.00±0.00 | 0.29±0.21 | −0.00±0.00 | −0.00±0.00 | −0.00±0.00 |
| LT: change rot. + pos. | 1.00±0.00 | −0.00±0.00 | 0.45±0.01 | −0.00±0.00 | −0.00±0.00 | 0.32±0.02 | 0.00±0.00 | 0.80±0.09 | 0.27±0.00 | 0.27±0.01 | 0.27±0.01 |
| DA: rot. + crop (lg) + col. dist. | 0.99±0.00 | 0.23±0.04 | 0.26±0.07 | 0.26±0.01 | 0.51±0.14 | 0.21±0.01 | −0.00±0.00 | −0.00±0.00 | 0.21±0.04 | 0.28±0.02 | 0.22±0.02 |
| DA: rot. + crop (sm) + col. dist. | 1.00±0.00 | 0.26±0.02 | 0.48±0.01 | 0.21±0.02 | 0.61±0.01 | 0.31±0.00 | −0.00±0.00 | 0.34±0.02 | 0.30±0.00 | 0.30±0.01 | 0.29±0.01 |
| LT: change rot. + pos. + hues | 1.00±0.00 | 0.03±0.05 | 0.46±0.01 | 0.01±0.01 | 0.01±0.02 | 0.29±0.01 | −0.00±0.00 | −0.00±0.00 | 0.27±0.00 | 0.28±0.01 | 0.28±0.01 |

output dimensionality of 100. While it is clear that all $R^2$ scores are increased across the board, we can notice that certain latents which were discarded in the final layer, were not in an intermediate layer. For example, with "LT: change hues", in the final layer the $z$-position was discarded ($R^2 = 0.15$ in Tab. 6), inexplicably we may add, as position is content regardless of axis with this latent transformation. But in the intermediate layer, $z$-position was not discarded ($R^2 = 0.88$ in Tab. 8).

Table 8: Evaluation of an intermediate layer. Logistic regression used for class, kernel ridge regression used for all continuous variables.

| Views generated by | Class | Positions | | | | Hues | | | Rotations | | |
|---|---|---|---|---|---|---|---|---|---|---|---|
| | | object($x$) | object($y$) | object($z$) | spotlight | object | spotlight | background | object($\phi$) | object($\theta$) | object($\psi$) |
| DA: colour distortion | 0.71±0.02 | 0.68±0.02 | 0.80±0.01 | 0.63±0.01 | 0.25±0.01 | 0.13±0.00 | 0.02±0.01 | 0.01±0.01 | 0.44±0.01 | 0.48±0.01 | 0.39±0.00 |
| LT: change hues | 1.00±0.00 | 0.98±0.00 | 0.97±0.00 | 0.88±0.01 | 0.98±0.00 | 0.34±0.01 | −0.00±0.00 | 0.20±0.10 | 0.71±0.02 | 0.68±0.03 | 0.68±0.02 |
| DA: crop (large) | 0.43±0.03 | 0.41±0.05 | 0.35±0.05 | 0.32±0.04 | 0.41±0.13 | 0.88±0.00 | 0.14±0.03 | 1.00±0.00 | 0.03±0.02 | 0.06±0.01 | 0.08±0.00 |
| DA: crop (small) | 0.20±0.01 | 0.04±0.05 | 0.20±0.02 | 0.01±0.02 | 0.20±0.03 | −0.00±0.00 | −0.00±0.00 | 1.00±0.00 | −0.00±0.00 | −0.00±0.00 | −0.00±0.00 |
| LT: change positions | 1.00±0.00 | 0.78±0.02 | 0.90±0.01 | 0.75±0.01 | 0.59±0.02 | 0.82±0.01 | 0.18±0.02 | 0.99±0.00 | 0.64±0.02 | 0.55±0.02 | 0.56±0.02 |
| DA: crop (large) + colour distortion | 1.00±0.00 | 0.92±0.00 | 0.83±0.00 | 0.92±0.00 | 0.90±0.01 | 0.29±0.00 | 0.01±0.01 | 0.01±0.01 | 0.87±0.00 | 0.90±0.00 | 0.85±0.00 |
| DA: crop (small) + colour distortion | 1.00±0.00 | 0.92±0.00 | 0.87±0.01 | 0.90±0.00 | 0.97±0.00 | 0.46±0.04 | 0.02±0.02 | 0.58±0.12 | 0.79±0.01 | 0.83±0.00 | 0.79±0.00 |
| LT: change positions + hues | 1.00±0.00 | 0.83±0.04 | 0.90±0.01 | 0.81±0.04 | 0.75±0.08 | 0.42±0.09 | 0.04±0.02 | 0.52±0.20 | 0.72±0.05 | 0.69±0.07 | 0.67±0.06 |
| DA: rotation | 0.46±0.04 | 0.35±0.04 | 0.19±0.02 | 0.28±0.04 | 0.34±0.08 | 0.85±0.01 | 0.35±0.12 | 1.00±0.00 | 0.03±0.01 | 0.08±0.02 | 0.10±0.01 |
| LT: change rotations | 1.00±0.00 | 0.97±0.00 | 0.96±0.01 | 0.84±0.01 | 0.98±0.00 | 0.82±0.01 | 0.17±0.02 | 0.99±0.00 | 0.64±0.02 | 0.59±0.01 | 0.60±0.03 |
| DA: rotation + colour distortion | 0.87±0.02 | 0.76±0.01 | 0.81±0.01 | 0.71±0.01 | 0.39±0.08 | 0.19±0.02 | −0.00±0.00 | 0.02±0.02 | 0.55±0.03 | 0.55±0.03 | 0.48±0.02 |
| LT: change rotations + hues | 1.00±0.00 | 0.98±0.00 | 0.97±0.00 | 0.87±0.00 | 0.99±0.00 | 0.39±0.05 | 0.04±0.02 | 0.37±0.21 | 0.69±0.01 | 0.68±0.01 | 0.68±0.00 |
| DA: rot. + crop (lg) | 0.43±0.03 | 0.38±0.04 | 0.34±0.02 | 0.28±0.03 | 0.30±0.05 | 0.86±0.04 | 0.17±0.02 | 1.00±0.00 | 0.02±0.00 | 0.05±0.01 | 0.10±0.01 |
| DA: rot. + crop (sm) | 0.20±0.01 | 0.07±0.03 | 0.09±0.10 | 0.01±0.01 | 0.20±0.01 | −0.00±0.00 | −0.00±0.00 | 1.00±0.00 | −0.00±0.00 | −0.00±0.00 | −0.00±0.00 |
| LT: change rot. + pos. | 1.00±0.00 | 0.81±0.01 | 0.90±0.01 | 0.76±0.01 | 0.67±0.04 | 0.84±0.01 | 0.28±0.04 | 0.99±0.00 | 0.62±0.02 | 0.57±0.01 | 0.55±0.01 |
| DA: rot. + crop (lg) + col. dist. | 1.00±0.00 | 0.92±0.01 | 0.89±0.00 | 0.92±0.00 | 0.95±0.01 | 0.30±0.00 | 0.02±0.02 | 0.18±0.16 | 0.81±0.00 | 0.84±0.00 | 0.79±0.00 |
| DA: rot. + crop (sm) + col. dist. | 1.00±0.00 | 0.87±0.00 | 0.85±0.00 | 0.87±0.00 | 0.93±0.00 | 0.71±0.02 | 0.33±0.05 | 0.96±0.00 | 0.72±0.00 | 0.75±0.00 | 0.71±0.00 |
| LT: change rot. + pos. + hues | 1.00±0.00 | 0.84±0.02 | 0.91±0.01 | 0.82±0.02 | 0.78±0.06 | 0.40±0.01 | 0.06±0.01 | 0.50±0.05 | 0.72±0.04 | 0.70±0.05 | 0.67±0.04 |

Table 9: Evaluation of an intermediate layer. Logistic regression used for class, linear regression used for all continuous variables.

| Views generated by | Class | Positions | | | | Hues | | | Rotations | | |
|---|---|---|---|---|---|---|---|---|---|---|---|
| | | object($x$) | object($y$) | object($z$) | spotlight | object | spotlight | background | object($\phi$) | object($\theta$) | object($\psi$) |
| DA: colour distortion | 0.71±0.02 | 0.53±0.01 | 0.70±0.01 | 0.46±0.01 | 0.13±0.01 | 0.11±0.01 | −0.01±0.00 | 0.00±0.00 | 0.28±0.01 | 0.19±0.01 | 0.25±0.01 |
| LT: change hues | 1.00±0.00 | 0.93±0.00 | 0.93±0.00 | 0.60±0.04 | 0.95±0.00 | 0.31±0.00 | 0.01±0.01 | 0.06±0.04 | 0.44±0.02 | 0.41±0.02 | 0.42±0.00 |
| DA: crop (large) | 0.43±0.03 | 0.18±0.06 | 0.06±0.01 | 0.17±0.02 | 0.19±0.14 | 0.82±0.02 | 0.08±0.04 | 0.98±0.00 | 0.01±0.00 | 0.05±0.01 | 0.05±0.01 |
| DA: crop (small) | 0.20±0.01 | 0.01±0.01 | 0.03±0.02 | 0.00±0.01 | 0.02±0.01 | −0.00±0.00 | −0.01±0.00 | 0.99±0.00 | −0.01±0.00 | −0.01±0.00 | −0.00±0.01 |
| LT: change positions | 1.00±0.00 | 0.49±0.04 | 0.72±0.03 | 0.43±0.03 | 0.19±0.03 | 0.71±0.02 | 0.09±0.02 | 0.98±0.00 | 0.39±0.01 | 0.36±0.01 | 0.35±0.00 |
| DA: crop (large) + colour distortion | 1.00±0.00 | 0.67±0.03 | 0.56±0.01 | 0.66±0.02 | 0.67±0.03 | 0.28±0.00 | −0.01±0.00 | 0.01±0.01 | 0.58±0.02 | 0.61±0.02 | 0.56±0.01 |
| DA: crop (small) + colour distortion | 1.00±0.00 | 0.76±0.01 | 0.70±0.02 | 0.68±0.01 | 0.90±0.00 | 0.38±0.03 | 0.00±0.01 | 0.39±0.13 | 0.50±0.02 | 0.50±0.01 | 0.49±0.01 |
| LT: change positions + hues | 1.00±0.00 | 0.61±0.09 | 0.74±0.02 | 0.51±0.08 | 0.40±0.15 | 0.34±0.04 | 0.02±0.01 | 0.25±0.22 | 0.47±0.04 | 0.40±0.02 | 0.41±0.03 |
| DA: rotation | 0.46±0.04 | 0.21±0.02 | 0.10±0.01 | 0.10±0.02 | 0.21±0.09 | 0.77±0.01 | 0.25±0.11 | 0.97±0.01 | 0.02±0.01 | 0.06±0.02 | 0.08±0.01 |
| LT: change rotations | 1.00±0.00 | 0.92±0.00 | 0.88±0.01 | 0.51±0.02 | 0.95±0.00 | 0.70±0.06 | 0.07±0.02 | 0.98±0.00 | 0.36±0.01 | 0.34±0.00 | 0.34±0.01 |
| DA: rotation + colour distortion | 0.87±0.02 | 0.60±0.01 | 0.62±0.03 | 0.52±0.02 | 0.23±0.02 | 0.18±0.02 | −0.01±0.00 | 0.02±0.01 | 0.33±0.04 | 0.29±0.01 | 0.28±0.01 |
| LT: change rotations + hues | 1.00±0.00 | 0.94±0.00 | 0.92±0.01 | 0.58±0.01 | 0.96±0.00 | 0.33±0.02 | 0.02±0.01 | 0.15±0.10 | 0.40±0.02 | 0.38±0.01 | 0.41±0.03 |
| DA: rot. + crop (lg) | 0.43±0.03 | 0.24±0.04 | 0.08±0.02 | 0.16±0.03 | 0.07±0.01 | 0.80±0.04 | 0.10±0.01 | 0.98±0.00 | 0.01±0.00 | 0.05±0.01 | 0.06±0.01 |
| DA: rot. + crop (sm) | 0.20±0.01 | 0.01±0.01 | 0.03±0.01 | −0.00±0.01 | 0.04±0.01 | −0.01±0.00 | −0.01±0.00 | 0.99±0.00 | −0.01±0.00 | −0.01±0.00 | −0.00±0.01 |
| LT: change rot. + pos. | 1.00±0.00 | 0.55±0.05 | 0.72±0.02 | 0.44±0.04 | 0.31±0.08 | 0.76±0.01 | 0.14±0.01 | 0.99±0.00 | 0.38±0.01 | 0.35±0.01 | 0.36±0.02 |
| DA: rot. + crop (lg) + col. dist. | 1.00±0.00 | 0.71±0.01 | 0.69±0.01 | 0.69±0.00 | 0.84±0.03 | 0.28±0.00 | −0.00±0.00 | 0.07±0.07 | 0.51±0.01 | 0.50±0.02 | 0.51±0.01 |
| DA: rot. + crop (sm) + col. dist. | 1.00±0.00 | 0.66±0.02 | 0.69±0.01 | 0.65±0.02 | 0.83±0.00 | 0.57±0.03 | 0.18±0.02 | 0.89±0.01 | 0.46±0.01 | 0.45±0.02 | 0.44±0.01 |
| LT: change rot. + pos. + hues | 1.00±0.00 | 0.65±0.04 | 0.75±0.05 | 0.57±0.03 | 0.49±0.12 | 0.35±0.01 | 0.02±0.01 | 0.23±0.04 | 0.48±0.04 | 0.43±0.01 | 0.43±0.01 |

In [20], the value in evaluating an intermediate layer as opposed to a final layer is discussed, where the authors demonstrated that predicting the data augmentations applied during training is significantly more accurate from an intermediate layer as opposed to the final layer, implying that the intermediate layer contains much more information about the transformation applied. Our results suggest a distinct hypothesis, the value in using an intermediate layer as a representation for downstream tasks is not due to preservation of style information, as can be seen, $R^2$ scores on style variables are not significantly higher in Tab. 8 relative to Tab. 6. The value is in preservation of all content variables, as we can observe certain content variables are discarded in the final layer, but are preserved in an

Table 10: *BarlowTwins* $\lambda = 0.0051$ results: $R^2$ mean $\pm$ std. dev. over 3 random seeds. DA: data augmentation, LT: latent transformation, bold: $R^2 \geq 0.5$, red: $R^2 < 0.25$. Results for individual axes of object position & rotation are aggregated.

| Views generated by | Class | Positions | | Hues | | | Rotations |
| --- | --- | --- | --- | --- | --- | --- | --- |
| | | object | spotlight | object | spotlight | background | |
| DA: colour distortion | $0.48 \pm 0.02$ | $\mathbf{0.51} \pm 0.14$ | $0.07 \pm 0.01$ | $0.08 \pm 0.00$ | $0.00 \pm 0.00$ | $0.00 \pm 0.00$ | $0.21 \pm 0.04$ |
| LT: change hues | $\mathbf{1.00} \pm 0.00$ | $\mathbf{0.56} \pm 0.20$ | $\mathbf{0.76} \pm 0.07$ | $0.30 \pm 0.01$ | $0.00 \pm 0.00$ | $0.01 \pm 0.00$ | $0.35 \pm 0.01$ |
| DA: crop (large) | $0.17 \pm 0.02$ | $0.10 \pm 0.03$ | $0.06 \pm 0.02$ | $0.29 \pm 0.13$ | $0.11 \pm 0.05$ | $\mathbf{0.99} \pm 0.00$ | $0.02 \pm 0.01$ |
| DA: crop (small) | $0.15 \pm 0.00$ | $0.04 \pm 0.02$ | $0.05 \pm 0.02$ | $0.02 \pm 0.01$ | $0.00 \pm 0.01$ | $\mathbf{1.00} \pm 0.00$ | $0.00 \pm 0.01$ |
| LT: change positions | $\mathbf{0.88} \pm 0.00$ | $0.19 \pm 0.20$ | $0.05 \pm 0.00$ | $\mathbf{0.50} \pm 0.02$ | $0.04 \pm 0.01$ | $\mathbf{0.98} \pm 0.00$ | $0.27 \pm 0.03$ |
| DA: crop (large) + colour distortion | $\mathbf{0.87} \pm 0.02$ | $0.49 \pm 0.06$ | $0.32 \pm 0.03$ | $0.25 \pm 0.01$ | $0.00 \pm 0.00$ | $0.00 \pm 0.00$ | $\mathbf{0.50} \pm 0.02$ |
| DA: crop (small) + colour distortion | $\mathbf{0.81} \pm 0.01$ | $0.39 \pm 0.07$ | $0.42 \pm 0.06$ | $0.47 \pm 0.04$ | $0.03 \pm 0.01$ | $\mathbf{0.85} \pm 0.02$ | $0.30 \pm 0.02$ |
| LT: change positions + hues | $\mathbf{1.00} \pm 0.00$ | $0.28 \pm 0.20$ | $0.12 \pm 0.05$ | $0.31 \pm 0.00$ | $0.00 \pm 0.00$ | $0.01 \pm 0.01$ | $0.37 \pm 0.06$ |

Table 11: *BarlowTwins* $\lambda = 0.051$ results: $R^2$ mean $\pm$ std. dev. over 3 random seeds. DA: data augmentation, LT: latent transformation, bold: $R^2 \geq 0.5$, red: $R^2 < 0.25$. Results for individual axes of object position & rotation are aggregated.

| Views generated by | Class | Positions | | Hues | | | Rotations |
| --- | --- | --- | --- | --- | --- | --- | --- |
| | | object | spotlight | object | spotlight | background | |
| DA: colour distortion | $\mathbf{0.52} \pm 0.07$ | $0.43 \pm 0.18$ | $0.07 \pm 0.02$ | $0.10 \pm 0.03$ | $0.00 \pm 0.00$ | $0.00 \pm 0.00$ | $0.21 \pm 0.05$ |
| LT: change hues | $\mathbf{1.00} \pm 0.00$ | $\mathbf{0.55} \pm 0.24$ | $\mathbf{0.74} \pm 0.02$ | $0.30 \pm 0.00$ | $0.00 \pm 0.00$ | $0.01 \pm 0.01$ | $0.33 \pm 0.02$ |
| DA: crop (large) | $0.19 \pm 0.05$ | $0.08 \pm 0.02$ | $0.05 \pm 0.01$ | $0.39 \pm 0.36$ | $0.08 \pm 0.05$ | $\mathbf{0.96} \pm 0.05$ | $0.01 \pm 0.01$ |
| DA: crop (small) | $0.15 \pm 0.00$ | $0.05 \pm 0.02$ | $0.07 \pm 0.02$ | $0.00 \pm 0.01$ | $0.01 \pm 0.01$ | $\mathbf{1.00} \pm 0.00$ | $0.00 \pm 0.00$ |
| LT: change positions | $\mathbf{0.89} \pm 0.01$ | $0.19 \pm 0.20$ | $0.05 \pm 0.01$ | $0.48 \pm 0.04$ | $0.05 \pm 0.02$ | $\mathbf{0.98} \pm 0.00$ | $0.25 \pm 0.03$ |
| DA: crop (large) + colour distortion | $\mathbf{0.86} \pm 0.03$ | $0.40 \pm 0.07$ | $0.23 \pm 0.02$ | $0.24 \pm 0.01$ | $0.00 \pm 0.00$ | $0.00 \pm 0.00$ | $0.47 \pm 0.04$ |
| DA: crop (small) + colour distortion | $\mathbf{0.99} \pm 0.01$ | $\mathbf{0.63} \pm 0.03$ | $\mathbf{0.88} \pm 0.01$ | $0.32 \pm 0.02$ | $0.00 \pm 0.00$ | $0.16 \pm 0.13$ | $\mathbf{0.52} \pm 0.03$ |
| LT: change positions + hues | $\mathbf{1.00} \pm 0.00$ | $0.21 \pm 0.22$ | $0.07 \pm 0.01$ | $0.30 \pm 0.00$ | $0.00 \pm 0.00$ | $0.02 \pm 0.01$ | $0.46 \pm 0.06$ |

intermediate layer. With that being said, our theoretical result applies to the final layer, which is why said results were highlighted in the main paper. The discarding of certain content variables is an empirical phenomenon, likely a consequence of a limited number of negative samples in practice, leading to certain content variables being redundant, or unnecessary, for solving the contrastive objective.

The fact that we can recover certain content variables which appeared discarded in the output from the intermediate layer may suggest that we should be able to decode class. While scores are certainly increased, we do not see such drastic differences in $R^2$ scores, as was seen above. The drastic difference highlighted above was with regards to latent transformation, for which we always observed class encoded as a content variable. So, unfortunately, using an intermediate layer does not rectify the discrepancy between data augmentations and latent transformations. While latent transformations allow us to better interpret the effect of certain empirical techniques [20], as discussed in the main paper, we cannot make a one-to-one correspondence between data augmentations used in practice and latent transformations.

**BarlowTwins:** We repeat our analysis from § 5.2 using `BarlowTwins` [128] (instead of `SimCLR`) which, as discussed at the end of § 4.2, is also loosely related to Thm. 4.4. The `BarlowTwins` objective consists of an invariance term, which equates the diagonal elements of the cross-correlation matrix to 1, thereby making the embedding invariant to the distortions applied and a redundancy reduction term, which equates the off-diagonal elements of the cross-correlation matrix to 0, thereby decorrelating the different vector components of the embedding, reducing the redundancy between output units.

In Tab. 10 we train `BarlowTwins` with $\lambda = 0.0051$, the default value for the hyperparameter which weights the redundancy reduction term relative to the invariance term. To confirm the insights are robust to the value of $\lambda$, in Tab. 11, we report results with $\lambda$ increased by an order of magnitude, $\lambda = 0.051$. We find that the results mirror Tab. 1, e.g. colour distortion yields a discarding of hue, crops isolate background hue where the larger the crop, the higher the identifiability of object hue, and crops & colour distortion yield high accuracy in inferring the object class variable.

## C.3 *MPI3D-real*

We ran the same experimental setup as in § 5.2 also on the *MPI3D-real* dataset [38] containing $> 1$ million *real* images with ground-truth annotations of 3D objects being moved by a robotic arm.

Table 12: *MPI3D-real* results: $R^2$ mean $\pm$ std. dev. over 3 random seeds for dim($\hat{\mathbf{c}}$)= 5. DA: data augmentation, bold: $R^2 \geq 0.5$, red: $R^2 < 0.25$.

| Views generated by | object color | object shape | object size | camera height | background color | horizontal axis | vertical axis |
|---|---|---|---|---|---|---|---|
| DA: colour distortion | $0.39 \pm 0.01$ | $0.00 \pm 0.00$ | $0.16 \pm 0.01$ | $\mathbf{1.00} \pm 0.00$ | $0.09 \pm 0.15$ | $\mathbf{0.60} \pm 0.06$ | $0.42 \pm 0.08$ |
| DA: crop (large) | $\mathbf{0.65} \pm 0.17$ | $0.01 \pm 0.02$ | $0.31 \pm 0.03$ | $\mathbf{1.00} \pm 0.00$ | $\mathbf{1.00} \pm 0.00$ | $0.37 \pm 0.06$ | $0.08 \pm 0.03$ |
| DA: crop (small) | $0.09 \pm 0.02$ | $0.03 \pm 0.00$ | $0.19 \pm 0.01$ | $\mathbf{1.00} \pm 0.00$ | $\mathbf{1.00} \pm 0.00$ | $0.21 \pm 0.02$ | $0.07 \pm 0.00$ |
| DA: crop (large) + colour distortion | $0.34 \pm 0.00$ | $0.00 \pm 0.00$ | $0.22 \pm 0.03$ | $\mathbf{1.00} \pm 0.00$ | $0.39 \pm 0.02$ | $\mathbf{0.54} \pm 0.01$ | $0.29 \pm 0.01$ |
| DA: crop (small) + colour distortion | $0.25 \pm 0.02$ | $0.00 \pm 0.00$ | $0.10 \pm 0.01$ | $\mathbf{1.00} \pm 0.00$ | $\mathbf{0.75} \pm 0.16$ | $\mathbf{0.54} \pm 0.01$ | $0.29 \pm 0.03$ |

Table 13: **Supervised** *MPI3D-real* results: $R^2$ mean $\pm$ std. dev. over 3 random seeds. DA: data augmentation. bold: $R^2 \geq 0.5$, red: $R^2 < 0.25$.

| Views generated by | object color | object shape | object size | camera height | background color | horizontal axis | vertical axis |
|---|---|---|---|---|---|---|---|
| Original | $\mathbf{0.90} \pm 0.01$ | $0.25 \pm 0.02$ | $\mathbf{0.61} \pm 0.02$ | $\mathbf{0.99} \pm 0.00$ | $\mathbf{0.97} \pm 0.01$ | $\mathbf{1.00} \pm 0.00$ | $\mathbf{1.00} \pm 0.00$ |
| DA: colour distortion | $\mathbf{0.61} \pm 0.01$ | $0.11 \pm 0.00$ | $0.47 \pm 0.01$ | $\mathbf{0.98} \pm 0.00$ | $\mathbf{0.93} \pm 0.00$ | $\mathbf{0.99} \pm 0.00$ | $\mathbf{1.00} \pm 0.00$ |
| DA: crop (large) | $\mathbf{0.82} \pm 0.01$ | $0.05 \pm 0.01$ | $0.42 \pm 0.02$ | $\mathbf{0.97} \pm 0.01$ | $\mathbf{0.91} \pm 0.00$ | $\mathbf{0.96} \pm 0.00$ | $\mathbf{0.97} \pm 0.01$ |
| DA: crop (small) | $\mathbf{0.71} \pm 0.04$ | $0.01 \pm 0.00$ | $0.32 \pm 0.02$ | $\mathbf{0.95} \pm 0.00$ | $\mathbf{0.85} \pm 0.01$ | $\mathbf{0.79} \pm 0.02$ | $\mathbf{0.90} \pm 0.01$ |
| DA: crop (large) + colour distortion | $0.45 \pm 0.02$ | $0.02 \pm 0.00$ | $0.22 \pm 0.00$ | $\mathbf{0.95} \pm 0.01$ | $\mathbf{0.67} \pm 0.01$ | $\mathbf{0.91} \pm 0.00$ | $\mathbf{0.94} \pm 0.00$ |
| DA: crop (small) + colour distortion | $0.45 \pm 0.02$ | $0.00 \pm 0.00$ | $0.17 \pm 0.02$ | $\mathbf{0.91} \pm 0.02$ | $\mathbf{0.55} \pm 0.03$ | $\mathbf{0.69} \pm 0.01$ | $\mathbf{0.79} \pm 0.08$ |

As *MPI3D-real* contains much lower resolution images ($64 \times 64$) compared to ImageNet & Causal3DIdent ($224 \times 224$), we used the standard convolutional encoder from the disentanglement literature [82], and ran a sanity check experiment to verify that by training the same backbone as in our unsupervised experiment with supervised learning, we can recover the ground-truth factors from the augmented views. In Tab. 13, we observe that only five out of seven factors can be consistently inferred, object shape and size are somewhat ambiguous even when observing the original image. Note that while in the self-supervised case, we evaluate by training a nonlinear regression for each ground truth factor separately, in the supervised case, we train a network for all ground truth factors simultaneously from scratch for as many gradient steps as used for learning the self-supervised model.

In Tab. 12, we report the evaluation results in the self-supervised scenario. Subject to the aforementioned caveats, the results show a similar trend as those on *Causal3DIdent*, i.e. with colour distortion, color factors of variation are decoded significantly worse than positional/rotational information.

## D  Experimental details

**Ground-truth generative model.** The generative process used in our numerical simulations (§ 5.1) is summarised by the following:

$$\mathbf{c} \sim p(\mathbf{c}) = \mathcal{N}(0, \Sigma_{\mathbf{c}}), \quad \text{with} \quad \Sigma_{\mathbf{c}} \sim \text{Wishart}_{n_c}(\mathbf{I}, n_c),$$

$$\mathbf{s}|\mathbf{c} \sim p(\mathbf{s}|\mathbf{c}) = \mathcal{N}(\mathbf{a} + B\mathbf{c}, \Sigma_{\mathbf{s}}), \quad \text{with} \quad \Sigma_{\mathbf{s}} \sim \text{Wishart}_{n_s}(\mathbf{I}, n_s), \quad a_i, b_{ij} \overset{\text{i.i.d.}}{\sim} \mathcal{N}(0, 1),$$

$$\tilde{\mathbf{s}}_A|\mathbf{s}_A, A \sim p(\tilde{\mathbf{s}}_A|\mathbf{s}_A) = N(\mathbf{s}_A, \Sigma(A)) \quad \text{with} \quad \Sigma \sim \text{Wishart}_{n_s}(\mathbf{I}, n_s),$$

$$(\tilde{\mathbf{x}}, \mathbf{x}) = (\mathbf{f}_{\text{MLP}}(\tilde{\mathbf{z}}), \mathbf{f}_{\text{MLP}}(\mathbf{z})),$$

where the set of changing style vectors $A$ is obtained by flipping a (biased) coin with p(chg.) = 0.75 for each style dimension independently, and where $\Sigma(A)$ denotes the submatrix of $\Sigma$ defined by selecting the rows and columns corresponding to subset $A$.

When we do not allow for *statistical dependence* (Stat.) within blocks of content and style variables, we set the covariance matrices $\Sigma_{\mathbf{c}}$, $\Sigma_{\mathbf{s}}$, and $\Sigma$ to the identity. When we do not allow for *causal dependence* (Cau.) of style on content, we set $a_i, b_{ij} = 0, \forall i, j$.

For $\mathbf{f}_{\text{MLP}}$, we use a 3-layer MLP with LeakyReLU ($\alpha = 0.2$) activation functions, specified using the same process as used in previous work [54, 55, 129]. For the square weight matrices, we draw $(n_c + n_s) \times (n_c + n_s)$ samples from $U(-1, 1)$, and perform $l_2$ column normalisation. In addition, to control for invertibility, we re-sample the weight matrices until their condition number is less than or equal to a threshold value. The threshold is pre-computed by sampling $24,975$ weight matrices, and recording the minimum condition number.

**Training encoder.** Recall that the result of Thm. 4.4 corresponds to minimizing the following functional (5):

$$\mathcal{L}_{\text{AlignMaxEnt}}(\mathbf{g}) := \mathbb{E}_{(\mathbf{x}, \tilde{\mathbf{x}}) \sim p_{\mathbf{x}, \tilde{\mathbf{x}}}} \left[ \left( \mathbf{g}(\mathbf{x}) - \mathbf{g}(\tilde{\mathbf{x}}) \right)^2 \right] - H \left( \mathbf{g}(\mathbf{x}) \right).$$

Note that InfoNCE [20, 91] (1) can be rewritten as:

$$\mathcal{L}_{\text{InfoNCE}}(\mathbf{g}; \tau, K) = \mathbb{E}_{\{\mathbf{x}_i, \tilde{\mathbf{x}}_i\}_{i=1}^K \sim p_{\mathbf{x}, \tilde{\mathbf{x}}}} \left[ -\sum_{i=1}^K \text{sim}(\mathbf{g}(\mathbf{x})_i, \mathbf{g}(\tilde{\mathbf{x}})_i)/\tau + \log \sum_{j=1}^K \exp\{\text{sim}(\mathbf{g}(\mathbf{x})_i, \mathbf{g}(\tilde{\mathbf{x}})_j)/\tau\} \right]. \quad (32)$$

Thus, if we consider $\tau = 1$, and $\text{sim}(u, v) = -(u - v)^2$,

$$\mathcal{L}_{\text{InfoNCE}}(\mathbf{g}; K) = \mathbb{E}_{\{\mathbf{x}_i, \tilde{\mathbf{x}}_i\}_{i=1}^K \sim p_{\mathbf{x}, \tilde{\mathbf{x}}}} \left[ \sum_{i=1}^K \left( \mathbf{g}(\mathbf{x})_i - \mathbf{g}(\tilde{\mathbf{x}})_i \right)^2 + \log \sum_{j=1}^K \exp\{-(\mathbf{g}(\mathbf{x})_i - \mathbf{g}(\tilde{\mathbf{x}})_j)^2\} \right] \quad (33)$$

we can approximately match the form of (5). In practice, we use $K = 6, 144$.

For $\mathbf{g}$, as in [129], we use a 7-layer MLP with (default) LeakyReLU ($\alpha = 0.01$) activation functions. As the input dimensionality is $(n_c + n_s)$, we consider the following multipliers $[10, 50, 50, 50, 50, 10]$ for the number of hidden units per layer. In correspondence with Thm. 4.4, we set the output dimensionality to $n_c$.

We train our feature encoder for $300,000$ iterations, using Adam [66] with a learning rate of $10^{-4}$.

**Causal3DIdent.** We here elaborate on details specific to the experiments in § 5.2. We train the feature encoder for $200,000$ iterations using Adam with a learning rate of $10^{-4}$. For the encoder we use a ResNet18 [46] architecture followed by a single hidden layer with dimensionality 100 and LeakyReLU activation function using the default (0.01) negative slope. The scores are evaluated on a test set consisting of $25,000$ samples not included in the training set.

**Data augmentations.** We here specify the parameters for the data augmentations we considered:

- colour distortion: see the paragraph labelled "Color distortion" in Appendix A of [20] for details. We use $s = 1.0$, the default value.
- crop: see the paragraph labelled "Random crop and resize to $224 \times 224$" in Appendix A of [20] for details. For small crops, a crop of random size (uniform from $0.08$ to $1.0$ in area) of the original size is made, which corresponds to what was used in the experiments reported in [20]. For large crops, a crop of random size (uniform from $0.8$ to $1.0$ in area) of the original size is made.
- rotation: as specified in the captions for Figure 4 & Table 3 in [20], we sample one of $\{0°, 90°, 180°, 270°\}$ uniformly. Note that for the pair, we sample two values without replacement.

A visual overview of the effect of these image-level data augmentations is shown in Fig. 11.

**Latent transformations.** To generate views via latent transformations (LT) in our experiments on Causal3DIdent (§ 5.2), we proceed as follows.

Let $\mathbf{z}$ refer to the latent corresponding to the original image. For all latents specified to change, we sample $\hat{\mathbf{z}}'$ from a truncated normal distribution constrained to $[-1, 1]$, centered at $\mathbf{z}$, with $\sigma = 1$.. Then, we use nearest-neighbor matching to find the latent $\hat{\mathbf{z}}$ closest to $\hat{\mathbf{z}}'$ (in $L^2$ distance) for which there exists an image rendering.[15]

**Evaluation.** Recall that Thm. 4.4 states that $\mathbf{g}$ block-identifies the true content variables in the sense of Defn. 4.1, i.e., there exists an *invertible* function $\mathbf{h} : \mathbb{R}^{n_c} \to \mathbb{R}^{n_c}$ s.t. $\hat{\mathbf{c}} = \mathbf{h}(\mathbf{c})$.

Since this is different from typical evaluation in disentanglement or ICA in that we do not assume independence and do not aim to find a one-to-one correspondence between inferred and ground truth latents, existing metrics, such as MCC [54, 55] or MIG [18], do not apply.

We therefore treat identifying $\mathbf{h}$ as a regression task, which we solve using kernel ridge regression with a Gaussian kernel [88]. Since the Gaussian kernel is universal, this constitutes a nonparametric regression technique with universal approximation capabilities, i.e., any nonlinear function can be approximated arbitrarily well given sufficient data.

---

[15]see [129] for further details

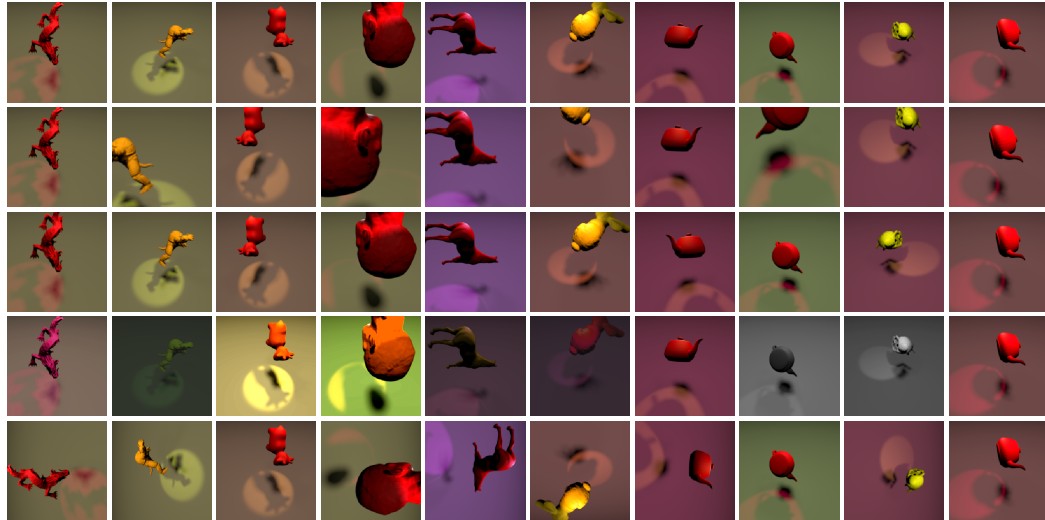

Figure 11: Visual overview of the effect of different data augmentations (DA), applied to 10 representative samples. Rows correspond to *(top to bottom)*: original images, small random crop (+ random flip), large random crop (+ random flip), colour distortion (jitter & drop), and random rotation.

We sample $4096 \times 10$ datapoints from the marginal for evaluation. For kernel ridge regression, we standardize the inputs and targets, and fit the regression model on $4096 \times 5$ (distinct) datapoints. We tune the regularization strength $\alpha$ and kernel variance $\gamma$ by 3-fold cross-validated grid search over the following parameter grids: $\alpha \in [1, 0.1, 0.001, 0.0001]$, $\gamma \in [0.01, 0.22, 4.64, 100]$.

**Compute.** The experiments in § 5.1 took on the order of 5-10 hours on a single GeForce RTX 2080 Ti GPU. The experiments in § 5.2 on 3DIdent took 28 hours on four GeForce RTX 2080 Ti GPUs. The creation of the Causal3DIdent dataset additionally required approximately 150 hours of compute time on a GeForce RTX 2080 Ti.