# OpenReview forum: "Self-Supervised Learning with Data Augmentations Provably Isolates Content from Style"
_NeurIPS.cc/2021/Conference — NeurIPS 2021 Poster_

### Official Review · Reviewer_Lc8M · 2021-07-14

**Rating:** 6
**Confidence:** 4

**Summary:**

This paper studies the effect of data augmentation in self-supervised representation learning. They take a latent variable perspective where there is a content component and style component of the latent variable, and consider settings where the content is invariant to the augmentation whereas the style component changes a lot under the data augmentation. In the latent variable model, arbitrary dependencies between the content and style latent variables are allowed.

The theoretical contribution of this paper is to show that, under certain assumptions on how the augmentation influences the latent variables, 1) a generative model for the data would perfectly identify the true content variables (up to an invertible function) and 2) minimizing the (entropy-regularized) squared difference between encoder outputs on two different augmentations of the data would also give an encoder which perfectly recovers the content variables (also up to an invertible function). The assumptions for these results are that the augmentations don't change the content latent variable and only the style latent variable, and that the augmentations can map the coordinates of the style latent variables to any value.

This paper also provides experiments for verifying aspects of the theory, first on synthetic Gaussian data, and also in a semi-synthetic setting of rendered images of 3D objects where different latents control different aspects of the image. They compare the effect of using InfoNCE objective on pairs of examples obtained via data augmentation against the effect of using InfoNCE objective on pairs of examples obtained via directly modifying the style latent variables. They show that training InfoNCE with modifying the latent variables does indeed preserve a lot of content information while discarding style information. Data augmentation is observed to behave in a similar way in some instances, though the equivalence is not exact.

**Limitations And Societal Impact:**

Yes

**Main Review:**

This paper is generally well-written, and the perspective of analyzing data augmentation through the influence on the latent variables is novel and quite interesting. However, there are shortcomings of the results: 1) the theoretical results are for a quite idealized setting with strong assumptions, and appear to be somewhat brittle and may not translate well outside of the idealized scenarios and 2) the empirical results are not very convincing in terms of mapping the data augmentations to the latent variable transformations. More detailed comments below.

Originality: the perspective of using data augmentation to disentangle content and style is quite interesting and novel, to my best knowledge.

Quality: The theorems are technically sound, but require strong assumptions to hold, and do not necessarily provide a complete picture on why data augmentation is useful in self-supervised learning. First, they require the augmentation to align perfectly with the style variables. In particular, Condition (iv) of Theorem 4.2 seems concerning because it requires the distribution on style change induced by the augmentation to be supported almost everywhere. This is likely unrealistic for data augmentations in practice. Second, in this result, there is no quantifiable bound on the complexity of the mapping between $\hat{z}$, the latent recovered in the identifiability result, and $z$, the true latent.

Furthermore, Theorem 4.3 and 4.4 require knowing the exact dimension of the content variable to set the dimension of the encoder -- this seems quite restrictive. Do the authors have a sense of what happens if there is a dimension mismatch here?

There also appears to be large mismatches between the theoretical model of data augmentation and what actually happens in practice in Table 1, though this is already addressed by the authors in Section 6.

Clarity: The paper is generally well-written, and the theoretical results are clear and easy to follow.

Significance: The idea of reasoning about data augmentation based on their effect on the latent space is potentially impactful and something for future work to build on.

========================================================================================

Update post rebuttal: I'd like to thank the authors for the clarifications and answers to my questions. My overall opinion of the paper is still the same, so I'm keeping my score at 6.

**Time Spent Reviewing:**

3

---

> ### Author Response · Authors · 2021-08-10
> **Response to Reviewer Lc8M**
>
> Thank you for the thoughtful feedback, some of which inspired us to revisit our assumptions and improve our results. We provide detailed responses to each of your comments below.
>
> ---
>
> >*“the theoretical results are for a quite idealized setting with strong assumptions, and appear to be somewhat brittle and may not translate well outside of the idealized scenarios”, “The theorems are technically sound, but require strong assumptions to hold, and do not necessarily provide a complete picture on why data augmentation is useful in self-supervised learning. First, they require the augmentation to align perfectly with the style variables. In particular, Condition (iv) of Theorem 4.2 seems concerning because it requires the distribution on style change induced by the augmentation to be supported almost everywhere. This is likely unrealistic for data augmentations in practice.”*
>
>
> We agree that our work does not *“provide a complete picture on why data augmentation is useful in self-supervised learning”*, but it does constitute a step in this direction. We also agree that we require some assumptions that may not hold in practice, as discussed in detail (l.376-397). On this point, we would like to (i) provide a clarification regarding a potential misunderstanding, and (ii) report an improvement triggered by your comment on condition (iv) of Thm. 4.2.
>
> - (i) **We actually do *not*  *“require the augmentation to align perfectly with the style variables”*.** The style variables are (implicitly) defined as those latents that may change under some of the augmentations, that is, for each augmentation/pair $(x,\tilde{x})$, not all style variables change, but only a subset thereof, see l.169, 172-173, Assumption 3.2, l.180-184, and assumption (iii) of Thm. 4.2. For example, if color distortion is applied with probability $0.5$ and otherwise flipping, then both color and orientation latents would be style even if they will not change across views half of the time.
>
> - (ii) We agree that assumption (iv) of Theorem 4.2 is strong---much stronger than necessary, in fact. Upon revisiting our proofs, **we realised that we can strongly relax assumption (iv)**: we actually only need it to hold for at least one subset $A$ with $p_A(A)>0$ for each style coordinate from assumption (iii); and, more importantly, we also do not require $p_{\tilde{s}_A|s_A}$ to be fully supported everywhere, but only within a local neighbourhood around the original style value $s_A$. More formally, we can replace assumptions (iii) and (iv) with a single new assumption:
> >**New Assumption (iii)’:** for any style coordinate $l\in\{1, ..., n_s\}$, $\exists A\subseteq\{1, ..., n_s\}$ s.t. $l\in A$; $p_A(A)>0$; $p(\tilde{s}_A|s_A)$ is smooth with respect to both $\tilde{s_A}$ and $s_A$; and for any $s_A$, $p(\cdot|s_A)>0$ within some open, non-empty set containing $s_A$.
>
> This means that we no longer require any (arbitrarily large) style changes to have positive probability, but instead only require this for some small change, thus drastically weakening the assumptions and increasing the applicability of the result. We will acknowledge your feedback leading to this result.
>
> ---
>
> > *“there is no quantifiable bound on the complexity of the mapping between z^, the latent recovered in the identifiability result, and z, the true latent.”*
>
> This would most likely require additional assumptions on the mixing function and/or latent distribution; we preferred to consider the general case with arbitrary nonlinear mixing and arbitrary $p(z)$. We believe that stronger notions of identifiability may be possible under stronger assumptions as in existing ICA work that relies on independence, parametric assumptions, and/or auxiliary variables [36,51,52,55,59]. In the present work, we aimed to make our assumptions as realistic as possible for the data augmentation setting while still being able to prove that content can be isolated from style. Incorporating additional assumptions to attain stronger notions of identifiability is an interesting and relevant direction for future work.
>
> ---
>
> >*“Theorem 4.3 and 4.4 require knowing the exact dimension of the content variable to set the dimension of the encoder -- this seems quite restrictive. Do the authors have a sense of what happens if there is a dimension mismatch here?”*
>
> The inset figure in Section 5.2 provides an ablation on dim($\hat{c}$) for applying the common SimCLR augmentation of combining random crops and colour distortion on Causal3DIdent: we observe relatively robust behaviour to misspecification in $n_c$ in this case (see l.353-355).
>
> On a more intuitive level, if the learnt representation is lower-dimensional than the true content, it has insufficient capacity to encode all content information and should thus learn a lower-dimensional mixture of the true content. Conversely, if it is higher dimensional, it has extra capacity so that some style information will likely be encoded, too (as this can help increase the entropy of the representation).
>
> To test this in a more controlled setting, **we ran an additional ablation on the synthetic settings from Section 5.1** where all our assumptions are satisfied and only dim($\hat{c}$) is varied over $\{1, …, 10\}$. The **results are summarised in this figure:** https://ibb.co/zGG7dR1
> The four plots correspond to the four rows from the inset figure in Section 5.1, see plot titles for details. Here, the true $n_c=5$, and error bars are over three random seeds. We find the results, for the most part, match the above intuition.
>
> ---
>
> >*“the empirical results are not very convincing in terms of mapping the data augmentations to the latent variable transformations”, “There also appears to be large mismatches between the theoretical model of data augmentation and what actually happens in practice in Table 1, though this is already addressed by the authors in Section 6.”*
>
> We agree that the results obtained with data augmentation (DA) do not perfectly match those obtained by latent transformations (LT). This is one of the main findings of the experiment in Section 5.2, see l.361. However, this should not be considered as “not very convincing”, but rather as pointing to an important challenge faced when designing and applying data augmentations that are intended to leave specific factors invariant, since our results show that LTs form more effective positive pairs for SSL at identifying invariant latents and discarding those that vary, when compared to their DA counterparts which are humanly engineered to have a similar effect. Note that in response to reviewer `D9kF`, we now demonstrate this point not only with SimCLR, but with Barlow Twins (see https://ibb.co/TT5LdSC and our response to reviewer `D9kF` for further detail). This may have important implications for creating multiple views for SSL which are better aligned with LTs, e.g., by using subsequent frames from video, which is an interesting direction for future work.

---

### Official Review · Reviewer_Pvr4 · 2021-07-15

**Rating:** 7
**Confidence:** 4

**Summary:**

This paper studies identifiability of the latent representation when samples are augmented in a way that preserves the "content" component of the latent variable while changing the "style" component. By focusing on an approximate (asymptotic) form of the InfoNCE cost function, the paper provides theoretical results on identifiability and experimental results on synthetic datasets, where style variables depend on content variables.

**Limitations And Societal Impact:**

Yes.

**Main Review:**

The paper has some novelty and the theoretical development appears correct. However, I think the issues listed below overweigh its advantages.

Major:
- There is somewhat of a disconnect between the theory and experiments, some of which are discussed at the end of the paper. I would like to point out an aspect that was not discussed and that appears to be responsible for some of the discrepancy between the theory and the experimental results: the theory considers the content as a continuous variable, but the proposed dataset and the associated experiment has discrete object classes.

- Baseline comparisons are missing, especially in the second experiment. The cited literature, where the latent factors may be assumed to not depend on each other, would provide meaningful baselines.

- It would be very useful to include a non-synthetic dataset. Some common choices in the cited literature include MNIST, FashionMNIST, etc.

Minor:
- line 289: f^-1 acts on x, why is the domain Z?
- Would the authors please explain why the negative L2 is used in one experiment and cosine similarity in the other experiment? I wonder if there is insight to be gained through these choices.
- The paper focuses on images as input data, which is fine, but there are components of the development that are agnostic to the underlying data. It could be useful, perhaps in the introduction only, to consider unstructured datasets as well.

--- Originality: The theoretical development is useful and novel.

--- Quality: The paper is well written.

--- Clarity: The expressions and explanations are clear. Proofs are detailed enough.

--- Significance: I think the disconnect mentioned above and the already discussed caveats in the experimental results significantly decrease the significance of the paper.

------------------------- UPDATE -------------------------
I decided to increase my score from 5 to 6, mostly in consideration of the authors’ positive and satisfying response on discrete vs. continuous content latents. Please see the discussion with the authors.

------------------------- UPDATE 2 -------------------------
I decided to further increase my score from 6 to 7, after the latest improvements the authors suggested and a re-evaluation of the paper overall.


**Time Spent Reviewing:**

~7 hours

---

> ### Author Response · Authors · 2021-08-10
> **Response to Reviewer Pvr4**
>
> Thank you for your time and feedback. We address all of your (major & minor) questions and comments below.
>
> ---
>
> >*“There is somewhat of a disconnect between the theory and experiments [...] the theory considers the content as a continuous variable, but the proposed dataset and the associated experiment has discrete object classes.”*
>
> We agree that our theory does not account for discrete variables even though Causal3DIdent (and other datasets) contain discrete object classes. **We will add this point to the “Theory vs. practice.” paragraph in our Discussion. However, continuity is a very common assumption** that is also made for theoretical results in the related ICA and disentanglement literature studying identifiability [36,51,52,55,59,60,64,76,77,119], even though most disentanglement datasets also contain discrete factors.
>
> More generally, the aforementioned works typically make the following three common assumptions: (i) continuous latents, (ii) independent latents, (iii) invertibility. In our work, we also assume (i), but we relax (ii) throughout by allowing arbitrary dependence, and also (iii) in our main result, Thm. 4.4. The most closely related multi-view results [36,64,77,119] additionally assume that (iv) all latents change between views, which we also relax to only hold for (random subsets of) the subset of style latents.
>
> We agree that a gap remains between our theoretical results and common SSL practice: (i) we openly discuss this at the end of the paper; and (ii) our paper substantially decreases this gap relative to existing work for the practical setting of SSL with data augmentation. Note that Thm. 4.2 is sufficient to show identifiability, without directly resulting in an algorithm. We additionally show that identifiability can be achieved with discriminative approaches (Thm. 4.3), even if the architecture is not invertible (Thm. 4.4), as done in practice.
>
> ---
>
> >*“Baseline comparisons are missing, especially in the second experiment. The cited literature, where the latent factors may be assumed to not depend on each other, would provide meaningful baselines.”*
>
> ICA/disentanglement approaches make very different assumptions in terms of
> - the distribution $p(z)$ (factorised vs. arbitrary),
> - the relation between the pairs $(x,\tilde{x})$ (all factors change vs. only a subset), and
> - the type of identifiability (1:1 correspondence between ground truth and learnt factors vs. isolating the invariant part).
>
> Further, multi-environment/temporal ICA requires access to an auxiliary variable $u$ which renders the latents conditionally independent [51,52,55,59] or suitable temporal structure in the observations [42], neither of which hold in our setting. This only leaves multi-view works [36,64,77,119] as potential baselines.
> - SlowVAE [64] and AdaGVAE [77]---which assume independent latents, all of which may change, with sparse changes between pairs of latents---due to being VAE-based and relying on reconstruction, do not scale to 224x224x3 images. (This was also the main motivation for moving beyond Thm 4.2, see l.251-256.)
> - Multi-view ICA [36]---which also assumes independent latents all of which change between views with probability 1---requires invertibility of the encoder function, which makes it impossible to apply it to the high-dimensional images in Causal3DIdent.
> - [119] does not propose a new algorithm but shows that contrastive learning with InfoNCE (e.g., SimCLR) identifies the latent variables under very different assumptions on the generative process (independent uniform latents on the hypersphere/hypercube + a von Mises-Fisher/truncated Normal/Laplace distribution as conditional, i.e., all latents change across views & there are no content variables). The learning approach is thus the same as in our experiments (InfoNCE) and we show that for our setting (LT), InfoNCE behaves consistently with our theory, which was one of the main points of the second experiment. Furthermore, SimCLR (InfoNCE w/ data augmentations) does not lead to the identification of all latents (the result of [119]), but instead leads to isolation of a subset of variables as our theory predicts (e.g., with colour distortion hue latents are discarded, see Table 1). Our results thus show that our theory predicts the behaviour of InfoNCE under data augmentation better than that of [119].
>
>
> Following a suggestion by reviewer `D9kF`, we carried out an additional experiment in which we ran BarlowTwins on Causal3DIdent. Since BarlowTwins is related to our result (Thm. 4.4) that SSL under data augmentation optimising for alignment and maximising entropy/minimising redundancy should isolate content (see l.306-310), this is a more meaningful comparison. Our main findings for SimCLR (l. 361-363) carry over for BarlowTwins, as the results are quite similar to those in Table 1 (see https://ibb.co/TT5LdSC). For more details, we refer to our response to reviewer `D9kF`.
>
> ---
>
> >*“It would be very useful to include a non-synthetic dataset. Some common choices in the cited literature include MNIST, FashionMNIST, etc.”*
>
> - To evaluate content identifiability (the main subject of our work) we **need the ground truth latents**. We, therefore, created the Causal3DIdent dataset where this is the case while allowing for latent dependence and approximately preserving the main characteristics of datasets to which SSL is typically applied (high-dimensional observations, here: 224x224x3 images; different objects; 3D rotation; lighting conditions). On (Fashion)MNIST or ImageNet, such ground truth is not available (or only available for a single latent factor: the digit/object class).
> - While numerous works have empirically studied downstream classification for SSL methods (i.e., predicting a single latent factor), one of the key aspects of our theory and experiments is that, depending on the used augmentations, we can also identify other latent factors such as position or colour. The **lack of colour information and centred objects** also makes (Fashion)MNIST less interesting in this regard.
> - Finally, we **would not be able to compare latent transformations (LT) and image-level data augmentations (DA)** on such datasets, which is an important component of our experiments.
>
> ---
>
> >*“the disconnect mentioned above and the already discussed caveats in the experimental results significantly decrease the significance of the paper”*
>
> As argued above, the *“disconnect”* between theory and practice is smaller in our work than in existing results, and we have made efforts to move our analysis closer to practice. The significance of our theoretical results (our main contribution) should not be decreased by *“caveats in the experimental results”*, given that our experiments were designed to study the remaining gap between SSL theory and practice. We will happily address any remaining concerns you may have.
>
> ---
>
> **OTHER MINOR POINTS**
>
> >*“line 289: f^-1 acts on x, why is the domain Z?”*
>
> Thank you for spotting this! This is a typo: $f^{-1}$ should be $f$, i.e., the line should read $h:=g \circ f: \mathcal{Z}\rightarrow (0,1)^{n_c}$. Note that the full proof (Appendix A.4, l.902) contains the correct form of $h$. We will fix the typo in the main text.
>
> ---
>
> >*“Would the authors please explain why the negative L2 is used in one experiment and cosine similarity in the other experiment?”*
>
> Our first experiment was designed to validate our theory by considering an idealised setting where our assumptions were fulfilled. As Theorem 4.4 assumes an objective function that corresponds to InfoNCE with a negative L2 similarity metric, we used the corresponding objective.
>
> Our second experiment was designed to systematically evaluate SSL with data augmentations through a comparison with latent transformations which correspond to the theory. To evaluate SSL in practice, we used SimCLR for these experiments, as of all state-of-the-art methods proposed for high-dimensional images, SimCLR is closest to the assumptions of Theorem 4.4. However, there are (minor) discrepancies, including the fact that SimCLR uses the cosine similarity metric in favour of negative L2. We will elaborate on this in the revision.
>
> ---
>
> >*“The paper focuses on images as input data, which is fine, but there are components of the development that are agnostic to the underlying data. It could be useful, perhaps in the introduction only, to consider unstructured datasets as well.”*
>
> We agree: our theory is agnostic of the type of observations and also applies to other types of data. We will emphasise this more. In our experiments and to build intuition, we focus on still images as this is one of the most common and successful application domains for SSL with data augmentation.

---

> > ### Comment · Reviewer_Pvr4 · 2021-08-25
> > **continue discussing**
> >
> > The authors responded openly to my concerns. I would like to continue discussing:
> >
> > - While I don’t expect the authors to run new experiments in this limited time, I’d like to mention that it shouldn’t be too hard, for instance, to extract rotation and stroke thickness estimates directly from MNIST images. (e.g., angle between y-axis and principal axis of inertia of the active pixels) With the caveat that real life datasets always come with issues, including a non-synthetic dataset would have added a lot of value to the paper. Such a dataset wouldn’t be a replacement of the synthetic data presented in the paper, rather it would be an addition. The current response sounds dangerously close to saying that not many datasets of interest are suitable for this method. I suggest careful editing of the discussion surrounding this point.
> > - I am generally satisfied with the authors’ response on discrete vs continuous latents. However, could I ask that this point is raised in the introduction instead of (or in addition to) the discussion? I think the current flow is prone to confusing the reader in terms of the disconnect I discussed.
> > - I still think not comparing with baselines is a weakness. The authors are correct that the assumptions of the other methods are different, but I think this is all the more reason to compare, preferably on a non-synthetic dataset where no assumption is perfectly satisfied, and show that this new set of assumptions is more advantageous in practice.

---

> > > ### Author Response · Authors · 2021-08-26
> > > **Continued discussion**
> > >
> > > Thank you for your response, we are happy to continue the discussion. We address each of your points separately.
> > >
> > > ___
> > >
> > > ### Synthetic Datasets
> > >
> > > > *“With the caveat that real life datasets always come with issues, including a non-synthetic dataset would have added a lot of value to the paper. Such a dataset wouldn’t be a replacement of the synthetic data presented in the paper, rather it would be an addition.”*
> > >
> > > It appears that there may be a misunderstanding regarding the source of potential violations of our assumptions. To clarify, we need to distinguish two separate processes (and the corresponding assumptions):
> > > 1. How the original (single-view) dataset is generated---here, our assumptions are minimal since we allow for arbitrary $p(z)$ and arbitrary nonlinear mixings $f$.
> > > 2. How such a dataset is turned into a multi-view one---this is where our main assumptions (content invariance and style changes under data augmentation) come into play.
> > >
> > > Since our assumptions are only restrictive regarding point 2., critical potential “caveats" encountered in practice concern how the views are generated. This is precisely what we study in our second set of experiments (Sec. 5.2) where views perfectly matching our assumptions are denoted by “LT”, while views generated by SimCLR augmentations are denoted by “DA”. The former correspond to a synthetic setting in which we have access to the ground truth latents, but **the latter constitutes a more realistic, non-synthetic setting** as encountered in practice (e.g., by applying SimCLR augmentations to a given dataset).
> > >
> > > We therefore believe that our empirical results already validate our theoretical results compellingly, in that we apply real data augmentations used in practice as opposed to just synthetic latent transformations. Given point 1., the fact that the data is generated with Blender is of secondary importance.
> > >
> > > > *“The current response sounds dangerously close to saying that not many datasets of interest are suitable for this method.”*
> > >
> > > We strongly disagree with this statement. **We do not propose a method**; we propose a theory for analyzing SSL methods with data augmentation as very commonly applied to image datasets in practice. In our experiments, we use annotations of all latent content and style variables to exactly measure whether SimCLR or BarlowTwins learn representations matching our theoretical predictions, despite the mismatch in assumptions.
> > >
> > > ___
> > >
> > > ### Discrete Factors
> > >
> > > > *“I am generally satisfied with the authors’ response on discrete vs continuous latents. However, could I ask that this point is raised in the introduction instead of (or in addition to) the discussion? I think the current flow is prone to confusing the reader in terms of the disconnect I discussed.”*
> > >
> > > We also found this discussion useful and will update the introduction accordingly to avoid confusion.
> > >
> > > ___
> > >
> > > ### Baselines
> > >
> > > > *“The authors are correct that the assumptions of the other methods are different, but I think this is all the more reason to compare [...] and show that this new set of assumptions is more advantageous in practice.”*
> > >
> > > Could you perhaps clarify what is meant by “more advantageous in practice”? Since we do not propose a method, **the only “empirical” comparison that can be made is across different theories analyzing the same method using *real* augmentations.**
> > >
> > > **Comparing our results with the theory of [119]:** Our experiments in Sec. 5 are testing the same learning algorithm that was assumed by [119] (eq. (1); we elaborate on the connection to Thm. 4.4 in l.1064-1068 in the Appendix), using real data augmentation as done in [18] as opposed to only latent transformation as done in [119]. Our results correspond to what our theory predicts (Def. 4.1). The prediction of [119] would be that all latent factors are identified up to a simple transformation, which is clearly not the case, see the “DA” rows in Table 1. While neither our assumptions nor those of [119] are perfectly satisfied when analyzing the DA setting on Causal3DIdent, the results are consistent with our theory and not with that of [119], suggesting that our set of assumptions is more plausible for SSL with data augmentation.
> > >
> > > As discussed in our previous response, none of the other ICA theories applies to SimCLR with data augmentations.
> > >
> > > Finally, note that upon suggestion by reviewer `D9kF`, we performed our empirical evaluation on BarlowTwins, and the results (https://ibb.co/TT5LdSC), suggest that Barlow Twins, like SimCLR, also behaves as predicted by our theory.
> > >
> > > If this discussion on [119] was helpful, we will be happy to include it in the paper.

---

> > > > ### Comment · Reviewer_Pvr4 · 2021-08-28
> > > > **update**
> > > >
> > > > - Sorry, please disregard the ‘more advantageous in practice’ comment. That’s a mix-up on my behalf.
> > > > - Having read the authors’ response, I still view the lack of a non-synthetic experiment a weakness, especially considering that many papers cited in this manuscript study benchmark datasets such as MNIST. This would, among other things, help us gain a better sense on fundamental questions in this problem space: Are these augmentations easy to generate in practical applications? How confident should we feel about style/content separation in practical applications?
> > > > - Discussion on [119]: This is not necessary as far as I’m concerned - no reason to bloat the presentation. Thanks.
> > > >
> > > > I decided to increase my score from 5 to 6, mostly in consideration of the authors’ positive and satisfying response on discrete vs. continuous content latents.

---

> > > > > ### Author Response · Authors · 2021-09-01
> > > > > **New non-synthetic experiments**
> > > > >
> > > > > Following your suggestion, we quickly tried the **MPI3D-real** dataset from “On the Transfer of Inductive Bias from Simulation to the Real World: a New Disentanglement Dataset”, Gondal et al., NeurIPS 2019. This dataset contains $>1$ million *real* images with ground-truth latent annotations of physical 3D objects being moved by a robotic arm.
> > > > >
> > > > > As with Causal3DIdent, we applied SimCLR with random crops & flips and/or colour distortion as data augmentations (DA). We point out the following caveats:
> > > > > - MPI3D-real contains much lower resolution images (64x64) compared to ImageNet & Causal3DIdent (224x224).
> > > > > - We did not optimise the architecture---we used the standard convolutional encoder from the disentanglement_lib [76] (https://github.com/google-research/disentanglement_lib)---nor did we adapt the augmentation hyperparameters.
> > > > > - We found different objects to be visually (near) indistinguishable at this resolution, especially after applying DAs (see, e.g., https://ibb.co/bNRGP43 (color distortion), where only few patterns for the object and the background ring are visible). In order to verify this, we trained the same backbone as in our unsupervised experiment with *supervised* learning. The results of the supervised learning sanity check are summarised in the following table: https://ibb.co/vjJbtg6. Notably, *without augmentations*, the average $R^2$ values across three random seeds are $0.25$ and $0.61$ for shape and size respectively, compared to $\geq 0.9$ for the five other factors. With augmentations, these numbers drop further, e.g., with color distortion $0.11$ and $0.47$, respectively. Therefore, we only evaluate SimCLR with respect to the five other latent factors.
> > > > >
> > > > > **Results of the self-supervised SimCLR experiments are summarised in the following table:** https://ibb.co/Ry198mQ.
> > > > >
> > > > > *Subject to the above caveats*, **the results show a similar trend as those on Causal3DIdent**: DAs can affect many latent factors making the content-style separation less clean than on numerical data, but the overall trends are consistent. To highlight a few examples of the latter:
> > > > > - Background colour is maintained as content when applying crops but discarded as style under colour distortion. The same holds for object color, unless the cropped images are so small that the object is not visible.
> > > > > - The axes (joint positions) are maintained as content under colour distortion, but discarded as style under crops (as the arm may not be visible).
> > > > > - Camera height is preserved as content throughout as it is affected by neither crops nor colour distortion, and can be inferred based on the relative position of the robotic arm and the ring of the platform.
> > > > >
> > > > > These results add further support to the claims that
> > > > > 1. it can be difficult to target individual latents with DAs (as already stated in the main paper in l.361 and discussed further in l.381-386);
> > > > > 2. a content-style separation can still be observed (to some extent) for real-world datasets, validating our position that our results are not restricted to synthetic scenarios.
> > > > >
> > > > >
> > > > > We will include (a more polished version) of the above results & discussion in the revised manuscript. We hope that this satisfactorily addresses your points regarding non-synthetic datasets.
> > > > >
> > > > > ---
> > > > >
> > > > > Final remarks on MNIST: We’d like to point out that none of the existing work in identifiability and disentanglement cited in the manuscript evaluated w.r.t. any annotated ground-truth latents on benchmark datasets, such as MNIST [9,12,16,22,36,42,46,51-55,59-61,64,66,71,76,77,95,96,102,103,119]. Of the works which did experiment with a benchmark dataset, they either (a) looked at qualitative measures of disentanglement, i.e. latent traversals [61, 103] or (b) computed the similarity between learned representations with distinct random initializations [60, 96]. We picked MPI3D over MNIST since it’s much bigger, has slightly higher resolution, and comes with annotated factors we can immediately use to validate our results.

---

> > > > > > ### Comment · Reviewer_D9kF · 2021-09-02
> > > > > > **Supervised vs Self-supervised**
> > > > > >
> > > > > > Interesting results. It is a little odd that camera height and background color achieve perfect $R^2$ score under self-supervision but it only achieves smaller score under supervised learning. In other latent factors this trend is reversed. Do you have any explanation for this?

---

> > > > > > > ### Author Response · Authors · 2021-09-02
> > > > > > > **Explanation for Supervised vs Self-supervised**
> > > > > > >
> > > > > > > We thank the reviewer for noticing this discrepancy in the results. This is due to the fact that in the self-supervised case, we evaluate by training a nonlinear regression **for each ground truth factor separately**, while in the supervised case, we train a network **for all ground truth factors simultaneously** from scratch for as many gradient steps as used for learning the self-supervised model.
> > > > > > >
> > > > > > > Note that the supervised experiment was intended as a quick sanity check for confirming that indeed shape & size were not adequately visible. Regardless, we will revise the supervised experiment and verify that this discrepancy is resolved for the final version.

---

> > > > > > > > ### Comment · Reviewer_D9kF · 2021-09-02
> > > > > > > > **Why not report shape and size?**
> > > > > > > >
> > > > > > > > Thanks for the clarification. Please provide reasons for this discrepancy in the next revision.
> > > > > > > >
> > > > > > > > Even though it is hard to learn shape and size, why do the authors not report the R^2 scores for shape and size? This is not expensive as it doesn't require re-training self-supervised model.

---

> > > > > > > > > ### Author Response · Authors · 2021-09-02
> > > > > > > > > **Reporting shape and size**
> > > > > > > > >
> > > > > > > > > Thanks for the continued discussion. The reviewer is correct that it is indeed not expensive to evaluate the self-supervised model on shape and size. Our reasoning for only evaluating the self-supervised model on factors we could reliably decode was to avoid having additional uninformative columns which prevent the trends from being easily discernible from the presented table.
> > > > > > > > >
> > > > > > > > > However, given the reviewers’ comment, we realize that this unsolicited omission may have been presumptuous. Hence, we have taken the opportunity to evaluate the self-supervised model on shape and size, and have added said results as columns to the previously presented table, see https://ibb.co/4J4D4wK. Regardless of augmentation, we yield poor scores in decoding shape and size from the self-supervised model, which is unsurprising given we also yielded poor scores for said factors in our supervised experiments.
> > > > > > > > >
> > > > > > > > > We will include the full table in the revision, as well as provide reasons for the previously discussed discrepancy. We hope the reviewer agrees with this decision.
> > > > > > > > >
> > > > > > > > > =================
> > > > > > > > > EDIT: corrected the heading in the linked table for the third column "object size"

---

### Official Review · Reviewer_8wZr · 2021-07-17

**Rating:** 6
**Confidence:** 4

**Summary:**

This paper provides some soft theory towards explaining the effect of data augmentation on self-supervised learning. Specifically, data observations are generated from content information that is invariant to data augmentation and style variables that will be modified. The theoretical results stem from ICA and show that with a squared alignment loss or regularization one can identify the content variables.



**Limitations And Societal Impact:**

Yes.

**Main Review:**


It is very hard for me to identify the novelty of this paper.


First, I think the overall message of the paper, that data augmentation only modifies style information, is intuitive and believable. However, it is not entirely new.
The authors should have properly discussed and compared with a whole line of prior theoretical work on SSL (e.g. [1-6] and many more I have not listed), many of which share very similar conceptual messages as the paper. Whether it is named as 'content' as in this paper or latent variables in related papers are just a matter of writing style.

~~The theoretical part is not that satisfactory. Unlike existing work, the authors only establish the conditions to identify the learned representation, but it doesn't necessarily explain why downstream tasks can be learned with simple functions (such as linear functions) from the representations. This is more essential for SSL. Otherwise, it's not clear to me why the theoretical results are specifically for explaining the success of SSL and what its distinction to existing ICA work is.
Identifying an invariant representation has been extensively studied in prior work, as discussed by the authors, and I don't see how the theoretical results specifically apply to SSL.~~
==================
Post rebuttal: My original concern is that even if the paper proves that with infoNCE one can identify the content features, the relation between downstream task and the content feature still remains unclear and the sample complexity might still be high. This is why I thought that proving the identifiability alone was not sufficient to show the advantage of SSL. After reading the rebuttal and other reviews I realized this is not the focus of this paper. Proving that the style features being removed is quite meaningful itself, and it can be theoretical support on why the features learned by SSL will be useful for transfer learning.
==================

The dataset is quite interesting, but I guess it might be more suitable for some causal relation work. I guess it is not the main contribution of the paper anyway, since it's not newly collected but comes from a preprocessing and combination of different existing datasets.

To summarize, I think this paper did a lot of work but no result is good enough for the level of NeurIPS. Plus, significant prior work has been missing.


[1] Arora, Hrishikesh Khandeparkar, Mikhail Khodak, Orestis Plevrakis, and Nikunj Saunshi. A theoretical analysis of contrastive unsupervised representation learning, 2019

[2] Christopher Tosh, Akshay Krishnamurthy, and Daniel Hsu. Contrastive estimation reveals
topic posterior information to linear models, 2020

[3] JD Lee, Q Lei, N Saunshi, J Zhuo. Predicting what you already know helps: Provable self-supervised learning, 2020

[4] Tosh, Christopher, Akshay Krishnamurthy, and Daniel Hsu. "Contrastive learning, multi-view redundancy, and linear models, 2020

[5] Tsai, Yao-Hung Hubert, et al. "Self-supervised learning from a multi-view perspective".

[6] Tian, Yuandong, Xinlei Chen, and Surya Ganguli. "Understanding self-supervised learning dynamics without contrastive pairs."

**Time Spent Reviewing:**

5

---

> ### Author Response · Authors · 2021-08-10
> **Response to Reviewer 8wZr**
>
> Thank you for your time and for reviewing our work.
>
> We respectfully disagree with your assessment. Regarding the novelty and contributions of the paper, we kindly refer to l. 56-68, 74-84, and the summary by reviewer `D9kF` which provide a concise account.
>
> There appear to be three main criticisms and two minor points which we address below:
>
> ---
>
> >*“The theoretical results stem from ICA”, “it's not clear to me [...] what its distinction to existing ICA work is”*
>
> **We strongly disagree that our theoretical results stem from ICA.** While identifiability of learnt representations is indeed commonly studied in the field of ICA, our proof techniques (see Appendix A) are entirely unrelated to any existing ICA results we are aware of. In particular:
>
> - **Our assumptions, identifiability notion, learning signal, and proof techniques are different from prior work in ICA.** As emphasized throughout the manuscript (e.g., l. 42-68, 130-145, 186-188, 209-213, 249-250, 319-321, 414-419), our problem setting differs from both ICA and other identifiability theory [59,60,96,119]. Only the first step in the proof of Thm. 4.2 is inspired by [77], as clearly stated in footnote 7, although the assumptions of [77] do not hold in our case (see next point).
> - **All prior works assume that all latents are independent or conditionally independent given an auxiliary variable, while we allow for non-linear causal dependencies.** Further, we consider a different notion of identifiability (see Defn. 4.1) and rely on data augmentation instead of distribution changes across environments [51,55], transition probabilities across successive time-steps [42,52,64] or views [36,77,119]. Finally, unlike prior work on ICA, our theory allows for non-invertible encoders, matching common practice in SSL.
>
> ___
>
> >*“The authors should have properly discussed and compared with a whole line of prior theoretical work on SSL (e.g. [1-6] and many more I have not listed), many of which share very similar conceptual messages as the paper.”*
>
> Thank you for sharing these references. To avoid confusion with references from our paper, we will refer to them as [1*], [2*], etc in our response.
>
> - **Our theoretical analyses are related to identifiability as opposed to linear predictability for a specific downstream task.** In fact, we specify a generative process of both original data and augmented views and only make assumptions about this underlying generative process, irrespective of the task, which is in any case not known when learning the representation without extra supervision (see l. 56-68, 186-188, 209-210, 417-419). Clearly, our results highlight that the choice of augmentations directly impacts which downstream tasks can be solved from the pre-trained representation (irrespective of linearity).
> - **Our assumptions are weak compared to prior work.** We start with [1*,3*,4*,5*], who assume access to data $X$ and a self-supervised signal $S$ ($x$ and $\tilde{x}$ in our notation, respectively) and reason about sample complexity and predictive performance for a given downstream task $T$ (or $Y$) based on a representation $Z$ learnt by applying different SSL techniques to $(X, S)$. For their theoretical results, assumptions on the relationship between $(X, S, T)$ are needed: [1*,3*] assume (approximate) independence between the two views $(X, S)$ given the task $T$, while [4*,5*] rely on (approximate) independence between one view $X$ and the task $T$ given the other view $S$.
> - **Our assumptions are weaker than** [1*,3*] because the latter assume (approximate) independence between the two views given the task. In our case, $(x,\tilde{x})$ are not (approximately) conditionally independent given $c$ due to the backdoor path $x\leftarrow s \rightarrow \tilde{s} \rightarrow \tilde{x}$ (i.e., both views are still correlated through style commonalities).
> - **Our assumptions are weaker than** [4*,5*] because the latter assume (approximate) independence between one view $X$ and the task $T$ given the other view $S$. In our case, $(x,c)$ are not (approximately)  conditionally independent given $\tilde{x}$ due to both the direct ($c\rightarrow x$) and indirect ($c\rightarrow s \rightarrow x$) influences of content (i.e., the task/content does not depend on the original view only via the second view).
> - [2*] focuses on topic modelling and SSL in an NLP context and thus considers discrete observations. Moreover, they study a semi-supervised setting in which some labels for the prediction task are available. It mostly bears some similarity to our work in that they also specify a generative process, and study recovering the topic posterior.
> - [6*] studies the learning dynamics of non-contrastive SSL without negative pairs (e.g., SimSiam, MoCo) and focuses on understanding why architectural and training choices (predictor networks, stop-gradients, exponential moving averages, weight decay) can avoid collapsed representations. While providing valuable insights for linear networks, this work is quite different from our nonlinear setting and focus on identifiability.
> - **We will add these discussions to the related works of our paper**, but strongly disagree with the statement that these references undermine the novelty of our work in any way.
>
> ---
>
> >*The paper “doesn't necessarily explain why downstream tasks can be learned with simple functions (such as linear functions) from the representations” and it is not clear “why the theoretical results are specifically for explaining the success of SSL”*
>
> - **Our result explicitly highlights the dependency between the choice of data augmentation and the information preserved in the representation learned with InfoNCE.** Note that Thm. 4.2 would already be sufficient to show identifiability under our generative model and realistic assumptions about the effect of data augmentation. However, our goal was to provide identifiability results closer to common SSL learning objectives, which is why we provide the series of results Thm. 4.2 → Thm 4.3 → Thm. 4.4, extending the identifiability to the discriminative case (Thm 4.3) and without the invertibility assumption (Thm 4.4).
> - **Training with InfoNCE (asymptotically) leads to identifying the full content representation** (i.e., anything that is left invariant by augmentation) under arbitrary dependence between individual content variables and between content and other latent (style) variables. This result is about *what* information is encoded in the representation, irrespective of linearity (i.e., *how* the information is encoded) and without assuming a specific task. We do not claim that this result is the only explanation for the empirical success of SSL, but it clearly pinpoints the role of augmentations in terms of *what* information is encoded.
> - **Experiments on linear evaluation**: see l.337-338, footnote 14, l. 988-990, 1017-1019, and Tables 5, 7 and 9 in Appendix C for both sets of experiments repeated using linear evaluation. We observe similar behaviour for linear prediction as for the nonlinear case. In the main paper, we only report experiments for non-linear evaluation since it matches our theoretical notion of identifiability.
>
> ---
>
> **OTHER MINOR CONCERNS**
>
> >*“The dataset is quite interesting, but I guess it might be more suitable for some causal relation work ”  and “it is not the main contribution of the paper anyway”*
>
> The Causal3DIdent dataset bears some similarity to existing work (3DIdent from [120]), namely in that both use the blender rendering engine. However, Causal3DIdent has several new key characteristics (7 object classes instead of just a single one; causal dependence structure instead of independent latents; see l. 344-347) and was rendered completely from scratch, which took approximately 150 GPU hours (see l.1110). Full details are presented in Appendix B.
>
> ---
>
> >*“​Whether it is named as 'content' as in this paper or latent variables in related papers are just a matter of writing style.”*
>
> The distinction between latent variables and content is a crucial one to our work: the content variables $c$ are only a subset of all latent variables $z$ (which also include style, $z=(c,s)$) and are implicitly defined as the part of the representation that remains invariant under a given choice of augmentations, see l. 64-68, 165-184, 404-413.

---

> > ### Comment · Reviewer_8wZr · 2021-08-24
> > **Response to the rebuttal**
> >
> > I thank the authors for the detailed comments. I think most of my concerns are properly addressed. Plus I might have some misunderstandings on some of the claims of the paper. (I will reflect it in the original review.) Therefore I am willing to increase my score to 6.

---

### Official Review · Reviewer_D9kF · 2021-07-30

**Rating:** 8
**Confidence:** 4

**Summary:**

Two contributions of this work are (1) an identifiability result for self-supervised learning (SSL), (2) a new synthetic dataset and empirical results evaluating the theoretical results and it’s comparison to standard SSL techniques with data augmentation.

Theoretically, authors prove that using
(a) “anchor” samples $x$ from a generative model which generates data using (potentially dependent) content $c$ and style $s$ latent random factors through an invertible mapping $x=f(c,s)$, and
(b) positive samples generated by the same content signal $c$ and an augmented style signal $\tilde{s}$ which is independent of $c$ given $s$,
one can learn a perfect encoder $\hat{c} = g(x)$ using a regularized SSL loss or by generative modeling, such that $\hat{c}$ contains all the information to perfectly predict $c$ in a statistical sense. This means that under some regularity conditions, SSL approaches using data augmentation can learn to isolate content factors (invariant to augmentation) from style factors (which can vary under augmentations). This identifiability result follows the recently popular line of similar results in non-linear ICA and representation learning thoery. However, different from previous results this result does not need independence of latent (content or style) random factors.

Next authors provide a new synthetic dataset to quantify the SSL representation learning. They quantify the performance of SimCLR contrastive (different from non-contrastive theory) SSL representation learning method on this dataset under their generative model and standard data augmentation methods. This showcase both the validity and the limitations of their theory and assumption.

**Limitations And Societal Impact:**

Good discussion of limitations of the work and some discussion of the negative societal impact.

**Main Review:**

### Strengths
1. This is the first such identifiability result in SSL, which is able to explain the empirical success of SSL methods. This could lead to better framework to understand SSL methods which are currently poorly understood.
2. Although I couldn’t check the math very carefully, part of the analysis seems novel. These could be useful in future results.
3. Novel Benchmark dataset
4. Fair empirical study to validate their theory and assumption. Fair discussion of limitations of the work.
5. Good discussion to past relevant works.
6. Very clear exposition of the contributions. Provides most of the details of empirical study (see below).
7. Discussion of open theory questions.
8. Provides guidelines for selecting data augmentation. Also compares the usage of pre-final layer as downstream embedding (usage of poorly-understood “projection” MLP).

### Questions
1. No empirical comparison to state of the art contrastive methods like MoCo v2 and non-contrastive methods like SimSiam and Barlow Twins.
2. Although the theory is for non-contrastive SLL loss with regularization, the experiments use constrastive SimCLR loss. It not clear how (31) approximately match (5).
3. Table 1: What is “latent transformation (LT)”. It is not clear how this is done and what distribution was used for $P_A$ and $P(\tilde{s}|s)$?
4. line 313: What does "using CL to estimate the entropy term in (5)" mean?

### Other
1. Typo line 982: “Object hue is dependent on object class, 983 background hue, & object hue, see Tab. 4”
2. Might be good to use the color formatting used main text to the Appendix tables?

## Update
The author response and additional experiments reasonably address most of my concerns. As mentioned by reviewers ``8wZr`` and ``Lc8M``, one drawback for the current theory is that these guarantees do not provide any upper-bound for the complexity of the relation between content factors $c$ and the estimator content factors $\hat{c}$. Additionally, I would also like to note that these are not finite sample guarantees. However, one may overlook because of the paper’s novelty and the interesting insights it gives. I am keeping my score.

**Time Spent Reviewing:**

6

---

> ### Author Response · Authors · 2021-08-10
> **Response to Reviewer D9kF**
>
> Thank you for the thoughtful and encouraging feedback. We believe that your summary accurately captures the main points and contributions of our work.
>
> Thank you also for noting that *“different from previous results this result does not need independence of latent (content or style) random factors”*. Relaxing the strong independence assumption required for all previous identifiability results is indeed one of the central aspects of our work.
>
> We address your questions one by one below.
> ___
>
> >1. *“No empirical comparison to state of the art contrastive methods like MoCo v2 and non-contrastive methods like SimSiam and Barlow Twins.”*
>
> Our work does not propose a new SSL algorithm, nor does it aim to beat SOTA. Rather it is mostly theoretical in nature and aims to provide understanding for why SSL with data augmentation works by proving identifiability of the invariant part of the latent representation. Our experiments, therefore, aim at verifying our theory in a controlled setting (Sec. 5.1) and probing its validity when assumptions do not perfectly hold, as expected in practice (Sec. 5.2). We focus on SimCLR because it is directly related to the scenario covered by our theory, i.e., alignment with approximate entropy maximisation (see l. 296-304 in the main paper and our response to questions 2. and 4. below for more details).
>
> Such connections are much less obvious for SSL methods that rely on “architectural” regularisation such as SimSiam or MoCo v2, so it is not entirely clear what the added value of comparing with these works would be.
>
> As pointed out in the paper (l.306-310), the redundancy reduction approach of BarlowTwins may be viewed as more closely related to our theoretical result on entropy maximisation (Thm. 4.4). We, therefore, followed your suggestion and ran an additional experiment to investigate whether BarlowTwins behaves similarly to SimCLR when it comes to identifying content and discarding style. Specifically, we replaced the SimCLR objective with that of  BarlowTwins (keeping all else the same) and re-computed the first 8 rows of Table 1. **We find the results with Barlow Twins to mirror the results observed with SimCLR, see:** https://ibb.co/TT5LdSC ; $\lambda$ is the weight on the redundancy reduction term, where $\lambda=0.0051$ was selected for experiments in the original paper.  We thank the reviewer for suggesting this experiment, as it has allowed us to demonstrate that the insights gained from our experimentation with SimCLR hold as well for Barlow Twins.
> ___
>
> >2. *“Although the theory is for non-contrastive SLL loss with regularization, the experiments use constrastive SimCLR loss. It not clear how (31) approximately match (5).”*
>
> Objective (31) approximates (5) in the following sense as $K\rightarrow\infty$:
> - the first term $\frac{1}{K}\sum_{i=1}^K(g(x_i)-g(\tilde{x}_i))^2$ in (31)  being an empirical mean converges to its expectation, i.e., the first term in (5); and
> - the second term---when a factor $1/K$ is added in front of the sum over $j$, which only introduces an additive factor $-\log K$ and is thus equivalent to objective (31) for any finite $K$---converges to the negative entropy, i.e., the second term in (5) as shown by [114] (see point 4. below for more details).
>
> ___
>
> >3. *“Table 1: What is “latent transformation (LT)”. It is not clear how this is done and what distribution was used for $p_A$ and $p(\tilde{s}|s)$?”*
>
> For each latent transformation (LT), a subset of the ground truth latents are treated as content (kept fixed) while the remaining ones are treated as style and resampled. For example, for “LT: change hues” the three hue latents are style and all remaining ones content. $p_A$ thus assigns probability one to the full set of style variables. As explained in lines 1089-1094 in Appendix D, $p_{\tilde{s}|s}$ is a Gaussian with variance 1 centred at $s$ and truncated to the range $[-1,1]$, and to avoid rendering at training time, we use that sample from the dataset whose associated ground truth latent is closest (in L2 distance) to the latent for the augmented view obtained as described above.
> ___
>
> >4. *“line 313: What does "using CL to estimate the entropy term in (5)" mean?”*
>
> By this, we refer to the result of Wang & Isola [114], who show that the use of negative samples in contrastive learning is connected to entropy estimation. Specifically, they show that asymptotically (i.e., in the limit of infinitely many negative samples, $K\rightarrow\infty$) the denominator of InfoNCE constitutes a nonparametric (von Mises-Fisher kernel density resubstitution) estimator of the entropy $H(\mathbf{z})$ of $\mathbf{z}=\mathbf{f}^{-1}(\mathbf{x})$ (up to an additive constant which does not depend on the encoder), see lines 297-299 in the main paper and [114] (in particular, the paragraph ”Relation with feature distribution entropy estimation.” in Section 4.2) for a more detailed account. Since SimCLR uses InfoNCE as an objective, it can thus be seen as approximating the entropy term in (5) via contrastive learning (CL). We will improve the presentation by including a more detailed account of the corresponding result from [114] in Appendix D.
>
> ---
>
> We will fix the typo and adopt the same colour formatting used in the main paper also for the Appendix tables, as suggested.

---

> > ### Comment · Reviewer_D9kF · 2021-08-27
> > **Update comments**
> >
> > I thank the authors for their detailed response and extra experiments. These resonably address most of my questions.

---

### Author Response · Authors · 2021-08-10
**General response to all reviewers and the AC**

Thank you for the valuable reviews, pointing out that *“This is the first such identifiability result in SSL which is able to explain the empirical success of SSL methods”* (`D9kF`), *“the perspective of analyzing data augmentation through the influence on the latent variables is novel”*, and that *“the theorems are technically sound”* (`Lc8M`). The reviewers seem to agree that our theoretical development is *“novel”* and *“useful”* (`D9kF`, `Pvr4`), and the paper is *“clear”* and *“well-written”* (`Pvr4`, `Lc8M`).

The reviewers have some questions, mostly relating to the assumptions underlying our theoretical analysis, as well as their applicability to SSL as performed in practice, and our empirical results, which we address in detailed comments to each of the reviewers.

Prompted by the reviews, **we report two additional empirical results**:
- Following a suggestion by reviewer `D9kF`, **we repeat our experiments on Causal 3DIdent using BarlowTwins** [118], and find that the results (see https://ibb.co/TT5LdSC ) mirror the SimCLR results we reported in Table 1. For details, see our response to reviewer `D9kF`.
- Following a suggestion by reviewer `Lc8M`, **we perform an ablation for the assumption that** $\text{dim}(\hat{c})=n_c$ in the first experimental setting, where all of our (other) assumptions are met, (see https://ibb.co/zGG7dR1), complementing the ablation on $\text{dim}(\hat{c})$ on Causal 3DIdent (inset figure in Sec. 5.2). Results clarify what happens when there is a mismatch between the number of content variables and the dimension of the encoder; for details, see our response to reviewer `Lc8M`.

We are also pleased to report that, following a comment by reviewer `Lc8M`, **we revisited assumptions (iii) and (iv) of our theorems and were able to relax them** (fully supported conditionals for all subsets → conditionals supported in a local neighbourhood of the original style for at least one subset per style variable, see the detailed response to `Lc8M` for the formal statement) which makes our results more broadly applicable, further strengthening the paper.

We will acknowledge the anonymous reviewers in the final version of the paper.

---

> ### Author Response · Authors · 2021-09-01
> **Additional experiments on non-synthetic dataset**
>
> We added results on a non-synthetic dataset https://ibb.co/Ry198mQ, see our last answer to reviewer `Pvr4` for more detailed discussion: https://openreview.net/forum?id=4pf_pOo0Dt&noteId=V-qPjwHi5-b (click on “View 1 More Reply”).

---

### Decision · Program_Chairs · 2021-09-27

**Decision:**

Accept (Poster)

**Comment:**

The reviewers have reached a consensus that the paper is a good addition to the neurips. Please see the reviewers' discussion on the pros and cons for the paper.